# The genetic architecture of multimodal human brain age

Junhao Wen [1] ✉, Bingxin Zhao [2], Zhijian Yang[3], Guray Erus [3], Ioanna Skampardoni[3], Elizabeth Mamourian [3], Yuhan Cui[3], Gyujoon Hwang [3], Jingxuan Bao [4], Aleix Boquet-Pujadas[5], Zhen Zhou [3], Yogasudha Veturi[6], Marylyn D. Ritchie[7], Haochang Shou[3], Paul M. Thompson[8], Li Shen [4], Arthur W. Toga [9] & Christos Davatzikos [3]

The complex biological mechanisms underlying human brain aging remain incompletely understood. This study investigated the genetic architecture of three brain age gaps (BAG) derived from gray matter volume (GM-BAG), white matter microstructure (WM-BAG), and functional connectivity (FC-BAG). We identified sixteen genomic loci that reached genome-wide significance (P-value < 5×10$^{-8}$). A gene-drug-disease network highlighted genes linked to GM-BAG for treating neurodegenerative and neuropsychiatric disorders and WM-BAG genes for cancer therapy. GM-BAG displayed the most pronounced heritability enrichment in genetic variants within conserved regions. Oligodendrocytes and astrocytes, but not neurons, exhibited notable heritability enrichment in WM and FC-BAG, respectively. Mendelian randomization identified potential causal effects of several chronic diseases on brain aging, such as type 2 diabetes on GM-BAG and AD on WM-BAG. Our results provide insights into the genetics of human brain aging, with clinical implications for potential lifestyle and therapeutic interventions. All results are publicly available at https://labs.loni.usc.edu/medicine.

The advent of artificial intelligence (AI) has provided approaches to investigate various aspects of human brain health[1,2], such as normal brain aging[3], neurodegenerative disorders such as Alzheimer's disease (AD)[4], and brain cancer[5]. Based on magnetic resonance imaging (MRI), AI-derived measures of the human brain age[6–8] have emerged as a valuable biomarker for evaluating brain health. More precisely, the difference between an individual's AI-predicted brain age and chronological age – brain age gap (BAG) – provides a means of quantifying an individual's brain health by measuring deviation from the normative aging trajectory. BAG has demonstrated sensitivity to several common brain diseases, clinical variables, and cognitive functions[9], presenting the promising potential for its use in the general population to capture relevant pathological processes.

Brain imaging genomics[10], an emerging scientific field advanced by both computational statistics and AI, uses imaging-derived phenotypes (IDP[11]) from MRI and genetics to offer mechanistic insights

[1]Laboratory of AI and Biomedical Science (LABS), Stevens Neuroimaging and Informatics Institute, Keck School of Medicine of USC, University of Southern California, Los Angeles, CA, USA. [2]Department of Statistics and Data Science, University of Pennsylvania, Philadelphia, PA, USA. [3]Artificial Intelligence in Biomedical Imaging Laboratory (AIBIL), Center for AI and Data Science for Integrated Diagnostics (AI2D), Perelman School of Medicine, University of Pennsylvania, Philadelphia, PA, USA. [4]Department of Biostatistics, Epidemiology and Informatics, University of Pennsylvania Perelman School of Medicine, Philadelphia, PA, USA. [5]Biomedical Imaging Group, EPFL, Lausanne, Switzerland. [6]Department of Biobehavioral Health and Statistics, Penn State University, University Park, PA, USA. [7]Department of Genetics and Institute for Biomedical Informatics, Perelman School of Medicine, University of Pennsylvania, Philadelphia, PA, USA. [8]Imaging Genetics Center, Mark and Mary Stevens Neuroimaging and Informatics Institute, Keck School of Medicine of USC, University of Southern California, Marina del Rey, CA, USA. [9]Laboratory of Neuro Imaging (LONI), Stevens Neuroimaging and Informatics Institute, Keck School of Medicine of USC, University of Southern California, Los Angeles, CA, USA. ✉e-mail: junhaowe@usc.edu

into healthy and pathological aging of the human brain. Recent large-scale genome-wide association studies (GWAS)[11–17] have identified a diverse set of genomic loci linked to gray matter (GM)-IDP from T1-weighted MRI, white matter (WM)-IDP from diffusion MRI [fractional anisotropy (FA), mean diffusivity (MD), neurite density index (NDI), and orientation dispersion index (ODI)], and functional connectivity (FC)-IDP from functional MRI. While previous GWAS[18] have associated BAG with common genetic variants [e.g., single nucleotide polymorphism (SNP)], they primarily focused on GM-BAG[9,19–21] or did not comprehensively capture the genetic architecture of the multimodal BAG[18] via post-GWAS analyses in order to biologically validate the GWAS signals. It is crucial to holistically identify the genetic factors associated with multimodal BAGs (GM, WM, and FC-BAG), where each BAG reflects distinct and/or similar neurobiological facets of human brain aging. Furthermore, dissecting the genetic architecture of human brain aging may determine the causal implications, which is essential for developing gene-inspired therapeutic interventions. Finally, numerous risk or protective lifestyle factors and neurobiological processes may also exert independent, synergistic, antagonistic, sequential, or differential influences on human brain health. Therefore, a holistic investigation of multimodal BAGs is urgent to fully capture the genetics of human brain aging, including the genetic correlation, gene-drug disease network, and potential causality. In this study, we postulate that AI-derived GM, WM, and FC-BAG can serve as robust, complementary endophenotypes[22]–close to the underlying etiology–for precise quantification of human brain health.

The present study sought to uncover the genetic architecture of multimodal BAG and explore the causal relationships between protective/risk factors and decelerated/accelerated brain age. To accomplish this, we analyzed multimodal brain MRI scans from 42,089 participants from the UK Biobank (UKBB) study[23] and used 119 GM-IDP, 48 FA WM-IDP, and 210 FC-IDP to derive GM, WM, and FC-BAG, respectively. Refer to Method 1 for selecting the final feature sets for each BAG. We compared the age prediction performance of different machine learning models using these IDPs. We then performed GWAS to identify genomic loci associated with GM, WM, and FC-BAG in the European ancestry population. In post-GWAS analyses, we constructed a gene-drug-disease network, estimated the genetic correlation with several brain disorders, assessed their heritability enrichment in various functional categories or specific cell types, and calculated the polygenic risk scores (PRS) of the three BAGs. Finally, we performed Mendelian Randomization (MR)[24] to infer the potential causal effects of several clinical traits and diseases on the three BAGs.

## Results

We provide an overview of the main results from our experiments. First, we objectively compared the age prediction performance of four machine learning methods using the GM, WM, and FC-IDPs (Fig. 1A). To this end, we employed a nested cross-validation (CV) procedure in the training/validation/test dataset ($N = 4000$); an independent test dataset ($N = 38,089$)[25,26] was held out – unseen until we finalized the models using only the training/validation/test dataset (Method 1). The GM, WM, and FC-IDPs were derived from three MRI modalities (Method 2). The four machine learning models included support vector regression (SVR), LASSO regression, multilayer perceptron (MLP), and a five-layer neural network (i.e., three linear layers and one rectified linear unit layer; hereafter, NN)[27] (Method 3). We then performed the three primary GWASs using the European ancestry population ($31,557 < N < 32,017$) and extensively scrutinized the genetic signals in seven quality check scenarios (Method 4A). Finally, we validated the GWAS findings in several post-GWAS analyses, including genetic correlation, gene-drug-disease network, partitioned heritability, PRS calculation, and Mendelian randomization (Method 4).

## GM, WM, and FC-BAG derived from three MRI modalities and four machine learning models

Several findings were observed based on the results from the independent test dataset ($N = 38,089$, Method 1). First, GM-IDP ($4.39 < $ mean absolute error (MAE) $< 5.35$; $0.64 < r < 0.66$), WM-IDP ($4.92 < $ MAE $< 7.95$; $0.42 < r < 0.65$), and FC-IDP ($5.48 < $ MAE $< 6.05$; $0.43 < r < 0.46$) achieved gradually a higher MAE and smaller Pearson's correlation ($r$) (Fig. 1B and C). Second, LASSO regression obtained the lowest MAE for GM, WM, and FC-IDP; linear models obtained a lower MAE than non-linear networks (Fig. 1B). Third, all models generalized well from the training/validation/test dataset ($N = 4000$, Method 1) to the independent test dataset. However, simultaneously incorporating WM-IDP from FA, MD, NDI, and ODI resulted in severely overfitting models (Supplementary table 1A). The observed overfitting may be attributed to many parameters ($N = 38,364$) in the network or strong correlations among the diffusion metrics (i.e., FA, MD, ODI, and NDI). Fourth, the experiments stratified by sex did not exhibit substantial differences, except for a stronger overfitting tendency observed in females compared to males using WM-IDP incorporating the four diffusion metrics (Supplementary table 1B). In all subsequent genetic analyses, we reported the results using BAG derived from the three LASSO models with the lowest MAE in each modality (Fig. 1A), with the "age bias" corrected as in De Lange et al.[28].

Our age prediction results align with previous literature using low-dimensional imaging features, but the convolutional neural network (CNN) trained on voxel-wise MRI scans achieved a lower MAE. Other studies[29–32] have thoroughly evaluated age prediction performance using different machine learning models and input features. More et al.[33] systematically compared the performance of age prediction of 128 workflows (MAE between 5.23–8.98 years) and showed that voxel-wise feature representation (MAE approximates 5-6 years) outperformed parcel-based features (MAE approximates 6-9 years) using conventional machine learning algorithms (e.g., LASSO regression). Using deep neural networks, Peng et al.[29] and Leonardsen et al.[30] reported a lower MAE (nearly 2.5 years) with voxel-wise imaging scans. However, we previously showed that a moderately fitting CNN obtained significantly higher differentiation (a larger effect size) than a tightly fitting CNN (a lower MAE) between the disease and health groups[34]. In addition, we assessed the impact of a lower MAE using GWAS summary statistics shared by Leonardsen et al.[20] on the GWAS results in the sensitivity check analyses (Supplementary note 1).

Finally, we calculated the phenotypic correlation ($p_c$) between GM, WM, and FC-BAG using Pearson's correlation coefficient. GM-BAG and WM-BAG showed the highest positive correlation ($p_c = 0.38$; $P$ value $< 1 \times 10^{-10}$; $N = 30,733$); GM-BAG ($p_c = 0.09$; $P$ value $< 1 \times 10^{-10}$; $N = 30,660$) and WM-BAG ($p_c = 0.10$; $P$ value $< 1 \times 10^{-10}$; $N = 31,574$) showed weak correlations with FC-BAG (Fig. 1D).

## GM, WM, and FC-BAG are associated with sixteen genomic loci

In the European ancestry populations, GWAS (Method 4 A) revealed 6, 9, and 1 genomic loci linked to GM ($N = 31,557$), WM ($N = 31,674$), and FC-BAG ($N = 32,017$), respectively (Fig. 2A). The top lead SNP and mapped genes of each locus are presented in Supplementary Table 2. The three BAGs were significantly heritable ($P$ value $< 1 \times 10^{-10}$) after adjusting for multiple comparisons using the Bonferroni method using the genome-wide complex trait analysis (GCTA) software[35]. GM-BAG showed the highest SNP-based heritability ($h^2 = 0.47 \pm 0.02$), followed by WM-BAG ($h^2 = 0.46 \pm 0.02$) and FC-BAG ($h^2 = 0.11 \pm 0.02$). Our GM-BAG showed a higher SNP-based heritability than several previous GM-BAG GWAS[9,20,21] ($0.19 < h^2 < 0.27$), which used the linkage disequilibrium score regression (LDSC) software[36]. LDSC uses GWAS summary statistics but not the individual genotype data as in GCTA. This discrepancy may depend on the choice of methods, genetic data employed, underlying statistical assumptions, and allele frequency[37,38].

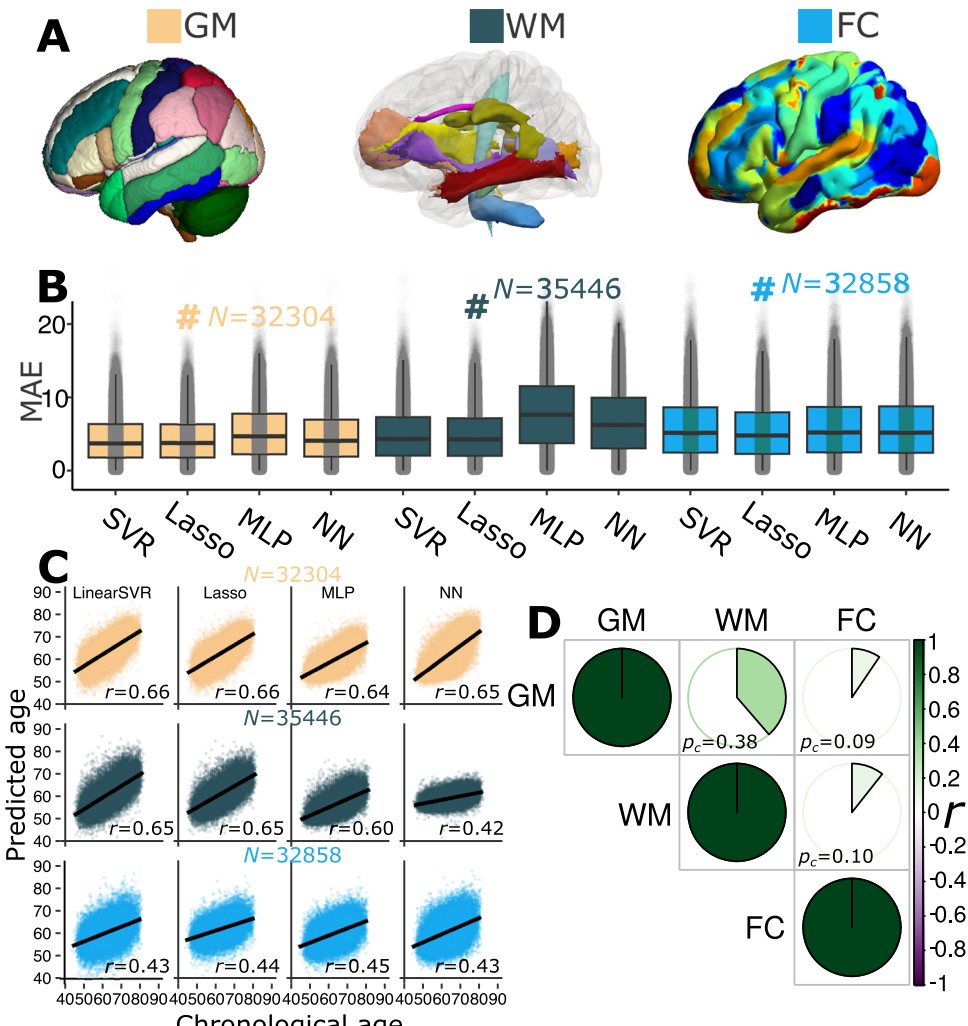

**Fig. 1 | Brain age prediction using three MRI modalities and four machine learning models. A** Multimodal brain MRI data were used to derive imaging-derived phenotypes (IDP) for T1-weighted MRI (119 GM-IDP), diffusion MRI (48 WM-IDP), and resting-state functional MRI (210 FC-IDP). IDPs for each modality are shown here using different colors based on predefined brain atlases or ICA for FC-IDP. **B** Linear models achieved lower mean absolute errors (MAE) than non-linear models using support vector regression (SVR), LASSO regression, multilayer perceptron (MLP), and a five-layer neural network (NN). The MAE for the independent test dataset is presented ($N$ is shown in the figure), and the # symbol indicates the model with the lowest MAE for each modality. We present the results using box plots for the median and jitter plots for underlying MAE distributions. **C** Scatter plot for the predicted brain age and chronological age. Pearson's correlation ($r$) and $N$ are presented for all feature and model combinations. **D** Phenotypic correlation ($p_c$) between the GM, WM, and FC-BAG using Pearson's correlation coefficient ($r$).

We showed the robustness of our GWAS findings with several different approaches. We first calculated the genomic inflation factor ($\lambda$) and the LDSC intercept ($b$) for the GWAS of GM-BAG ($\lambda = 1.118$; $b = 1.0016 \pm 0.0078$), WM-BAG ($\lambda = 1.124$; $b = 1.0187 \pm 0.0073$), and FC-BAG ($\lambda = 1.046$; $b = 1.0039 \pm 0.006$). All LDSC intercepts were close to 1, indicating no substantial genomic inflation. The individual Manhattan and QQ plots of the three GWASs are presented in Supplementary Fig. 1 and are publicly available at the MEDICINE knowledge portal: https://labs.loni.usc.edu/medicine. We also checked the robustness of the main GWASs using the European populations (Fig. 2A) via seven sensitivity analyses (Method 4 A). Overall, the primary GWASs were robust across sexes (female vs. male), random splits, imaging features (ROI vs. voxel-wise images), GWAS methods (linear vs. mixed linear model[39]), and machine learning methods (Lasso regression vs. SVR vs. CNN[20]); however, their generalizability to non-European populations ($4646 < N < 5091$) and independent disease-specific populations (i.e., ADNI[40], $N = 1104$) is limited potentially due to the small sample sizes. It's worth noting that their $\beta$ values compared to the primary GWASs were significantly correlated: $r = 0.83$ for ADNI and $r = 0.97$–$0.99$ for

the non-European populations. (Supplementary note 1, Supplementary data 1–7, and Supplementary Figs. 1–7). All subsequent post-GWAS analyses were conducted using the main GWAS results of European ancestry.

We performed a query in the GWAS Catalog[41] for these genetic variants within each locus to understand the phenome-wide association of these identified loci in previous literature (Method 4B). Notably, the SNPs within each locus were linked to other traits previously reported in the literature (Supplementary data 8). Specifically, the GM-BAG loci were uniquely associated with neuropsychiatric disorders such as major depressive disorder (MDD), heart disease, and cardiovascular disease. We also observed associations between these loci and other diseases (e.g., anemia), as well as biomarkers from various human organs (e.g., liver) (Fig. 2B). We then performed positional and functional annotations to map SNPs to genes associated with GM, WM, and FC-BAG loci (Method 4C). Figure 2C-E showcased the regional Manhattan plot of one genomic locus linked to GM, WM, and FC-BAG. A detailed discussion of these exemplary loci, SNPs, and genes is presented in Supplementary note 2.

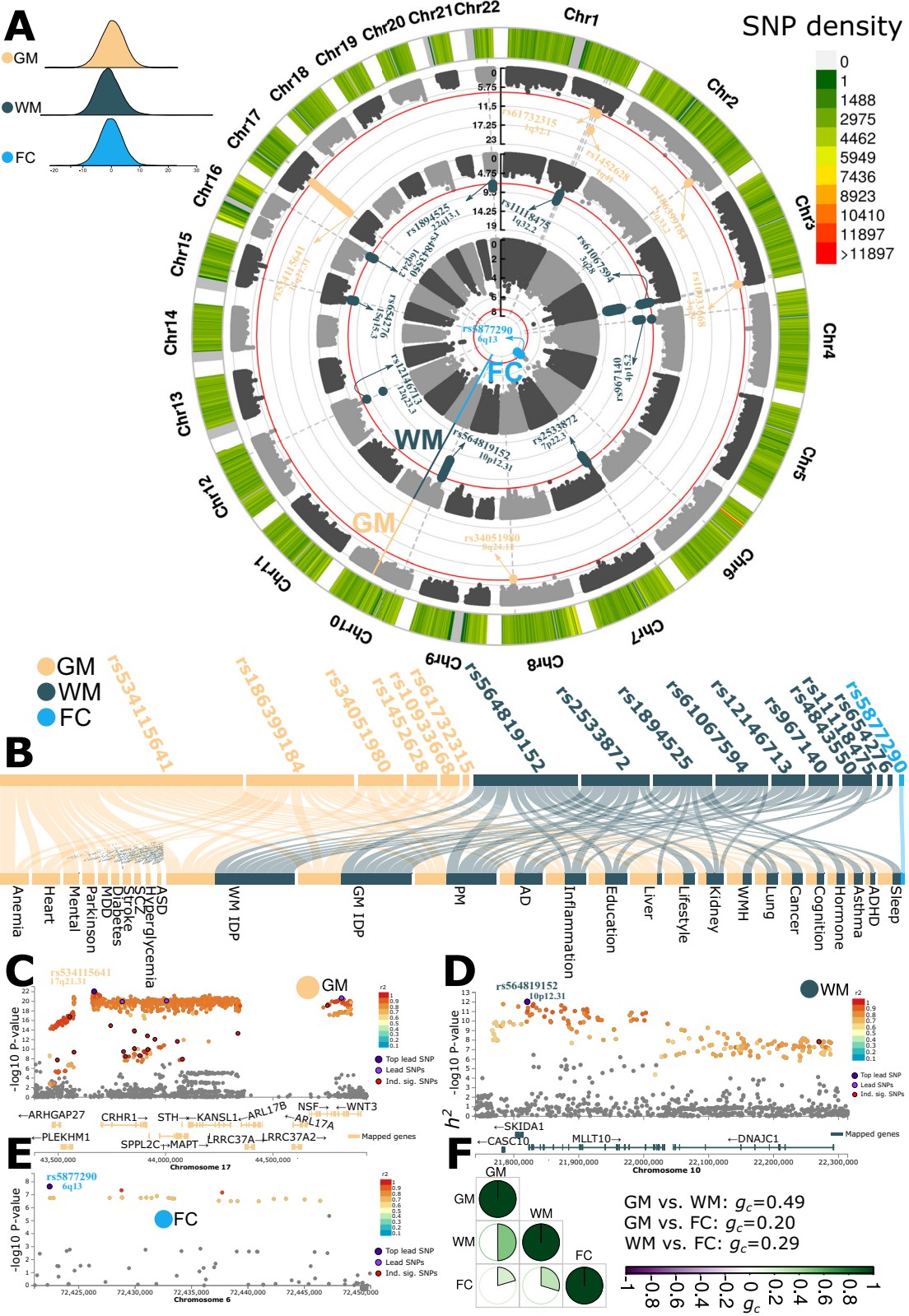

Finally, we calculated the genetic correlation ($g_c$) between the GM, WM, and FC-BAG using the LDSC software. GM-BAG and WM-BAG showed the highest positive correlation ($g_c = 0.49$; $P$ value $< 1 \times 10^{-10}$); GM-BAG ($g_c = 0.20$; $P$ value $= 0.025$) and WM-BAG ($g_c = 0.29$; $P$ value $= 0.005$) showed weak correlations with FC-BAG (Fig. 2F). We also verified that these genetic correlations exhibited consistency between the two random splits (split1 and spit2:

$15,778 < N < 16,008$), sharing a similar age and sex distribution (Supplementary Fig. 8).

**The gene-drug-disease network highlights disease-specific drugs that bind to genes associated with GM and WM-BAG**
After mapping the SNPs to genes (Method 4C), we investigated the potential "druggable genes[42]" by constructing a gene-drug-disease

**Fig. 2 | Genome-wide associations of multimodal brain age gaps. A** Genome-wide associations identified sixteen genomic loci associated with GM (6), WM (9), and FC-BAG (1) using a genome-wide *P* value threshold [−log₁₀(*P* value) > 7.30]. The top lead SNP and the cytogenetic region number represent each locus. **B** Phenome-wide association query from GWAS Catalog[41]. Candidate SNPs inside each locus were largely associated with many traits. We further classified these traits into several trait categories, including biomarkers from multiple body organs (e.g., heart and liver), neurological disorders (e.g., Alzheimer's disease and Parkinson's disease), and lifestyle risk factors (e.g., alcohol consumption). **C** Regional plot for a genomic locus associated with GM-BAG. Color-coded SNPs are decided based on their highest $r^2$ to one of the nearby independent significant SNPs. Gray-colored

SNPs are below the $r^2$ threshold. The top lead SNP, lead SNPs, and independent significant SNPs are denoted as dark purple, purple, and red, respectively. Mapped, orange-colored genes of the genomic locus are annotated by positional, eQTL, and chromatin interaction mapping (Method 4C). **D** Regional plot for a genomic locus associated with WM-BAG. **E** The genomic locus associated with FC-BAG did not map to any genes. We used the Genome Reference Consortium Human Build 37 (GRCh37) in all genetic analyses. For Figure (**C**–**E**), the two-sided P-value was derived from the linear regression used in our GWAS. **F** Genetic correlation ($g_c$) between the GM, WM, and FC-BAG using the LDSC software. Abbreviation: AD Alzheimer's disease, ASD autism spectrum disorder, PD Parkinson's disease, ADHD attention-deficit/hyperactivity disorder.

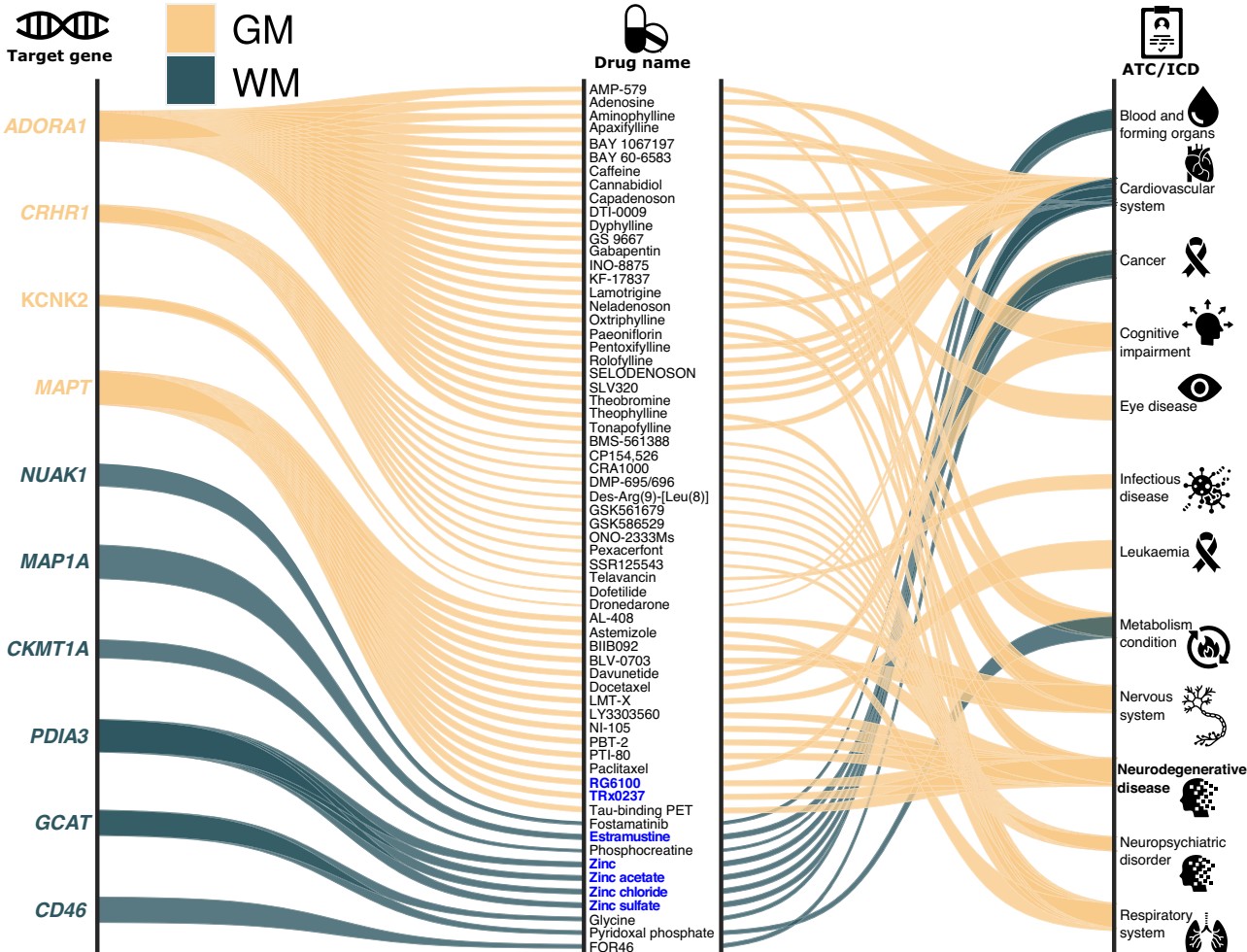

**Fig. 3 | Gene-drug-disease network of multimodal brain age gaps.** The gene-drug-disease network derived from the mapped genes revealed a broad spectrum of targeted diseases and cancer, including brain cancer, cardiovascular system diseases, Alzheimer's disease, and obstructive airway disease, among others. The

thickness of the lines represented the *P* values −log₁₀) from the brain tissue-specific gene set enrichment analyses using the GTEx v8 dataset. We highlight several drugs under the blue-colored and bold text. Abbreviation: ATC Anatomical Therapeutic Chemical, ICD International Classification of Diseases.

network (Method 4D). The network connects genes with drugs (or drug-like molecules) targeting specific diseases currently active at any stage of clinical trials.

We revealed four and six mapped genes associated with GM-BAG and WM-BAG currently used in clinical trials. The GM-BAG genes were linked to clinical trials for treating heart, neurodegenerative, neuropsychiatric, and respiratory diseases. On the other hand, the WM-BAG genes were primarily targeted for various cancer treatments and cardiovascular diseases (Fig. 3). For example, the GM-BAG-associated *MAPT* gene was involved in several drugs or drug-like molecules currently being evaluated for treating AD. Semorinemab (RG6100), an anti-tau IgG4 antibody, was being investigated in a phase-2 clinical trial

(trial number: NCT03828747), which targets extracellular tau in AD, to reduce microglial activation and inflammatory responses[43]. Another drug is the LMTM (TRx0237)—a second-generation tau protein aggregation inhibitor currently being tested in a phase-3 clinical trial (trial number: NCT03446001) for treating AD and frontotemporal dementia[44]. Regarding WM-BAG genes, they primarily bind with drugs for treating cancer and cardiovascular diseases. For instance, the *PDIA3* gene, associated with the folding and oxidation of proteins, has been targeted for developing several zinc-related FDA-approved drugs for treating cardiovascular diseases. Another example is the *MAP1A* gene, which encodes microtubule-associated protein 1 A. This gene is linked to the development of estramustine, an FDA-approved drug for

prostate cancer (Fig. 3). Detailed results are presented in Supplementary data 9.

## Multimodal BAG is genetically correlated with AI-derived subtypes of brain diseases

We showed specific phenome-wide associations of the three BAGs at the single variant level (Fig. 2B). Here, we calculated the genetic correlation using the GWAS summary statistics to examine genetic covariance between the multimodal BAGs and 16 clinical traits. The selection procedure and quality check of the GWAS summary statistics are detailed in Method 4E. These traits encompassed common brain diseases and their AI-derived disease subtypes, as well as education and intelligence (Fig. 4A and Supplementary Table 3). The AI-generated disease subtypes were established in our previous studies utilizing semi-supervised clustering methods[45] and IDP from brain MRI scans. To illustrate this, AD1 and AD2 distill the neuroanatomical heterogeneity of Alzheimer's disease into two distinct imaging patterns: AD1 represents a widespread brain atrophy pattern, while AD2 exhibits a focal atrophy pattern in the medial temporal lobe[4]. These subtypes, in essence, capture more homogeneous disease effects than the conventional "unitary" disease diagnosis, hence serving as robust endophenotypes[22].

Our analysis revealed significant genetic correlations between GM-BAG and AI-derived subtypes of AD (AD1[4]), autism spectrum disorder (ASD) (ASD1 and ASD3[46]), schizophrenia (SCZ1[47]), and obsessive-compulsive disorder (OCD)[48]; WM-BAG and AD1, ASD1, SCZ1, and SCZ2; and FC-BAG and education[49] and SCZ1. Detailed results for $r_g$ estimates are presented in Supplementary data 10. Furthermore, we found that the WM BAG ($g_c = -0.23 \pm 0.10$; $P$ value = 0.02; $N = 28,967$ European ancestry) was negatively associated with longevity, defined as cases surviving at or beyond the age corresponding to the 99th survival percentile[50].

## Multimodal BAG shows specific enrichment of heritability in different functional categories and cell types

As the three BAGs showed significant SNP-based heritability estimates, we conducted a partitioned heritability analysis[51] to investigate further the heritability enrichment of these genetic variants in the 53 functional categories and specific cell types (Method 4F).

For GM-BAG, the regions conserved across mammals, as indicated by the label "conserved" in Fig. 4B, displayed the most notable enrichment of heritability: ~2.61% of SNPs were found to explain $0.43 \pm 0.07$ of SNP heritability ($P$ value = $5.80 \times 10^{-8}$). Additionally, transcription start site (TSS)[52] regions employed 1.82% of SNPs to explain $0.16 \pm 0.05$ of SNP heritability ($P$ value = $8.05 \times 10^{-3}$). TSS initiates the transcription at the 5′ end of a gene and is typically embedded within a core promoter crucial to the transcription machinery[53]. The heritability enrichment of Histone H3 at lysine 4, as denoted for "H3K4me3_peaks" in Fig. 4B, and histone H3 at lysine 9 (H3K9ac)[54] were also found to be large and were known to highlight active gene promoters[55]. For WM-BAG, 5′ untranslated regions (UTR) used 0.54% of SNPs to explain $0.09 \pm 0.03$ of SNP heritability ($P$ value = $4.24 \times 10^{-3}$). The 5′ UTR is a crucial region of a messenger RNA located upstream of the initiation codon. It is pivotal in regulating transcript translation, with varying mechanisms in viruses, prokaryotes, and eukaryotes.

Additionally, we examined the heritability enrichment of the three BAG in three different cell types (Fig. 4C). WM-BAG ($P$ value = $1.69 \times 10^{-3}$) exhibited significant heritability enrichment in oligodendrocytes, one type of neuroglial cells. FC-BAG ($P$ value = $1.12 \times 10^{-2}$) showed such enrichment in astrocytes, the most prevalent glial cells in the brain. GM-BAG showed no enrichment in any of these cells. Our findings are consistent with understanding the molecular and biological characteristics of GM and WM. Oligodendrocytes are primarily responsible for forming the lipid-rich myelin structure, whereas astrocytes play a crucial role in various cerebral

functions, such as brain development and homeostasis. Convincingly, a prior GWAS[14] on WM-IDP also identified considerable heritability enrichment in glial cells, especially oligodendrocytes. Detailed results for the 53 functional categories and cell-specific analyses are presented in Supplementary data 11.

## Prediction ability of the polygenic risk score of the multimodal BAG

We aim to derive an individual-level biomarker (i.e., PRS) to quantify overall brain health's genetics susceptibility/liability. To this end, we derived the PRS for GM, WM, and FC-BAG using the conventional C + T (clumping plus $P$ value threshold) approach[56] via PLINK and a Bayesian method via PRS-CS[57] (Method 4G).

We found that the GM, WM, and FC-BAG-PRS derived from PRS-CS significantly predicted the phenotypic BAGs in the test data (split2 GWAS, 15,697 < $N$ < 15,940), with an incremental $R^2$ of 2.17%, 1.85%, and 0.19%, respectively (Fig. 4D). Compared to the PRS derived from PRS-CS, the PLINK approach achieved a lower incremental $R^2$ of 0.81%, 0.45%, and 0.14% for GM, WM, and FC-BAG, respectively (Supplementary Fig. 9). Overall, the predictive power of PRS is not high, in line with earlier discoveries involving raw imaging-derived phenotypes, as demonstrated in ref. 13. The authors developed PRSs for seven selective brain regions, which explained roughly 1.18–3.93% of the phenotypic variance associated with these traits.

## The potential causal relationships between GM and WM-BAG and other clinical traits

Our genetic correlation results motivated us to investigate the potential causal effects of several risk factors (i.e., exposure variable) on multimodal BAG (i.e., outcome variable) using a bidirectional two-sample MR approach[58] (Method 4H). We hypothesized that several diseases and lifestyle risk factors (Supplementary Table 4) might contribute to accelerating or decelerating human brain aging.

We found putative causal effects of triglyceride-to-lipid ratio in very large very-low-density lipoprotein (VLDL)[59] [$P$ value = $5.09 \times 10^{-3}$, OR (95% CI) = 1.08 (1.02, 1.13), number of SNPs = 52], type 2 diabetes[60] [$P$ value = $1.96 \times 10^{-2}$, OR (95% CI) = 1.05 (1.01, 1.09), number of SNPs = 10], and breast cancer[61] [$P$ value = $1.81 \times 10^{-2}$, OR (95% CI) = 0.96 (0.93, 0.99), number of SNPs = 118] on GM-BAG (i.e., accelerated brain age). We also identified causal effects of AD[62] [$P$ value = $7.18 \times 10^{-5}$, OR (95% CI) = 1.04 (1.02, 1.05), number of SNPs = 13] on WM-BAG (Fig. 5A). We subsequently examined the potential inverse causal effects of multimodal BAG (i.e., exposure) on these risk factors (i.e., outcome). However, owing to the restricted power [number of instrumental variables (IV) < 6], we did not observe any significant signals (Supplementary Fig. 10 and File 10).

## Sensitivity analyses for Mendelian randomization

As Mendelian randomization is sensitive to underlying IV assumptions (Method 4H), we performed sensitivity analyses to investigate potential violations. To illustrate this, we showcased the sensitivity analysis results for the causal effect of the triglyceride-to-lipid in VLDL ratio on GM-BAG (Fig. 5B–E). In a leave-one-out analysis, we found that no single SNP overwhelmingly drove the overall effect (Fig. 5B). There was evidence for the presence of minor heterogeneity[63] of the causal effect amongst SNPs (Cochran's Q value = 76.06, $P$ value = $5.09 \times 10^{-3}$). Some SNPs exerted opposite causal effects compared to the model using all SNPs (Fig. 5C). The scatter plot (Fig. 5D) indicated one obvious SNP outlier (rs11591147), and the funnel plot showed minor asymmetry with only an outlier denoted in Fig. 5E (rs4507142). Finally, the MR Egger estimator allows for pleiotropic effects independent of the effect on the exposure of interest (i.e., the InSIDE assumption[64]). Our results from the Egger estimator showed a small positive intercept ($5.21 \times 10^{-3} \pm 2.87 \times 10^{-3}$, $P$ value = 0.07) and a lower OR [inverse-variance weighted (IVW): 1.08 (1.02, 1.13) vs. Egger: 1.01 (0.93, 1.10)], which

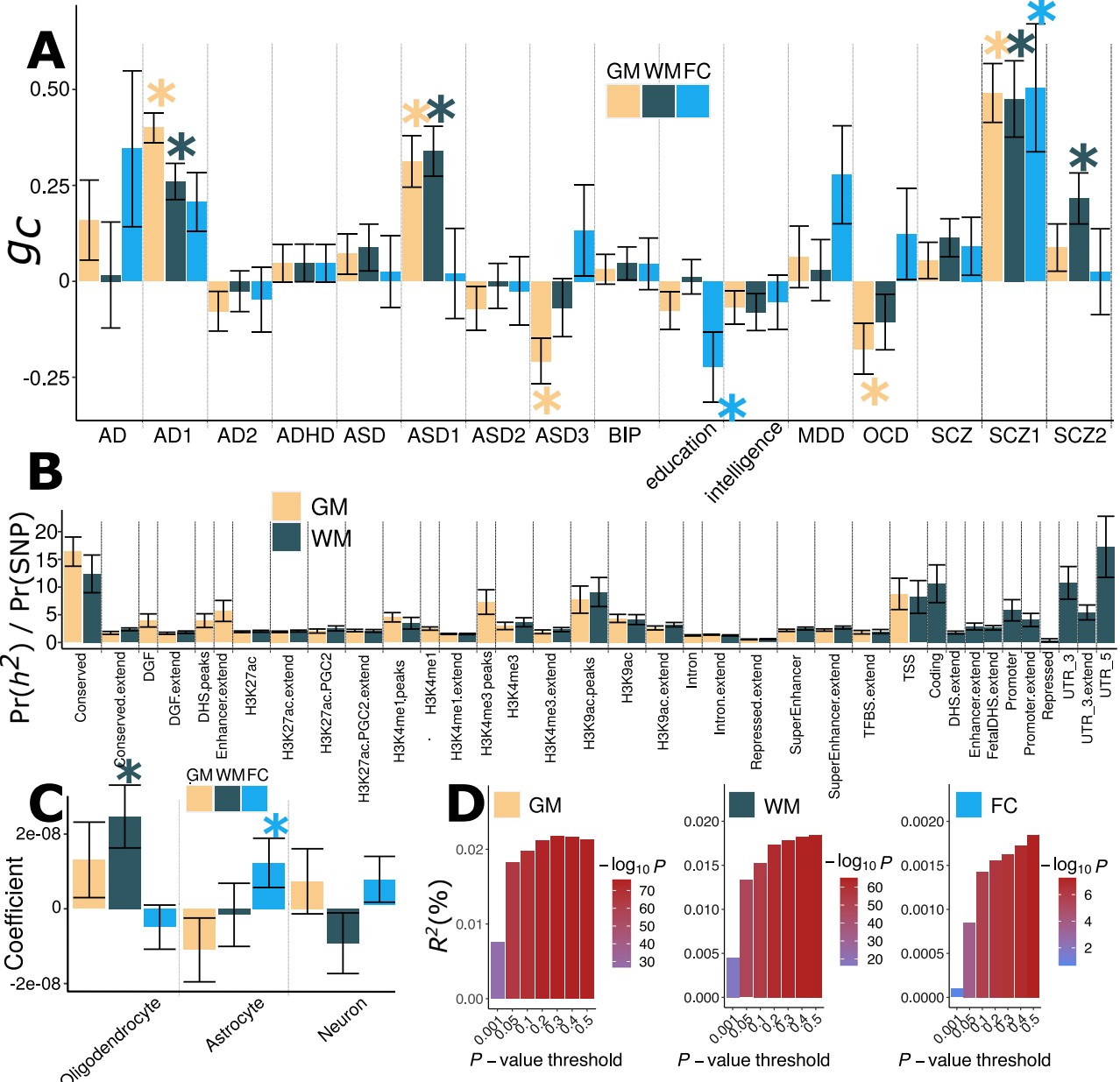

**Fig. 4 | Genetic correlation, partitioned heritability enrichment, and PRS prediction accuracy on multimodal brain age gaps. A** Genetic correlation ($g_c$) between GM, WM, and FC-BAG and 16 clinical traits. These traits include neurodegenerative diseases (e.g., AD) and their AI-derived subtypes (e.g., AD1 and AD2[4]), neuropsychiatric disorders (e.g., ASD) and their subtypes (ASD1, 2, and 3[46]), intelligence, and education. After adjusting for multiple comparisons using the FDR method, the * symbol denotes statistical significance (two-sided $P$ value < 0.05). Supplementary table 3 and data 10 presents the sample size and $P$ value. **B** The proportion of heritability enrichment for the 53 functional categories[51]. We only show the functional categories that survived the correction for multiple comparisons using the FDR method. **C** Cell type-specific partitioned heritability estimates. We included gene sets from Cahoy et al.[104] for three main cell types (i.e., astrocyte, neuron, and oligodendrocyte). After adjusting for multiple comparisons using the FDR method, the * symbol denotes statistical significance ($P$ value < 0.05). Detailed

results, including P-values, are presented in Supplementary data 11. LDSC resulted in an empirical covariance matrix of coefficient estimates and tested whether the per-SNP heritability is greater in the category/cell type than out of the category/cell type (i.e., one-sided). For Figure (**A**–**C**), data are presented as the mean value of the estimated parameters and error bars representing the standard error of the estimated parameters. **D** The incremental $R^2$ of the PRS derived by PRC-CS to predict the GM, WM, and FC-BAG in the target/test data (i.e., the split2 GWAS). The $y$-axis indicates the proportions of phenotypic variation (GM, WM, and FC-BAG) that the PRS can significantly and additionally explain. The $x$-axis lists the seven $P$ value thresholds considered. Abbreviation: AD Alzheimer's disease, ADHD attention-deficit/hyperactivity disorder, ASD autism spectrum disorder, BIP bipolar disorder, MDD major depressive disorder, OCD obsessive-compulsive disorder, SCZ schizophrenia.

may indicate the presence of directional horizontal pleiotropy for some SNPs. We present sensitivity analyses for other significant exposure variables in Supplementary Fig. 11.

To investigate the potential directional pleiotropic effects, we re-analyzed the MR Egger regression by excluding the two outliers

identified in Fig. 5D (rs11591147) and E (rs4507142), which led to a slightly increased OR [1.04 (0.96, 1.12)] and a smaller positive intercept $(4.41 \times 10^{-3} \pm 2.65 \times 10^{-3}, P$ value = 0.09). Our findings support that these two outlier SNPs may have a directional pleiotropic effect on GM-BAG. Nevertheless, given the complex nature of

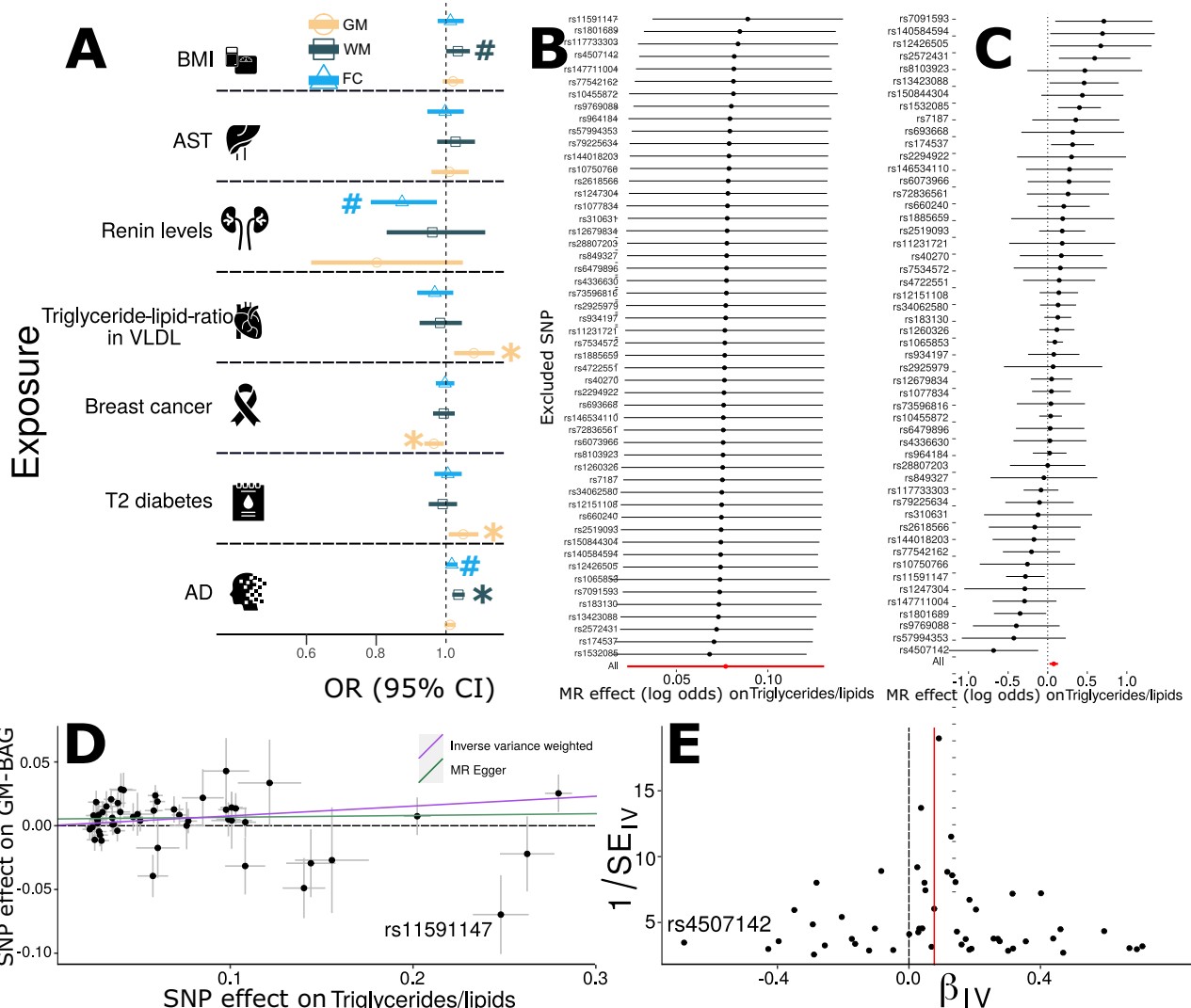

**Fig. 5 | Causal inference of multimodal brain age gaps.** Causal inference was performed using a two-sample Mendelian Randomization (MR, Method 4H) approach for seven selected exposure variables on three outcome variables (i.e., GM, WM, and FC-BAG). The symbol * denotes statistical significance (two-sided *P* value) after correcting for multiple comparisons using the FDR method (*N* = 7); the symbol # denotes the tests passing the nominal significance threshold (*P* value < 0.05) but not surviving the multiple comparisons. Shapes (circles, triangles, and rectangles) represent odds ratios (OR), and error bars show 95% confidence intervals (CI). **B**) Leave-one-out analysis of the triglyceride-to-lipid ratio on GM-BAG. Each dot represents the MR effect (log OR), and the error bar displays the 95% CI by excluding that SNP from the analysis. The red line depicts the IVW estimator using all SNPs. **C**) Forest plot for the single-SNP MR results. Each dot represents the MR effect (log OR)), and the error bar displays the 95% CI for the triglyceride-to-lipid ratio on GM-BAG using only one SNP; the red line shows the MR effect using all SNPs together. **D**) Scatter plot for the MR effect sizes of the SNP-triglyceride-to-

lipid ratio association (*x*-axis, SD units) and the SNP-GM-BAG associations (*y*-axis, log OR) with standard error bars. The slopes of the purple and green lines correspond to the causal effect sizes estimated by the IVW and the MR Egger estimator, respectively. We annotated a potential outlier. **E**) Funnel plot for the relationship between the causal effect of the triglyceride-to-lipid ratio on GM-BAG. Each dot represents MR effect sizes estimated using each SNP as a separate instrument against the inverse of the standard error of the causal estimate. The sample size for the 7 clinical traits is presented in Supplementary Table 4. The vertical red line shows the MR estimates using all SNPs. We annotated a potential outlier. Abbreviation: AD Alzheimer's disease, AST aspartate aminotransferase, BMI body mass index, VLDL very low-density lipoprotein, CI confidence interval, OR odds ratio, SD standard deviation, SE standard error. Interpreting these potential causal relationships should be cautiously undertaken despite our efforts to perform multiple sensitivity checks to evaluate the possible violations of underlying assumptions.

brain aging, many other biological pathways may also contribute to human brain aging. For instance, the SNP (rs11591147) was largely associated with other blood lipids, such as LDL cholesterol[65], and heart diseases, such as coronary artery disease[66]. Detailed results obtained from all five MR methods are presented in Supplementary data 12.

## Discussion

The present study harnessed brain imaging genetics from a cohort of 42,089 participants in UKBB to investigate the underlying genetics of multimodal BAG. Our approach commenced with objectively assessing

brain age prediction performance, encompassing various imaging modalities (T1-weighted, diffusion, and resting-state MRI), feature types (ROI vs. voxel), and machine learning algorithms. Subsequently, we conducted genome-wide associations, demonstrating the robustness of identified genetic signals in individuals of European ancestry across diverse factors. Lastly, our study encompassed several post-GWAS analyses, validating the GWAS results, shedding light on the intricate biological processes involved, and uncovering the multifaceted interplay between human brain aging and various health conditions and clinical traits. Our findings unveiled shared genetic factors and unique characteristics – varying degrees of phenotypic and

genetic correlation – within BAG across three distinct imaging modalities.

## Genetic architecture of GM-BAG

Our genetic results from GM-BAG substantiate that many diseases, conditions, and clinical phenotypes share genetic underpinnings with brain age, perhaps driven by macrostructural changes in GM (e.g., brain atrophy). The locus with the most significant signal (the top lead SNP rs534114641 at 17q21.31) showed substantial association with the traits mentioned above and was mapped to numerous genes associated with various diseases (Fig. 2C). Several previous GM-BAG GWAS[19,21] also identified this locus. Among these genes, the *MAPT* gene, known to encode a protein called tau, is a prominent AD hallmark and implicated in approximately 30 tauopathies, including progressive supranuclear palsy and frontotemporal lobar degeneration[67]. Our gene-drug-disease network also showed several drugs, such as Semorinemab[43], in active clinical trials currently targeting treatment for AD (Fig. 3). The heritability enrichment of GM-BAG was high in several functional categories, with conserved regions being the most prominent. The observed higher heritability enrichment in conserved regions compared to coding regions[68] supports the long-standing hypothesis regarding the functional significance of conserved sequences. However, the precise role of many highly conserved non-coding DNA sequences remains unclear[69]. The genetic correlation results of GM-BAG with subtypes of common brain diseases highlight the promise for the AI-derived subtypes, rather than the "one-for-all" unitary disease diagnosis, as robust endophenotypes[22]. These findings strongly support the clinical implications of re-evaluating pertinent hypotheses using the AI-derived subtypes in patient stratification and personalized medicine.

The elevated triglyceride-to-lipid ratio in VLDL, an established biomarker for cardiovascular diseases[70], is causally associated with higher GM-BAG (accelerated brain age). Therefore, lifestyle interventions that target this biomarker might hold promise as an effective strategy to enhance overall brain health. In addition, we revealed that one unit-increased likelihood of type 2 diabetes has a causal effect on GM-BAG increase. Research has shown that normal brain aging is accelerated by approximately 26% in patients with progressive type 2 diabetes compared with healthy controls[71]. The protective causal effect of breast cancer on GM-BAG is intriguing in light of existing literature adversely linking breast cancer to brain metastasis[72] and chemotherapy-induced cognitive impairments, commonly known as "chemo brain". In addition, it's important to exercise caution when considering the potential causal link between breast cancer and GM-BAG, as MR analyses are susceptible to population selection bias[73] due to the high breast cancer mortality rate.

## Genetic architecture of WM-BAG

The genetic architecture of WM-BAG exhibits strong correlations with cancer-related traits, AD, and physical measures such as BMI, among others. Our phenome-wide association query largely confirms the enrichment of these traits in previous literature. In particular, the *DNAJC1* gene, annotated from the most polygenic locus on chromosome 10 (top lead SNP: rs564819152), encodes a protein called heat shock protein 40 (Hsp40) and plays a role in protein folding and the response to cellular stress. This gene is implicated in various cancer types, such as breast, renal, and melanoma (Supplementary Fig. 12). In addition, several FDA-approved drugs have been developed based on these WM-BAG genes for different types of cancer in our gene-drug-disease network (Fig. 3). Our findings provide insights into the genetic underpinnings of WM-BAG and their potential relevance to cancer.

Remarkably, one unit-increased likelihood of AD was causally associated with increased WM-BAG. Our Mendelian randomization analysis confirmed the abundant association evidenced by the phenome-wide association query (Fig. 2B). Dementia, such as AD, is undeniably a significant factor contributing to the decline of the aging brain. Evidence suggests that AD is not solely a GM disease; significant microstructural changes can be observed in WM before the onset of cognitive decline[74]. We also identified a nominal causal significance of BMI [risk effect; P-value = $4.73 \times 10^{-2}$, OR (95% CI) = 1.03 (1.00, 1.07)] on WM-BAG. These findings underscore the potential of lifestyle interventions and medications currently being tested in clinical trials for AD to improve overall brain health.

## Genetic architecture of FC-BAG

The genetic signals for FC-BAG were weaker than those observed for GM and WM-BAG, which is consistent with the age prediction performance and partially corroborates Cheverud's conjecture: using genetic correlations (Fig. 2F) as proxies for phenotypic correlations (Fig. 1E) when collecting individual phenotypes is expensive and unavailable. A genomic locus on chromosome 6 (6q.13) harbors an independent variant (rs1204329) previously linked to insomnia[75]. The top lead SNP, rs5877290, associated with this locus is a deletion-insertion mutation type: no known association with any human disease or gene mapping has been established for this SNP. The genetic basis of FC-BAG covaries with educational performance and schizophrenia subtypes. Specifically, parental education has been linked to cognitive ability, and researchers have identified a functional connectivity biomarker between the right rostral prefrontal cortex and occipital cortex that mediates the transmission of maternal education to offspring's performance IQ[76]. On the other hand, schizophrenia is a highly heritable mental disorder that exhibits functional dysconnectivity throughout the brain[77]. AD was causally associated with FC-BAG with nominal significance [risk effect for per unit increase; P value = $4.43 \times 10^{-2}$, OR (95% CI) = 1.02 (1.00, 1.03), number of SNPs = 13] (Fig. 5A). The relationship between functional brain networks and the characteristic distribution of amyloid-β and tau in AD[78] provides evidence that AD is a significant factor in the aging brain, underscoring its role as a primary causative agent.

The comparative trend of genetic heritability among GM, WM, and FC-BAG is also consistent with previous large-scale GWAS of multimodal brain IDP. Zhao et al. performed GWAS on GM[13], WM[14], and FC-IDP[17], showing that FC-IDP is less genetically heritable than others. Similar observations were also demonstrated by ref. 11 in the large-scale GWAS using multimodal IDP from UKBB. The weaker genetic signal observed in FC-BAG can be attributed to many factors. One of the main reasons is the lower signal-to-noise ratio in FC measurements due to the dynamic and complex nature of brain activity, which can make it difficult to accurately measure and distinguish between the true signal and noise. Social-environmental and lifestyle factors can also contribute to the "missing heritability" observed in FC-BAG. For example, stress, sleep patterns, physical activity, and other environmental factors can impact brain function and connectivity[79]. In contrast, GM and WM measurements are more stable and less influenced by environmental factors, which may explain why they exhibit stronger genetic signals and higher heritability estimates.

## Limitations

This study has several limitations. We can employ deep learning on voxel-wise imaging scans to enhance brain age prediction performance. Nevertheless, it warrants additional exploration to determine whether the resulting reduction in MAE translates into more robust genome-wide associations, as our previous work has demonstrated that BAGs derived from a CNN with a lower MAE did not exhibit heightened sensitivity to disease effects such as AD[34]. Second, the generalization ability of the GWAS findings to non-European ancestry is limited, potentially due to small sample sizes and cryptic population stratification. Future investigations can be expanded to encompass a broader spectrum of underrepresented ethnic groups, diverse disease populations, and various age ranges spanning the entire lifespan. This

expansion can be facilitated by leveraging the resources of large-scale brain imaging genetic consortia like ADNI[40], focused on Alzheimer's disease, and ABCD[80], which centers on brain development during adolescence. Third, it's important to exercise caution when interpreting the results of this study due to the various assumptions associated with the statistical methods employed, including LDSC and MR. Lastly, it's worth noting that brain age represents a residual score encompassing measurement error. A recent study[81] has underscored the significance of incorporating longitudinal data when calculating brain age. Future research should be conducted once the longitudinal scans from the UK Biobank become accessible to explore this impact on GWASs.

## Outlook

In summary, our multimodal BAG GWASs provide evidence that the aging process of the human brain is a complex biological phenomenon intertwined with several organ systems and chronic diseases. We digitized the human brain from multimodal imaging and captured a complete genetic landscape of human brain aging. This opens research avenues for drug repurposing/repositioning and aids in identifying modifiable protective and risk factors that can ameliorate human brain health.

## Methods

### Method 1: Study populations

UKBB is a population-based study of more than 50,000 people recruited between 2006 and 2010 from Great Britain. The current study focused on participants from the imaging-genomics population who underwent both an MRI scan and genome sequencing (genotype array data and the imputed genotype data) under application number 35148. The UKBB study has ethical approval, and the ethics committee is detailed here: https://www.ukbiobank.ac.uk/learn-more-about-uk-biobank/governance/ethics-advisory-committee. The study design, phenotype and genetic data availability, and quality check have been published and detailed elsewhere[23]. Supplementary table 5 shows the study characteristics of the present work.

To train the machine learning model and compare the performance of the multimodal BAG, we defined the following two datasets:

- *Training/validation/test dataset*: To objectively compare the age prediction performance of different MRI modalities and machine learning models, we randomly sub-sampled 500 (250 females) participants within each decade's range from 44 to 84 years old, resulting in the same 4000 participants for GM, WM, and FC-IDP. This dataset was used to train machine learning models. In addition, we ensured that the training/validation/test splits were the same in the CV procedure. As UKBB is a general population, we explicitly excluded participants with common brain diseases, including mental and behavioral disorders (ICD-10 code: F; $N = 2678$; Data-Field = 41270) and diseases linked to the central nervous system (ICD-10 code: G group; $N = 3336$).
- *Independent test dataset*: The rest of the population for each MRI modality ($N = 38089$) was set as independent test datasets—unseen until we finalized the training procedure[82].

The GM-IDP includes 119 GM regional volumes from the MUSE atlas, consolidated by the iSTAGING consortium. We studied the influence of different WM-IDP features: *i)* 48 FA values; *ii)* 109 TBSS-based[83] values from FA, MD, ODI, and NDI; *iii)* 192 skeleton-based mean values from FA, MD, ODI, and NDI. For FC-IDP, 210 ICA-derived functional connectivity components were included. The WM and FC-IDP were downloaded from UKBB (Method 2B and C).

### Method 2: Image processing

**(A):** T1-weighted MRI processing: The imaging quality check is detailed in Supplementary Method 1. All images were first corrected for magnetic field intensity inhomogeneity.[84] A deep learning-based skull stripping algorithm was applied to remove extra-cranial material. In total, 145 IDPs were generated in gray matter (GM, 119 ROIs), white matter (WM, 20 ROIs), and ventricles (6 ROIs) using a multi-atlas label fusion method[85]. The 119 GM ROIs were fit to the four machine learning models to derive the GM-BAG.

**(B):** Diffusion MRI processing: UKBB has processed diffusion MRI (dMRI) data and released several WM tract-based metrics for the Diffusion Tensor Imaging (DTI) model (single-shell dMRI) and Neurite Orientation Dispersion and Density Imaging (NODDI[86]) model (multi-shell dMRI). The Eddy[87] tool corrected raw images for eddy currents, head motion, and outlier slices. The mean values of FA, MD, ODI, and NDI were extracted from the 48 WM tracts of the "ICBM-DTI-81 white-matter labels" atlas[88], resulting in 192 WM-IDP (category code:134). In addition, a tract-skeleton (TBSS)[83] and probabilistic tractography analysis[89] were employed to derive weighted-mean measures within the 27 major WM tracts, referred to as the 108 TBSS WM-IDP (category code: 135). Finally, since we observed overfitting—an increase of MAEs from the cross-validated test results to the independent test results—when incorporating features from FA, MD, ODI, and NDI (as detailed in Supplementary Table 1A), we chose to use only the 48 FA WM-IDPs to train the models for generating GM-BAG.

**(C):** Resting-state functional MRI processing: For FC-IDPs, we used the $21 \times 21$ resting-state functional connectivity (full correlation) matrices (data-field code: 25750) from UKBB[90,91]. UKBB processed rsfMRI data and released 25 whole-brain spatial independent component analysis (ICA)-derived components[92]; four components were removed due to artifactual components. This resulted in 210 FC-IDP quantifying pairwise correlations of the ICA-derived components. Details of dMRI and rsfMRI processing are documented here: https://biobank.ctsu.ox.ac.uk/crystal/crystal/docs/brain_mri.pdf.

### Method 3: Multimodal brain age prediction using machine learning models

GM, WM, and FC-IDP were fit into four machine learning models (linear and non-linear) to predict brain age as the outcome. Specifically, we used SVR, LASSO regression, MLP, and a five-layer neural network (NN: three linear layers and one rectified linear unit layer; Supplementary Fig. 13).

To objectively and reproducibly compare the age prediction performance using different machine learning models and MRI modalities, we adopted a nested CV procedure and included an independent test dataset[26]. In detail, the outer loop CV was conducted with 100 repeated random splits: 80% of the data served for training and validation, while the remaining 20% was allocated for testing. In the inner loop, if applicable, a 10-fold CV was performed for a grid search for hyperparameter tuning of the machine learning models. In addition, we concealed an independent test dataset—unseen for testing until we finished fine-tuning the machine learning models[82] (e.g., hyperparameters for SVR). To compare the results of different models and modalities, we showed MAE's mean and empirical standard deviation instead of performing any statistical test (e.g., a two-sample t-test). This is because no unbiased variance estimate exists for complex CV procedures (refer to notes from ref. 93).

### Method 4: Genetic analyses

Imputed genotype data were quality-checked for downstream analyses. Our quality check pipeline (see below) resulted in 33,541 European ancestry participants and 8,469,833 SNPs. After merging with the multimodal MRI populations, we included 31,557 European participants for GM-BAG, 31,749 participants for WM-BAG, and 32,017 participants for FC-BAG GWAS. Details of the genetic protocol[94] are described elsewhere[95,96]. We summarize our genetic QC pipeline as below. First, we excluded related individuals (up to 2nd-degree) from the complete UKBB sample using the KING software for family

relationship inference[97]. We then removed duplicated variants from all 22 autosomal chromosomes. Individuals whose genetically identified sex did not match their self-acknowledged sex were removed. Other excluding criteria were: *i*) individuals with more than 3% of missing genotypes; ii) variants with minor allele frequency of less than 1%; iii) variants with larger than 3% missing genotyping rate; iv) variants that failed the Hardy-Weinberg test at $1 \times 10^{-10}$. To adjust for population stratification[98], we derived the first 40 genetic principle components (PC) using the FlashPCA software[99].

**(A): Genome-wide association analysis:** For GWAS, we ran a linear regression using Plink[100] for GM, WM, and FC-BAG, controlling for confounders of age, dataset status (training/validation/test or independent test dataset), age × squared, sex, age × sex interaction, age-squared × sex interaction, total intracranial volume, the brain position in the scanner (lateral, transverse, and longitudinal), and the first 40 genetic principal components. The inclusion of these covariates is guided by pioneer neuroimaging GWAS conducted by refs. 11, 13. We adopted the genome-wide P-value threshold ($5 \times 10^{-8}$) and annotated independent genetic signals considering linkage disequilibrium (see below). We then estimated the SNP-based heritability using GCTA[35] using the individual-level genotype data with the same covariates in GWAS.

To check the robustness of our GWAS results using European ancestry, we performed seven sensitivity checks, including i) split-sample GWAS by randomly dividing the entire population into two sex and age-matched splits, ii) sex-stratified GWAS for males and females, iii) non-European GWAS, iv) fastGWA[39] for a mixed linear model that accounts for cryptic population stratification, v) machine learning-specific GWAS, vi) feature type-specific GWAS, and vii) independent GWAS using whole-genome sequencing (WGS) from ADNI (the quality check steps are detailed elsewhere[4] and also in the caption of Supplementary Fig. 7).

**(B):** Phenome-wide association query for genomic loci associated with other traits in the literature: We queried the candidate SNPs within each locus in the GWAS Catalog (query date: January 10th, 2023 via FUMA version: v1.5.0) to determine their previously identified associations with other traits. For these associated traits, we further mapped them into several high-level categories for visualization purposes (Fig. 2B).

**(C):** Annotation of genomic loci and genes: The annotation of genomic loci and mapped genes was performed via FUMA[101] (https://fuma.ctglab.nl/, version: v1.5.0). For the annotation of genomic loci, we first defined lead SNPs (correlation $r^2 \le 0.1$, distance <250 kilobases) and assigned them to a genomic locus (non-overlapping); the lead SNP with the lowest *P* value (i.e., the top lead SNP) was used to represent the genomic locus. For gene mappings, three different strategies were considered. First, positional mapping assigns the SNP to its physically nearby genes (a 10 kb window by default). Second, eQTL mapping annotates SNPs to genes based on eQTL associations. Finally, chromatin interaction mapping annotates SNPs to genes when there is a significant chromatin interaction between the disease-associated regions and nearby or distant genes.[101] The definition of top lead SNP, lead SNP, independent significant SNP, and candidate SNP can be found in Supplementary Method 2.

**(D):** Gene-drug-disease network construction: We curated data from the Drug Bank database (v.5.1.9)[102] and the Therapeutic Target Database (updated by September 29th, 2021) to construct a gene-drug-disease network. Specifically, we constrained the target to human organisms and included all drugs with active statuses (e.g., patented and approved) but excluded inactive ones (e.g., terminated or discontinued at any phase). To represent the disease, we mapped the identified drugs to the Anatomical Therapeutic Chemical (ATC) classification system for the Drugbank database and the International Classification of Diseases (ICD-11) for the Therapeutic Target Database.

**(E):** Genetic correlation: We used LDSC[36] to estimate the pairwise genetic correlation ($r_g$) between GM, WM, and FC-BAG and several pre-selected traits (Supplementary Table 3) by using the precomputed LD scores from the 1000 Genomes of European ancestry. The following pre-selected traits were included: Alzheimer's disease (AD), autism spectrum disorder (ASD), attention-deficit/hyperactivity disorder (ADHD), OCD, major depressive disorder (MDD), bipolar disorder (BIP), schizophrenia (SCZ), education and intelligence, as well as the AI-derived subtypes for AD (AD1 and AD2[4]), ASD (ASD1, ASD2, and ASD3[46]), and SCZ (SCZ1 and SCZ2[47]). To ensure the suitability of the GWAS summary statistics, we first checked that the selected study's population was of European ancestry. We then guaranteed a moderate SNP-based heritability $h^2$ estimate and excluded the studies with spurious low $h^2$ (<0.05). Notably, LDSC corrects for sample overlap and provides an unbiased estimate of genetic correlation[103]. The $h^2$ estimate from LDSC is generally lower than that of GCTA because LDSC uses GWAS summary statistics and pre-computed LD information and has slightly different model assumptions across different software.

**(F):** Partitioned heritability estimate: Partitioned heritability analysis estimates the percentage of heritability enrichment explained by annotated genome regions[51]. First, the partitioned heritability was calculated for 53 main functional categories. The 53 functional categories are not specific to any cell type, including coding, UTR, promoter, and intronic regions. Details of the 53 categories are described elsewhere[51] and are also presented in Supplementary data 11A. Subsequently, cell type-specific partitioned heritability was estimated using gene sets from ref. 104 for three main cell types (i.e., astrocyte, neuron, and oligodendrocyte) (Supplementary data 11B).

**(G):** PRS prediction: We calculated the PRS using the GWAS results from the split-sample analyses. The weights of the PRS were defined based on split1 data (training/base data), and the split2 GWAS summary statistics were used as the test/target data. The QC steps for the base data are as follows: *i*) removal of duplicated and ambiguous SNPs for the base data; *ii*) clumping the base GWAS data; *iii*) pruning to remove highly correlated SNPs in the target data; *iv*) removal of high heterozygosity samples in the target data; *v*) removal of duplicated, mismatching and ambiguous SNPs in the target data. After rigorous QC, we employed two methods to derive the three BAG-PRS in the split2 population: *i*) PLINK with the classic C + T method (clumping + thresholding) and *ii*) PRS-CS[57] with a Bayesian approach.

To determine the "best-fit" PRS P-value threshold, we performed a linear regression using the PRS calculated at different *P* value thresholds (0.001, 0.05, 0.1, 0.2, 0.3, 0.4, 0.5), controlling for age, sex, total intracellular volume, brain position during scanning (lateral, transverse, and longitudinal), and the first forty genetic PCs. A null model was established by including only the abovementioned covariates. The alternative model was then constructed by introducing each BAG-PRS as an extra independent variable.

**(H):** Two-sample Mendelian Randomization: We investigated whether the clinical traits previously associated with our genomic loci (Fig. 2B) were a cause or a consequence of GM, WM, and FC-BAG using a bidirectional, two-sample MR approach. GM, WM, and FC-BAG are the outcome/exposure variables in the forward/inverse MR, respectively. We applied five different MR methods using the TwoSampleMR R package[58], including the inverse variance weighted (IVW), MR Egger[105], weighted median[106], simple mode, and weighted mode methods. We reported the results of IVW in the main text and the four others in Supplementary data 10. MR relies on a set of crucial assumptions to ensure the validity of its results. These assumptions include the requirement that the chosen genetic instrument exhibits a strong association with the exposure of interest while remaining free from direct associations with confounding factors that could influence the outcome. Additionally, the genetic variant used in MR should be independently allocated during conception and inheritance, guaranteeing its autonomy from potential confounders. Furthermore, this

genetic instrument must affect the outcome solely through the exposure of interest without directly impacting alternative pathways that could influence the outcome (no horizontal pleiotropy). The five MR methods handle pleiotropy and instrument validity assumptions differently, offering various degrees of robustness to violations. For example, MR Egger provides a method to estimate and correct for pleiotropy, making it robust in the presence of horizontal pleiotropy. However, it assumes that directional pleiotropy is the only form of pleiotropy present.

To ensure an unbiased selection of exposure variables, we followed a systematic procedure guided by the STROBE-MR Statement[107]. We pre-selected exposure variables across various categories based on our phenome-wide association query. These variables encompassed neurodegenerative diseases (e.g., AD), liver biomarkers (e.g., AST), cardiovascular diseases (e.g., the triglyceride-to-lipid ratio in VLDL), and lifestyle-related risk factors (e.g., BMI). Subsequently, we conducted an automated query for these traits in the IEU GWAS database[108], which provides curated GWAS summary statistics suitable for MR, using the *available_outcomes()* function. We ensured the selected studies used European ancestry populations and shared the same genome build as our GWAS (HG19/GRCh37). Additionally, we manually examined the selected studies to exclude any GWAS summary statistics overlapping with UK Biobank populations to prevent bias stemming from sample overlap[109]. This process yielded a set of seven exposure variables, comprising AD, breast cancer, type 2 diabetes, renin level, triglyceride-to-lipid ratio, aspartate aminotransferase (AST), and BMI. The details of the selected studies for the instrumental variables (IVs) are provided in Supplementary Table 4.

We performed several sensitivity analyses. First, a heterogeneity test was performed to check for violating the IV assumptions. Horizontal pleiotropy was estimated to navigate the violation of the IV's exclusivity assumption[63] using a funnel plot, single-SNP MR approaches, and MR Egger estimator[105]. Moreover, the leave-one-out analysis excluded one instrument (SNP) at a time and assessed the sensitivity of the results to individual SNP.

### Reporting summary

Further information on research design is available in the Nature Portfolio Reporting Summary linked to this article.

## Data availability

This study used the UK Biobank resource under Application Number 35148. No software was used for data collection. The GWAS summary statistics generated from our analyses are publicly available at the MEDICINE portal: https://labs.loni.usc.edu/medicine/organ_systems/brain. The raw imaging data are restricted to registered researchers and are protected and unavailable due to data privacy laws; access can be obtained at https://www.ukbiobank.ac.uk/. The gene-drug-disease network used data from the Drug Bank database (v.5.1.9: https://go.drugbank.com/) and the Therapeutic Target Database (updated by September 29th, 2021: https://idrblab.net/ttd/). Our genetic analyses also used GWAS summary statistics from the IEU OpenGWAS database (https://gwas.mrcieu.ac.uk/) (Supplementary Table 3) and GWAS Catalog (https://www.ebi.ac.uk/gwas/) (Supplementary Table 4).

## Code availability

The software and resources used in this study are all publicly available: MLNI: https://anbai106.github.io/mlni/, brain age prediction (v0.1.2), MEDICINE: https://labs.loni.usc.edu/medicine, knowledge portal for dissemination and GWAS summary statistics sharing, MUSE: https://www.med.upenn.edu/cbica/sbia/muse.html, image preprocessing for GM-IDP (v0.0.1), PLINK: https://www.cog-genomics.org/plink/, GWAS and PRS (plink 2.0), FUMA: https://fuma.ctglab.nl/, gene mapping, genomic locus annotation (v1.5.0), GCTA: https://yanglab.westlake.edu.cn/software/gcta/#Overview, heritability estimates, and fastGWA

(v1.94.1), LDSC: https://github.com/bulik/ldsc, genetic correlation, partitioned heritability, and heritability estimates (git version: aa33296), TwoSampleMR: https://mrcieu.github.io/TwoSampleMR/index.html, MR (v0.5.6), PRS-CS: https://github.com/getian107/PRScs, PRS (Aug 10, 2023).

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

## Acknowledgements

We want to express our sincere gratitude to the UK Biobank team for their invaluable contribution to advancing clinical research in our field. The primary funding support for this present study is from the initial funding package as an Assistant Professor for WJ, provided by Stevens Neuroimaging and Informatics Institute, Keck School of Medicine of USC, University of Southern California. The iSTAGING consortium is a multi-institutional effort funded by NIA by RF1 AG054409 for DC. This

research has been conducted using the UK Biobank Resource under Application Number 35148. We thank Caroline O'Driscoll, Robert De La Cruz, and Ragini Jain for developing the MEDICINE web portal. Finally, we thank Dr. Yunpeng Wang for generously providing us with their GWAS summary statistics[20] during the revision.

## Author contributions

J.W. has full access to all the data in the study and takes responsibility for the integrity of the data and the accuracy of the data analysis. Study concept and design: J.W. Acquisition, analysis, or interpretation of data: J.W. Drafting of the manuscript: J.W. Critical revision of the manuscript for important intellectual content: J.W., B.Z., Z.Y., G.E., I.S., E.M., Y.C., G.H., J.B., A.B.P., Z.Z., Y.V., M.R., H.S., P.T., L.S., A.T., C.D. Statistical analysis: J.W.

## Competing interests

The authors declare no competing interests.
