## [Peer Review File · Nature Communications]

REVIEWER COMMENTS

Reviewer #1 (Remarks to the Author):

Wen et al. used multimodal magnetic resonance imaging and artificial intelligence to examine the genetic heterogeneity of the brain age gap (BAG) derived from gray matter volume (GM-BAG), white matter microstructure (WM-BAG), and functional connectivity (FC-BAG). The authors identified sixteen significant genomic loci. A gene-drug-disease network highlighted genes linked to GM-BAG for treating neurodegenerative and neuropsychiatric disorders and WM-BAG genes for cancer therapy. GM-BAG showed the highest heritability enrichment for genetic variants in conserved regions, whereas WM-BAG exhibited the highest heritability enrichment in the 5' untranslated regions, oligodendrocytes and astrocytes but not neurons. Notably, Mendelian randomization identified causal risk effects of triglyceride-to-lipid ratio in very low-density lipoprotein and type 2 diabetes on GM-BAG and AD on WM-BAG.

While the authors claimed that these results provide valuable insights into the genetic heterogeneity of human brain aging, with clinical implications for potential lifestyle and therapeutic interventions, I found the overall flow of their manuscript confusing and the key word "heterogeneity" not being adequately reflected. I have a few comments:

The overall connection and logics of this manuscript has a room for improvement. In the first paragraph of results, the key words should be (in my understanding) "heterogeneity" and "BAG", yet neither of them appeared in a timely manner. A huge paragraph was used to describe model prediction, model performance, models, sex stratification. BAG was defined as the difference between an individual's predictive brain age and his/her chronological age. Yet this difference was not introduced or described early enough (or strong enough) in this section of results (my understanding is that IDP does not equal to BAG, prediction neither, but rather the difference). At the very end, in the fifth point, finally comes the heterogeneity of BAG. This is very confusing and weakened the necessity of "leading us to investigate the underlying genetic determinants responsible for this phenotypic heterogeneity". I do not see the point, these one-sentence described results are not strong enough to be further pursued using a genetic approach.

The main problem with GWAS (the second and the third part of results) is a lack of real-meaning validation. Sensitivity analyses in split-sample experiments, sex stratified experiments, and non-Caucasian populations do not really account for an independent validation. Especially the non-Caucasian analysis, number one it's under-powered, number two it's way too early to generalize to another ethnicity. I would suggest tell a full story in the Caucasians as a first step. Another issue is that I do not really see a heterogeneity. Both GM-BAG and WM-BAG are associated with brain related issues, neurology or psychiatry. I cannot agree with what's been stated in their abstract, "with GM-BAG loci showing abundant associations with neurodegenerative and neuropsychiatric traits, WM-BAG loci implicated in cancer and Alzheimer's disease (AD), and FC-BAG in insomnia", if this reflects heterogeneity, it also reflects homogeneity.

In the "Multimodal BAG is genetically correlated with AI-derived subtypes of brain disorders" section, justify the reason why original subtype is not genetically associated with multimodal BAG? These phenotypes are true phenotypes (instead of imputed or calculated or derived), why these phenotypes did not stand out in the first place?

In "Polygenic risk scores of other diseases weakly predict multimodal BAG", how are the 36 diseases selected? While the authors just claimed in a few paragraphs ahead, that "These results indicate potential genetic overlap and causal links between the multimodal BAG and other clinical traits."

In the last paragraph of results, the title said, "Triglyceride, type 2 diabetes, breast cancer, and AD are causally associated with GM and WM-BAG", while the body said, "Based on our sensitivity analysis and the consistency of the results obtained from all five MR methods (Supplementary eFile 3), we showed evidence for a putative causal effect of the triglyceride-to-lipid ratio in VLDL on GM-BAG", and the abstract said "Mendelian randomization identified causal risk effects of triglyceride-to-lipid ratio in very low-density lipoprotein and type 2 diabetes on GM-BAG and AD on WM-BAG", which version is accurate? These three versions each might be correct to some extent, but the authors have the obligations to guide through your readers and be consistent throughout.

A few minor comments, several abbreviations appeared without any explanation, for example, MAE, FA. I don't even know that FA, MD, NDI and ODI are four diffusion metrics until I read a few lines further, and wonder how many of them still there? The "independent dataset" was never introduced – please note that Nature paper exhibits results before methods, so the authors are responsible for a properly organization of their texts. Not necessarily lengthy, but enough information should be revealed to support for a barrier-free reading).

"The genomic loci linked to GM, WM, and FC-BAG were distributed throughout the human genome" – not accurate. These sentences are best suit for a height or BMI GWAS, there we say signals distributed throughout human genome.

Reviewer #2 (Remarks to the Author):

The study "The Genetic Heterogeneity of Multimodal Human Brain Age" by Wen et al. modelled the brain age gap (BAG) using different brain imaging modalities (including grey matter volume, GM; white matter microstructure, WM; and functional connectivity, FC) and investigated their associated genetics. The study aims are very broad: The authors performed GWAS to identify genetic variants separately associated with BAGs derived from GM, WM and FC, calculated genetic correlations to estimate the degree of genetic overlap of BAG phenotypes with other trait and diseases, and constructed gene-drug-disease networks to identify druggable targets. They also performed Mendelian Randomisation to identify potential causal effects of risk factors (such as neurodegenerative diseases) on BAG. Finally, they calculated associations between polygenic risk for multiple clinical traits to predict BAG.

Many aspects of this study provide an important contribution to the field of neuroimaging genetics, including the systematic use of multiple imaging modalities, state-of-the-art statistical approaches, and the use of the largest available database that would permit investigating this question. The subject of multimodal brain age and its genetic underpinnings is worthwhile and timely, and it represents an impactful finding to show reliably that different imaging modalities yield distinct brain age phenotypes with distinct biological underpinnings, that may represent druggable targets.

I think, however, that this study requires some important additions to match the quality of research that tends to get published in Nature Communications. Those additions include, for example, a more nuanced discussion of the quality of their BAG phenotypes and GWAS which is fundamentally required to judge the quality and reliability of post-GWAS investigations. Furthermore, I suggest that the methods section requires more detail to make results reproducible, and that the manuscript requires a thorough discussion of methods limitations. I have outlined more specific suggestions below.

Analyses supporting the quality of the BAG phenotypes and GWAS

- Phenotypic correlations: Considering it seems that the heterogeneity of different BAGs is a major focus of this study, the authors do not seem to present empirical evidence that there is indeed heterogeneity between the BAG phenotypes. In one sentence ("Fifth, we found considerable heterogeneity in the expression of GM, WM, and FC-BAG in this independent dataset"), they point the reader to Fig.1E which is challenging to interpret – in a 3-dimensional plot on a 2-dimensional papers it is near impossible to identify the true location of a dot – and they do not quantify the extent to which BAG phenotypes correlate. Please consider reporting Pearson's correlations among the three BAG phenotypes used for subsequent GWAS investigations. If there truly is heterogeneity between BAGs derived from different imaging modalities, the correlations among BAGs should be moderate at best. This would represent an important part of the study to show genetic analyses are indeed investigating distinct phenotypes, which is especially important because BAGs are residual scores that capture measurement error in addition to true individual differences. Would the authors also be able to more clearly visualise the heterogeneity (Fig.1E) and avoid a three-dimensional plot?

- Genetic correlations: Similarly, the different BAGs should only be moderately genetically correlated if they are truly distinct phenotypes. I suggest that reporting genetic correlations among BAGs is crucial

to be able to interpret individual GWAS hits (as done in "Genomic loci associated with multimodal BAG show different phenome-wide associations"). Only if different BAGs are imperfectly genetically correlated, we can be confident that genetic follow-up analyses are truly indicative of distinct biological pathways underlying the heterogeneity of different BAGs.

- PRS prediction: It is unclear to me why the authors chose to model genetic propensity (i.e., PRS) towards 16 other traits, instead of modelling genetic propensity towards BAGs from the GWAS they created as part of this manuscript. In my experience, studies that calculate GWAS summary statistics use PRS predictions to showcase the predictive ability (and thereby quality and validity) of their GWAS summary statistics. If a PRS derived from GWAS predicts its own base phenotype well (usually indicated by R^2), we take it as evidence that the GWAS indeed captures genetic propensity towards this trait. Would the authors consider reporting R^2 estimates of BAG PRSs predicting their base BAG as outcome? Could they also please explain why they chose to predict BAG phenotypes using PRSs of other traits, considering this study seem to focus on the genetic architecture of BAGs?

- Heritability: Would the authors consider stating heritability estimates in the main manuscript (page 6)? Heritability estimates are important indicators of signal-to-noise captured by the GWAS (and thereby another indication of quality), and the reader should know that they are substantial here (Fig.2F).

Suggestion to include more general genome-wide trends in the discussion of the manuscript (rather than mostly focusing individual GWAS hits only)

- The manuscripts invests a lot of space into discussing the specific SNP associations yielded by different BAGs, and the authors end the section ("Genomic loci associated with multimodal BAG show different phenome-wide associations") by stating those associations are indicative of genetic overlap with other traits. Whether there is genetic overlap can indeed be directly tested using genetic correlations, and I would argue that the highly polygenic nature of BAGs does not permit interpreting distinct individual GWAS hits indicative of genetic overlap. I would like to suggest the authors focus their discussion less on individual GWAS hits because significant hits are directly dependent upon GWAS sample size, and their significance is sensitive towards sampling variance. In the grand scheme of GWAS, the UKB sample size used here is relatively small and it is not reasonable to expect that this analysis would pick up on all relevant associations (and the chances of spurious associations are high – winners curse). Instead, I think it would greatly improve the manuscript to give more space to the discussion of genome-wide trends and BAGs genetic overlap among one another and with other traits. Such genome-wide trends tend to be more reliable than individual SNP associations. Could the authors please consider a more measured discussion of genome-wide trends alongside individual GWAS hits?

- In a related fashion, the Sensitivity analysis in page 10 (Section Sensitivity analyses for the genome-wide associations in split-sample experiments, sex-stratified experiments, and non-Caucasian populations) must acknowledge that sample size is much smaller than in the original GWAS which dictates significance findings. Instead, genetic correlations among original, sex-specific, and split-sample GWAS would paint a broader picture on whether primary GWAS results generally converge and are robust. Also please indicate the sample sizes and effect sizes in this results section for better readability and interpretability.

Discussion is lacking a summary and limitation section

- The discussion section is lacking a strong introductory paragraph summarising the key trends found in this study. This may be a good opportunity to summarise evidence the authors may have for the fact that multimodal BAGs truly yield distinct phenotypes (i.e., phenotypic correlations between BAGs), and are associated with distinct genetic architecture (e.g., genetic correlations among BAGs, distinct GWAS hits & distinct druggable targets). Such a summary paragraph could then lead into a more nuanced discussion of the specific biological pathways.

- In my opinion it is imperative to include a limitation section in the discussion, especially because the methods used in this manuscript are not trivial and their results can only be interpreted with certain caveats. For example, LDSC and MR come with many assumptions, brain age is a residual score that captures measurement error, cross-sectional data is limited by cohort effects and we don't know how accurately brain age captures real longitudinal changes, and many more.

- Would the authors consider adding a broad conclusion section?

Methods are lacking details to allow future studies to replicate results

- Page 29, Method 1: It is not transparent how UKB participants were included in the study (e.g., availability of MRI data?). The authors state that participants with certain clinical diagnoses were excluded, but do not list the criteria (i.e., the use of "etc." prohibits replication). UKB field IDs and the sample size of exclusions and inclusions are required to allow future studies to replicate findings. Please search the manuscript for more instances of etc. and specify explicitly what was done (e.g., page 35). To make the Methods more accessible, please indicate the sample size of the independent test dataset, as well as Ns available for imaging phenotypes.

- Page 30, Method 2: I assume that the outcome variable here was age, and the model was trained using multiple imaging variables as predictors. This may be evident to a brain age researcher but needs to be made explicit. This section also requires more explicit details on how the models were trained: model assumptions and functionality, software packages, input parameters. Please explain what a ReLU layer is.

- Page 31, Method 3: Could the authors please explain why Image processing is positioned in the manuscript after the machine learning models. I assume the imaging variables were processed prior to brain age predictions serving as predictor variables. Please consider swapping the sections. Could the authors also please explain why they processed the variables themselves rather than using IDPs as provided by UKB? One may argue it would benefit comparability with other brain age studies to use pre-processed UKB IDPs.

- Page 32, Diffusion MRI processing, "Finally, to overcome the overfitting problem due to the high correlations between FA, MD, ODI, and NDI within the same WM tract, we used only the 48 FA WM-IDP to fit the models for brain age prediction": It is unclear which 48 variables were selected here, and why.

- Page 32, Resting-state FC: "four components were removed due to artifactual components": It is unclear how an artifactual component was identified.

- Page 32, genetic analyses: The authors do not state specifically which quality steps were performed to clean the genetic data. For example, did they exclude/ include genetic variants based on minor allele frequency, missingness, imputation quality? Were participants excluded due to missingness, or sex checks? What was the final amount of participants and SNPs? Please justify the choice of the PLINK software, instead of another more sophisticated software such as REGENIE (<https://rgcgithub.github.io/regenie/>). It is also unclear whether the GWAS were conducted in the whole sample, or whether it was performed in the training or validation subsets. Please include sample sizes throughout the manuscript, including the abstract.

- Page 35, two-sample MR: The authors do not name the five different MR methods they claim to have used, and they also do not name the 7 exposure variables included in the analysis. MR assumptions are not explicitly stated, and it is not stated how the different methods address MR assumptions. Please consider reporting MR findings according to standard guidelines as outlined here: Strengthening the reporting of observational studies in epidemiology using mendelian randomisation (STROBE-MR): explanation and elaboration | The BMJ

- Page 36, PRS prediction: It is unclear from reading this section what the outcome and predictor variables were. I gathered from elsewhere in the manuscript that the outcome variables are BAG phenotypes and the predictor variables are PRSs capturing genetic propensity towards 36 other traits, diseases and risk factors. Please be more specific in this methods section.

Minor points that would improve the readability/accessibility of the manuscript

- Page 5: "The phenotypic heterogeneity of multimodal human brain age derived from three MRI modalities and four machine learning models". This statement seems misleading – the authors did not derive the heterogeneity of multimodal human brain age, but they derived brain age based on different imaging modalities and discovered they were heterogeneous in the genetic correlates they yielded. Would the authors be able to be more specific here?

- Page 3: Please consider re-wording the following sentence which is challenging to comprehend: "This deviation can be caused by adverse factors, such as AD9, leading to an accelerated brain age (i.e., a

positive BAG) or protective factors, such as physical activity or education, resulting in a decelerated brain age (i.e., a negative BAG)."

- Page 4: "In this context, we postulate that AI-derived GM, WM, and FC-BAG can serve as robust, complementary endophenotypes²¹ – close to the underlying etiology – for precision medicine": Did the authors mean that endophenotypes are closer to their genetic underpinnings as compared with other phenotypes such as clinical diagnoses, for example?

- Page 5: Please provide references for the statement: "Other studies have thoroughly evaluated machine learning models for predicting brain age²⁸, but we selected these as they represent methods currently used in the field" [REF].

- Page 5: Considering the focus of the manuscript is the heterogeneity between BAGs and their distinct genetic underpinnings, I suggest it may improve the readability of the manuscript to shift the discussion of how different machine learning algorithms performed to the supplementary. Instead, I suggest it would be better suited to showcase evidence that there indeed is heterogeneity between BAGs.

- Page 5: Could the authors please explain which indicators they used to find that some machine learning models were over-fitted?

- Page 8, "The genomic loci linked to GM, WM, and FC-BAG were distributed throughout the human genome, with many locations being distinct": It is unclear what the authors mean by many locations being distinct. Is it that individual BAG phenotypes yielded associations with SNPs that were not found for other BAG phenotypes? Would the authors be able to make this statement clearer?

- Page 10, "While the other four genomic loci did not surpass the genome-wide P-value threshold, they all exhibited local minima and featured the same top lead SNPs or existed in a state of high linkage disequilibrium.": If this is what the authors mean by "local minima", would you not expect the most significant SNPs to have the largest rather than the smallest effect sizes?

- Figure 4: I suggest this Figure would be more accessible to the reader if there was one location in the x-axis for each trait, whereby estimates for GM, WM and FC would be displayed alongside each other. It took me a while to understand the x-axis repeated the same traits multiple times

Reviewer #3 (Remarks to the Author):

Overall

This is a very interesting article exploring the relation of Brain Age Gap (BAG) with genetic information. The authors consider X key questions:

- what is the genetic heterogeneity of multimodal BAG
- what are the phenotypes associated with BAG related loci and what are their functional / disease related characteristics
- can we establish the causal relationships between protective/risk factors and decelerated/accelerated brain age

I found the article methods well described and questions of interest. The findings are worth communicating but the general impression is that results are not very biologically compelling and significance maybe mostly due to the very large sample available in the UK Biobank. The authors present a large set of methods and results, each could deserve more in depth validation or analyses, for instance the image derived phenotypes used to compute brain age aren't the most sensitive ones reported (More et al, 2023). In terms of the general approach, the authors seem to first look at genomic loci linked to BAG - then look if these loci are related to disease with a genome wide association - but a stronger analysis would be to compare directly the loci found with BAG to the loci found in association to disease, and assess the likelihood of the proximity of these loci using non parametric techniques.

Specific

* BAG : could the authors explain why is the MAE greater than the one found in many brain age

studies ? (e.g. Leonardsen et al. 2022, He et al. 2021, Hahn et al. 2022, Wood et al. 2022, Peng et al. 2021)

* More et al 2023 suggest that voxel / vertex wise methods are to be preferred for brain age computation, do the author think the results would have been different with the associated BAG ?

* How does the genetic loci of BAG related to loci of AD / aging ? can you think of an analysis that would demonstrate if the relation isn't found by chance ?

* BAG computed cross sectionally may be questionable in the view of the brain charting results (Di Biase, PNAS 2023)

* There has been considerable analytical variability reported in brain imaging - are results robust with choice of pipelines to derived IDP ?

* "the locus associated with GM-BAG (top lead SNP: rs61732315, 1q32.1) and 164the locus related to WM-BAG (top lead SNP: rs11118475, 1q32.2) were in proximity" : it is unclear if it can be established that this proximity is indicative of a specific result: is this testable (can we test the hypothesis that loci are related) ?

* Was the discovery of the 6 loci done on the full sample before doing the split-sample analysis ? what is the number of time these loci are found again across many splits ?

* Genetic covariance: for interpretation, it would be important to report the heritability of the traits on which the correlation is computed

* Polygenic risk scores of other diseases weakly predict multimodal BAG: while the p-values are very small, this seems to be a side effect of the number of sample. The additional variance explained of less than 0.3% does not seem to be biologically relevant. One possible analysis would be to see which loci are shared between PRS predicting loci and those associated with BAGs.

* Causality analyses: The authors performed MR analyses to establish whether the clinical traits previously associated with the genomic loci associated with BAG were a cause or a consequence of GM, WM, and FC-BAG. These are interesting analyses but there are strong assumptions for MR analyses and it is unclear if these are met here (e.g., the "no horizontal pleiotropy" assumption).

Author response letter, ID: NCOMMS-23-20963-T

To facilitate the revision for the reviewers in this round, the comments from each reviewer are in black font in this response letter; our responses are in blue font. In the revised manuscript, we tracked the changes in the yellow-colored text. We look forward to additional comments and suggestions for improving our manuscript.

REVIEWER COMMENTS

Reviewer #1 (Remarks to the Author):

Wen et al. used multimodal magnetic resonance imaging and artificial intelligence to examine the genetic heterogeneity of the brain age gap (BAG) derived from gray matter volume (GM-BAG), white matter microstructure (WM-BAG), and functional connectivity (FC-BAG). The authors identified sixteen significant genomic loci. A gene-drug-disease network highlighted genes linked to GM-BAG for treating neurodegenerative and neuropsychiatric disorders and WM-BAG genes for cancer therapy. GM-BAG showed the highest heritability enrichment for genetic variants in conserved regions, whereas WM-BAG exhibited the highest heritability enrichment in the 5' untranslated regions, oligodendrocytes and astrocytes but not neurons. Notably, Mendelian randomization identified causal risk effects of triglyceride-to-lipid ratio in very low-density lipoprotein and type 2 diabetes on GM-BAG and AD on WM-BAG.

Thank you for the summary of our study.

First, we summarized the major changes in the revised manuscript and addressed the comments the three reviewers consistently raised. Many of these comments among reviewers converge to similar constructive suggestions.

- The term "heterogeneity" was found to be potentially misleading, as reviewers #1 and #2 commented. Consequently, we have adjusted the emphasis on the term "heterogeneity" throughout the manuscript. We also reorganized the paper and rewrote many places, guided by all three reviewers.
- To justify the brain age prediction performance as presented in our manuscript compared to those reported in prior studies (commented by all three reviewers), we have integrated a new paragraph to discuss these findings. We also conducted an additional experiment involving GM-BAG. Specifically, we employed and compared the performance (brain age prediction and also GWASs) with Support Vector Regression (SVR) on voxel-wise RAVENS maps (GM-BAG-voxel) and MUSE-ROI features (GM-BAG-ROI).
- We performed three additional sensitivity checks (on top of split-sample, sex-stratified, and non-European GWASs) to test the robustness of our main GWASs using European ancestry (suggested by all reviewers #1 and #2).
 - We used SVR to perform feature type-specific GWASs, focusing on different feature types: GM-BAG-ROI versus GM-BAG-voxel.
 - We used MUSE ROIs (GM-BAG) to perform machine learning-specific GWASs, comparing GM-BAG derived from Lasso regression vs. SVR.
 - We reran the three main GWASs using fastGWA (i.e., mixed linear models to account for cryptic population stratification) to justify that there is no substantial genomic inflation in our main GWAS using PLINK linear regression models.
- Furthermore, we calculated the genomic inflation factor (λ) and LDSC intercept (b) in the main GWASs to support further that there is no potential genomic inflation in our main GWASs using European ancestry.
- We derived the polygenic risk scores (PRS) for the GM, WM, and FC-BAG using both Plink (Clumping + Threshold approach) and a more advanced Bayesian approach (PRS-CS) (suggested by all reviewers #1, #2, and #3). Overall, PRS-CS outperforms PLINK in predicting the three BAG-PRSs in the test dataset (split2 GWAS).
- We calculated the phenotypic and genetic correlations for each pair of BAGs in Fig. 1E and Fig. 2F (suggested by reviewer #2).

- We developed a multi-organ web portal (MEDICINE: <http://labs.loni.usc.edu/medicine/>) to disseminate our findings, including showing the Manhattan & QQ plots and allowing the community to download our GWAS summary statistics.
- The keyword "Caucasian" implies racism. We changed the word "Caucasian" to "European" throughout the paper guided by this Nature article (<https://www.nature.com/articles/d41586-021-02288-x>), as the word Caucasian roots in racist taxonomies used to justify slavery.

Secondly, considering that all three reviewers have highlighted our study's significance to imaging genetics, we would like to provide a brief overview (non-exhaustive) of previous GWAS papers conducted on BAG and outline the scientific progress that our study contributes.

It's important to acknowledge that brain age is not a newly emerging biomarker; it has been extensively studied in the literature, as evidenced by pertinent literature cited by reviewer #3. What sets our study apart regarding significance and innovation lies in two aspects. Firstly, prior GWASs of brain age have often been confined to specific aspects, such as GWASs focusing solely on GM-BAG. Secondly, some studies have restricted themselves to the analyses of genome-wide associations without proceeding to undertake comprehensive post-GWAS analyses (e.g., portioned heritability, gene-drug-disease network, Mendelian randomization, etc.) aimed at partially validating the genetic signals identified. Our study advances the field by adopting a holistic approach that addresses both these limitations, thus contributing to a more comprehensive understanding of the genetic underpinnings of brain age. We have integrated this into the Main section in the revised manuscript.

We want first to acknowledge several representative previous GWAS works on this topic – science is an incremental process built upon the foundation of prior research:

- Kaufmann et al. Nat Neurosci, 2019 performed GWAS on **GM features**, and they then estimated the SNP-based heritability of the brain age using LDSC (<https://www.nature.com/articles/s41593-019-0471-7>).
- Jonsson et al., 2019, Nature Communications: (<https://www.nature.com/articles/s41467-019-13163-9>) also performed the GWAS on **GM-BAG** derived from a deep learning model.
- Smith, S. M. et al. 2020, eLfi (<https://elifesciences.org/articles/52677>) performed GWAS on BAGs derived from multimodal brain IDPs; they primarily only focused on GWAS but did not perform comprehensive post-GWAS analyses.
- Leonardsen et al., 2023, Molecular Psy (<https://www.nature.com/articles/s41380-023-02087-y>) performed GWAS on **GM-BAG** derived from a deep learning model and then investigated the causal relationship with other traits.

This summary is by no means exhaustive.

While the authors claimed that these results provide valuable insights into the genetic heterogeneity of human brain aging, with clinical implications for potential lifestyle and therapeutic interventions, I found the overall flow of their manuscript confusing and the key word "heterogeneity" not being adequately reflected.

We appreciate the reviewer's input, and we recognize that the term "heterogeneity" may not precisely capture the primary insights of this study. Heterogeneity typically denotes dissimilarities among constituent elements (such as individual-level neuroanatomical patterns in the context of Alzheimer's disease) that collectively form a larger whole.

We slightly changed the angle. Our present study posited that various MRI modalities, including T1-weighted, diffusion, and resting-state functional MRI, commonly employed within UKBB, could potentially capture distinct neurobiological facets of human brain aging. For instance, T1-weighted MRI typically aids in studying brain atrophy, diffusion MRI (with metrics such as DTI) is a common tool for examining disruptions in white matter microstructure, and resting-state fMRI is utilized to explore brain functional connectivity. Consequently, we aimed to unravel the genetic commonalities and distinctions underlying these diverse aspects of human brain aging. In response to this feedback, we have adjusted the study title from "The Genetic Heterogeneity of Multimodal Human Brain Age" to "The Genetic Architecture of Multimodal Human Brain Age" and have dialed down the emphasis on heterogeneity throughout the revised manuscript. In the Main section of the revised manuscript, we explicitly explained the hypothesis/motivation for the GWAS and post-GWAS analyses (Lines 70-81):

"It is crucial to holistically identify the genetic factors associated with multimodal BAGs (GM, WM, and FC-BAG), where each BAG reflects distinct and/or similar neurobiological facets of human brain aging. Furthermore, dissecting the genetic architecture of human brain aging may determine their causal implications, which is essential for developing gene-inspired therapeutic interventions. Finally, numerous risk or protective lifestyle factors and neurobiological processes may also exert independent, synergistic, antagonistic, sequential, or differential influences on human brain health. Therefore, a holistic investigation of multimodal BAGs is urgent to fully capture the genetics of human brain aging, including the genetic correlation, gene-drug disease network, and potential causality. In this study, we postulate that AI-derived GM, WM, and FC-BAG can serve as robust, complementary endophenotypes²³ – close to the underlying etiology – for precise quantification of human brain health."

I have a few comments:

The overall connection and logics of this manuscript has a room for improvement. In the first paragraph of results, the key words should be (in my understanding) "heterogeneity" and "BAG", yet neither of them appeared in a timely manner.

We really appreciate this comment from the reviewer. As suggested by the reviewer, we chose to down-tone the heterogeneity aspect of this work.

A huge paragraph was used to describe model prediction, model performance, models, sex stratification. BAG was defined as the difference between an individual's predictive brain age and his/her chronological age.

We agree with the reviewer on this comment. The reasons why we started the Result section with the brain age prediction performance are stated as follows: *i*) age prediction using MRI is fundamental to the following GWAS and post-GWAS analyses, although there was extensive previous literature systematically investigating this topic; *ii*) all three reviewers also commented on the age prediction performance of our machine learning methods, which we believed that we should give space to elaborate this point at the very beginning of the Result section.

In the revised manuscript (Lines: 127-138), we now rephrased the first paragraph to explain the three main sections/objectives:

"In the literature, other studies³⁰⁻³³ have thoroughly evaluated age prediction performance using different machine learning models and input features. More et al.³⁴ systematically compared the performance of age prediction of 128 workflows (MAE between 5.23–8.98 years) and showed that voxel-wise feature representation (MAE approximates 5-6 years) outperformed parcel-based features (MAE approximates 6-9 years) using conventional machine learning algorithms (e.g., Lasso regression). Using deep neural networks, Peng et al.³⁰ and Leonardsen et al.³¹ reported a lower MAE (nearly 2.5 years) with voxel-wise imaging scans. However, we previously showed that a moderately

fitting convolutional neural network (CNN) obtained significantly higher differentiation (a larger effect size) than a tightly fitting CNN (a lower MAE) between the disease and health groups³⁵. To summarize, our study's brain age prediction performance aligns with those reported in the existing literature, considering the utilization of low-dimensional hand-crafted IDPs and conventional machine learning algorithms³⁴.”

Yet this difference was not introduced or described early enough (or strong enough) in this section of results.

In the revised manuscript of the Main section (lines 54-59), we added one sentence to explain how the BAG is calculated:

“More precisely, the difference between an individual's AI-predicted brain age and chronological age – brain age gap (BAG) – provides a means of quantifying an individual's brain health by measuring deviation from the normative aging trajectory. BAG has demonstrated sensitivity to several common brain diseases, clinical variables, and cognitive functions⁹, presenting the promising potential for its use in the general population to capture relevant pathological processes.”

My understanding is that IDP does not equal to BAG, prediction neither, but rather the difference).

The reviewer's understanding is correct. In the context of this work, IDP refers to all imaging features (i.e., MUSE ROIs for GM; FA, MD of DTI metrics, and ODI, NDI of NODDI metrics for WM; ICA-derived functional connectivity components for FC). BAG was the AI-predicted brain age - chronological brain age; the AI models (SVR, LASSO regression, MLP, and neural network) fit these multimodal IDPs as features in a brain age prediction task. The term (IDP) was first introduced by Elliot et al. (Nature, 2018) in the first large-scale multimodal IDP GWAS (Elliott, L. T. *et al.* Genome-wide association studies of brain imaging phenotypes in UK Biobank. *Nature* **562**, 210–216 (2018).)

At the very end, in the fifth point, finally comes the heterogeneity of BAG. This is very confusing and weakened the necessity of "leading us to investigate the underlying genetic determinants responsible for this phenotypic heterogeneity". I do not see the point, these one-sentence described results are not strong enough to be further pursued using a genetic approach.

We agree with the reviewer on this comment. To further address the reviewer's suggestion, we *i*) revised the Main section by down-toning the keyword of heterogeneity (as explained above), *ii*) deleted the sentence "leading us to investigate the underlying genetic determinants responsible for this phenotypic heterogeneity", *iii*) added one sentence to state our overall hypothesis why we wanted to study the genetics of the multimodal BAG (lines 70-81), and added a new paragraph at the beginning of the Result section, highlighting the three main objectives of the current study (Lines: 96-107):

“In the first section, we objectively compared the age prediction performance of four machine learning methods using these GM, WM, and FC-IDPs (**Fig. 1A**). To this end, we employed a nested cross-validation (CV) procedure in the training/validation/test dataset ($N=4000$); an independent test dataset ($N=38,089$)^{26,27} was held out – unseen until we finalized the models using only the training/validation/test dataset (**Method 1**). The four machine learning models included support vector regression (SVR), LASSO regression, multilayer perceptron (MLP), and a five-layer neural network (i.e., three linear layers and one rectified linear unit layer; hereafter, NN)²⁸ (**Method 3**). The second section focused on the main GWASs using the European ancestry population ($31,557 < N < 32,017$) and their sensitivity checks in six scenarios (**Method 4A**). In the last section, we validated the GWAS findings in several post-GWAS analyses, including genetic correlation, gene-drug-disease network, partitioned heritability, PRS calculation, and Mendelian randomization (**Method 4**).”

The main problem with GWAS (the second and the third part of results) is a lack of real-meaning validation.

We concur with the reviewer's comment. A genome-wide association study (GWAS) – the second part of our results (**Fig. 2**) – fundamentally examines the association, often in a univariate manner, between a phenotype of interest and genetic variants, which can include single nucleotide polymorphisms (SNPs), genes, or copy number variants.

However, compared to previous GWASs on brain age gaps, our study may partially validate these GWAS signals by conducting several post-GWAS analyses using state-of-the-art computational genomics statistical methods. These include:

- Genetics correlation: we expected the three BAGs to be genetically correlated with other brain diseases and subtypes (**Fig. 4A**). For example, we found a significant positive genetic correlation between the GM- and WM-BAG and one subtype of AD, defined by a semi-supervised AI method (<https://www.biorxiv.org/content/10.1101/2022.09.16.508329v2>, under review at another journal) and characterized by global brain atrophy. Of note, AD1 was originally defined in ADNI, and the GM-, WM-BAG in this paper was defined in part of the UKBB data.
- Gene-set enrichment analysis (GSEA) (additional experiment suggested by reviewer #3): In **Fig. 2B**, we found that GM- and WM-BAG-associated loci were previously linked to AD in the GWAS Catalog. We performed one additional GSEA analysis using GENE2FUNC on FUMA to test (i.e., hypergeometric test) if the genes associated with the GM and WM-BAG were enriched in a pre-defined gene set defined by "Alzheimer's disease in APOE e4- carriers" by the GWAS Catalog. We indeed found significant enrichment for GM-BAG-associated genes: P-value = 2.84460968895911e-07.
- Cell type-specific partitioned heritability estimates (**Fig. 4C**): WM-BAG exhibited significant heritability enrichment in oligodendrocytes – one type of neuroglial cells. FC-BAG showed such enrichment in astrocytes, the most prevalent glial cells in the brain. This, especially with the WM-BAG, should be biologically meaningful, as oligodendrocytes are primarily responsible for forming the lipid-rich myelin structure, whereas astrocytes play a crucial role in various cerebral functions, such as brain development and homeostasis. Convincingly, our previous GWAS on WM-IDP (not BAG) (Zhao et al. Science, PMID: 34140357) also independently identified considerable heritability enrichment in glial cells, especially oligodendrocytes. This consistency largely validated our current GWAS findings.
- Gene-drug-disease network (**Fig. 3**): Our bioinformatic analyses simply performed gene mapping (by physical position, chromatin interaction, and/or eQTL mapping), searched these mapped genes as targets in the Drug Bank database and the Therapeutic Target Database, and provided putative evidence that these BAG-related genes were used as target genes for treating brain-related diseases, such as Semorinemab (RG6100), an anti-tau IgG4 antibody, being investigated in a phase-2 clinical trial (trial number: NCT03828747), which targets extracellular tau in AD – the most common form of dementia – to reduce microglial activation and inflammatory responses.
- Similarly, Mendelian randomization results also provided confirmation that other brain diseases (e.g., AD) or systemic diseases (e.g., type 2 diabetes) could causally influence human brain age.

That being said, we agree that all these post-GWAS analyses are not "real meaning" validation. It often requires additional expertise, resources, and time for real-time validation. For instance, a potential approach could involve the testing and validating BAG-associated genes using induced pluripotent stem cell (iPSC)-derived cortical neuron cultures to explore drug repositioning from drugs for neurodegenerative diseases. However, this approach may not be feasible from an ethical standpoint, as it would not be ethically sound to administer drugs to delay human brain aging. Rather, our study represents statistical, bioinformatic, and genetic confirmations/validation for the potential use as an AI-derived biomarker to digitize human brain health.

Sensitivity analyses in split-sample experiments, sex stratified experiments, and non-Caucasian populations do not really account for an independent validation.

This is totally correct. Within our manuscript, it's important to clarify that we did not assert our sensitivity analyses of the main European GWASs as an independent validation. Instead, these analyses served as a sensitivity check to assess the concordance/robustness of the main GWAS results against six alternative experimental setups.

All GWASs involving UKBB data have this limitation. As far as we know, there are other existing large-scale brain imaging genetics consortia in the field, such as the ADNI study (specific to AD at older ages), the ABCD study for brain development at adolescence, the PNC study, and the HPC study. However, due to the smaller sample sizes in these studies compared to the UKBB and differences in genetic/imaging sequences and population ethnic origins, it will be challenging to generalize the findings. As evidenced in our non-European GWAS sensitivity check (compared to the main three GWAS on European ancestry populations), we found low concordance rates (lines 212-215):

"The concordance rates of the GWASs using non-European ancestry populations (as replication, $4646 < N < 5091$) were low compared to the main GWASs using the European population: only 13.78% for GM-BAG and 41.94% for WM-BAG (P -value < 0.05) (Supplementary eFigure 5 and Supplementary eFile 4)."

However, the genetic signals were pretty robust when the sensitivity analyses were performed within European ancestry populations (Lines 196-239).

Many factors may impede the independent validation, as stated above. In the revised manuscript, we also discussed this as a limitation in the added Limitation section (Lines 539-550):

"Second, the generalization ability of the GWAS findings to non-European ancestry is limited, potentially due to small sample sizes and cryptic population stratification. Future investigations can be expanded to encompass a broader spectrum of underrepresented ethnic groups, diverse disease populations, and various age ranges spanning the entire lifespan. This expansion can be facilitated by leveraging the resources of large-scale brain imaging genetic consortia like ADNI⁸⁰, focused on Alzheimer's disease, and ABCD⁸¹, which centers on brain development during adolescence."

Especially the non-Caucasian analysis, number one it's under-powered, number two it's way too early to generalize to another ethnicity. I would suggest tell a full story in the Caucasians as a first step.

We apologize for the confusion. We presented the main genetic analyses using participants with European ancestry (i.e., all post-GWAS analyses as software specify the ancestry groups for the reference genome to compute the LD matrix). We have clarified this in the revised manuscript at all places (e.g., Lines 163-164).

Another issue is that I do not really see a heterogeneity. Both GM-BAG and WM-BAG are associated with brain related issues, neurology or psychiatry. I cannot agree with what's been stated in their abstract, "with GM-BAG loci showing abundant associations with neurodegenerative and neuropsychiatric traits, WM-BAG loci implicated in cancer and Alzheimer's disease (AD), and FC-BAG in insomnia", if this reflects heterogeneity, it also reflects homogeneity.

We agree with the reviewer on this comment. As addressed above, we have down-toned the heterogeneity in the writing. We removed this sentence in the abstract in the revised manuscript (Abstract).

Furthermore, as suggested by reviewer #2, we now changed **Fig. 1E** and **Fig. 2F** to show the phenotypic correlation (p_c , lines **139-143**) and genetic correlation (g_c , lines **187-194**) between each pair of BAGs, showing that these correlations are small to medium.

“Finally, we calculated the phenotypic correlation (p_c) between GM, WM, and FC-BAG using Pearson's correlation coefficient. GM-BAG and WM-BAG showed the highest positive correlation ($p_c=0.38$; P-value $<1\times 10^{-10}$; $N=30,733$); GM-BAG ($p_c=0.09$; P-value $<1\times 10^{-10}$; $N=30,660$) and WM-BAG ($p_c=0.10$; P-value $<1\times 10^{-10}$; $N=31,574$) showed weak correlations with FC-BAG (**Fig. 1E**).”

“Finally, we calculated the genetic correlation (g_c) between the GM, WM, and FC-BAG using the LDSC software. GM-BAG and WM-BAG showed the highest positive correlation ($g_c=0.49$; P-value $<1\times 10^{-10}$); GM-BAG ($g_c=0.20$; P-value=0.025) and WM-BAG ($g_c=0.29$; P-value=0.005) showed weak correlations with FC-BAG (**Fig. 2F**). The genetic correlations largely mirror the phenotypic correlations, supporting the long-standing Cheverud's Conjecture³⁹. We also verified that these genetic correlations exhibited consistency between the two random splits (split1 and split2: $15,778<N<16,008$), sharing a similar age and sex distribution (**Supplementary eFigure 2**).”

In the "Multimodal BAG is genetically correlated with AI-derived subtypes of brain disorders" section, justify the reason why original subtype is not genetically associated with multimodal BAG? These phenotypes are true phenotypes (instead of imputed or calculated or derived), why these phenotypes did not stand out in the first place?

To be certain, we think the reviewer intended to ask: Why is the original disease diagnosis (not the subtype)?

Thank the reviewer for the comment on this observation. These AI-generated disease subtypes were established in our previous studies utilizing semi-supervised clustering methods in a purely data-driven fashion. Please refer to our recent chapter on the intuition of semi-supervised clustering methods (https://link.springer.com/protocol/10.1007/978-1-0716-3195-9_16), and a new semi-supervised model recently published in Nature Communications using a generative adversarial network to model disease heterogeneity (<https://www.nature.com/articles/s41467-021-26703-z>). These AI-derived subtypes, in essence, capture more homogeneous disease effects (e.g., reflected by neuroanatomical patterns derived from T1-weighted MRI) than the conventional "unitary" disease diagnosis (such as AD was defined by cognitive scores and, more recently, neurobiological markers from PET and MRI).

Indeed, the significant genetic correlations were from the AI-derive subtypes. For instance, we found that GM-BAG is positively associated with AD1 (0.40 ± 0.04 ; P-value= 2.03×10^{-24} , **Supplementary eTable 4**), but this positive genetic correlation was not significant with AD (case-control diagnosis). This partially reflects the fact that the "true phenotype - AD diagnosis" used in the original GWAS study (PubMed ID: 30820047, Nature Genetics) was not precisely defined – most AD GWASs define the AD cases using the maternal/paternal history of dementia (such as in UKBB, where the AD diagnosis (G30) was only available for a very small proportion of the imaging population ($<10\%$)), but not based on any neurobiological evidence (e.g., tau, amyloid PET) or cognitive function (e.g., ADAS-

Cog-XX score from ADNI). This broadly defined AD diagnosis is not clinically precise but indeed increases the sample sizes required by modern GWASs to boost the statistical power. However, we also observed that FC-BAG was significantly genetically associated with OCD (Fig. 4A)

We explained and discussed this in lines 300-306 in this revised manuscript:

“Our analysis revealed significant genetic correlations between GM-BAG and AI-derived subtypes of AD (AD1⁴), autism spectrum disorder (ASD) (ASD1 and ASD3⁴⁶), schizophrenia (SCZ1⁴⁷), and obsessive-compulsive disorder (OCD)⁴⁸; WM-BAG and AD1, ASD1, SCZ1, and SCZ2; and FC-BAG and education⁴⁹ and SCZ1. Detailed results for r_g estimates are presented in **Supplementary eTable 4**. These subtypes, in essence, capture more homogeneous disease effects than the conventional "unitary" disease diagnosis, hence serving as robust endophenotypes²³.”

In "Polygenic risk scores of other diseases weakly predict multimodal BAG", how are the 36 diseases selected?

Thanks for this comment. We did not directly derive these PRSs. The 36 PRSs were downloaded directly from UKBB and were originally derived by Thompson et al. (<https://www.medrxiv.org/content/10.1101/2022.06.16.22276246v2>).

As reviewers #2 and #3 also gave very constructive comments on the PRS results, we removed these old results and performed a new PRS analysis. Specifically, we derived the PRSs using both Plink and PRC-CS (a Bayesian method) for GM, WM, and FC-BAG using the two-split GWASs. That is, we used split1 GWAS as training/base data for the weights and split2 GWAS as testing/target data to calculate the incremental R² to predict the phenotype of themselves (GM, WM, and FC-BAG). We compared the predictive power of the three BAG-PSCs and found that PRS-CS-derived PRSs are more predictive than those from the PLINK C+T approach (Fig. 4D and lines 340-350):

"We derived the PRS for GM, WM, and FC-BAG using the conventional C+T (clumping plus P-value threshold) approach⁵⁵ via PLINK and a Bayesian method via PRS-CS⁵⁶ (Method 4H).

We found that the GM, WM, and FC-BAG-PRS derived from PRS-CS significantly predicted the phenotypic BAGs in the test data (split2 GWAS, 15,697 < N < 15,940), with an incremental R² of 2.17%, 1.85%, and 0.19%, respectively (Fig. 4D). Compared to the PRS derived from PRS-CS, the PLINK approach achieved a lower incremental R² of 0.81%, 0.45%, and 0.14% for GM, WM, and FC-BAG, respectively (Supplementary eFigure 9). Overall, the predictive capacity of PRS is moderate, in line with earlier discoveries involving raw imaging-derived phenotypes, as demonstrated in Zhao et al.¹³, where PRSs developed for seven selective brain regions were able to explain roughly 1.18% to 3.93% of the phenotypic variance associated with these traits.”

While the authors just claimed in a few paragraphs ahead, that "These results indicate potential genetic overlap and causal links between the multimodal BAG and other clinical traits."

We removed this sentence. Initially, we thought it would be a good idea to summarize the current section's findings and connect the current section to the next section. These GWAS results could indicate a potential genetic correlation in the following post-GWAS analysis.

In the last paragraph of results, the title said, "Triglyceride, type 2 diabetes, breast cancer, and AD are causally associated with GM and WM-BAG", while the body said, "Based on our sensitivity analysis and the consistency of the results obtained from all five MR methods (Supplementary eFile 3), we showed evidence for a putative causal effect of the triglyceride-to-lipid ratio in VLDL on GM-BAG", and the abstract said "Mendelian randomization identified causal risk effects of triglyceride-to-lipid ratio in very low-density lipoprotein and type 2 diabetes on GM-BAG and AD on WM-BAG", which version is accurate? These three

versions each might be correct to some extent, but the authors have the obligations to guide through your readers and be consistent throughout.

We apologize for this confusion. The section title ("Triglyceride, type 2 diabetes, breast cancer, and AD are causally associated with GM and WM-BAG") was not precise.

To summarize, we found four significant causal relationships (after multiple comparisons and sensitivity checks. **Fig. 5A**):

- Type 2 diabetes on GM-BAG
- Breast cancer on GM-BAG
- Triglyceride-to-lipid ratio in VLDL on GM-BAG
- AD on WM-BAG

In the revised manuscript, we changed the section title to: "The potential causal relationships between GM and WM-BAG and other clinical traits".

Regarding this sentence: "*Based on our sensitivity analysis and the consistency of the results obtained from all five MR methods (Supplementary eFile 3), we showed evidence for a putative causal effect of the triglyceride-to-lipid ratio in VLDL on GM-BAG*", this sentence was precisely written based on our sensitivity checks. We carefully scrutinized the robustness of the Mendelian randomization results as MR has very strong instrumental variable (IV) assumptions (as also pointed out by Reviewer #3).

Fig. 5B-E showed the specific sensitivity check results for the causal relationship from Triglyceride-to-lipid ratio in VLDL on GM-BAG. In the revised manuscript, we created a new section (Sensitivity analyses for Mendelian randomization) to present the details of these sensitivity analyses (Lines: **391-414**).

To summarize, the causal effect is robust, although we indeed found two outlier SNPs potentially exerting directional horizontal pleiotropy (using MR Egger regression). All sensitivity check results for the other three significant results are presented in **Supplementary eFile 9**.

In the Abstract, we now corrected the writing:

"Mendelian randomization identified potential causal risk effects of several exposure variables on brain aging, such as type 2 diabetes on GM-BAG (odds ratio=1.05 [1.01, 1.09], P-value=1.96x10⁻²) and AD on WM-BAG (odds ratio=1.04 [1.02, 1.05], P-value=7.18x10⁻⁵)."

We also want to bring caution to the causal effect of breast cancer (exposure) on GM-BAG (outcome) due to population selection bias (DOI: <https://doi.org/10.1093/ije/dyy202>) – participants who are diagnosed with breast cancer are more likely to quit the study. We have now expanded this discussion into two sentences in the revised manuscript in the Discussion section (Lines **483-485**):

"In addition, it's important to exercise caution when considering the potential causal link between breast cancer and GM-BAG, as MR analyses are susceptible to population selection bias⁷³ due to the high breast cancer mortality rate."

Finally, we also included this as one limitation in the Limitation section (Lines **551-552**):

"Third, it's important to exercise caution when interpreting the results of this study due to the various assumptions associated with the statistical methods employed, including LDSC and MR."

A few minor comments, several abbreviations appeared without any explanation, for example, MAE, FA. I don't even know that FA, MD, NDI and ODI are four diffusion metrics until I read a few lines further, and wonder how many of them still there?

We appreciate this comment. In the revised manuscript, we defined these abbreviations in the Main section (lines **65-66**).

The "independent dataset" was never introduced – please note that Nature paper exhibits results before methods, so the authors are responsible for a properly organization of their texts. Not necessarily lengthy, but enough information should be revealed to support for a barrier-free reading).

As the reviewer mentioned, Nature journals arrange the Method section after the Result section. Sometimes, it is challenging for both authors and readers to follow these details.

First, we defined the training/validation/test dataset vs. the independent test dataset in **Method 1**. To address this point at the very first occurrence, we now added this information in lines **98-100**:

"To this end, we employed a nested cross-validation (CV) procedure in the training/validation/test dataset ($N=4000$); an independent test dataset ($N=38,089$)^{26,27} was held out – unseen until we finalized the models using only the training/validation/test dataset (**Method 1**)."

"The genomic loci linked to GM, WM, and FC-BAG were distributed throughout the human genome" – not accurate. These sentences are best suit for a height or BMI GWAS, there we say signals distributed throughout human genome.

Thank the reviewer for this suggestion and correction. All three reviewers raised questions about this sentence. In the revised manuscripts, we removed this sentence. Initially, we wanted to show that some genomic loci are consistently (by physical position) found between different BAGs. We could directly test this by Bayesian colocalization analysis, but we removed this sentence due to the current results' density.

Reviewer #2 (Remarks to the Author):

The study "The Genetic Heterogeneity of Multimodal Human Brain Age" by Wen et al. modelled the brain age gap (BAG) using different brain imaging modalities (including grey matter volume, GM; white matter microstructure, WM; and functional connectivity, FC) and investigated their associated genetics. The study aims are very broad: The authors performed GWAS to identify genetic variants separately associated with BAGs derived from GM, WM and FC, calculated genetic correlations to estimate the degree of genetic overlap of BAG phenotypes with other trait and diseases, and constructed gene-drug-disease networks to identify druggable targets. They also performed Mendelian Randomisation to identify potential causal effects of risk factors (such as neurodegenerative diseases) on BAG. Finally, they calculated associations between polygenic risk for multiple clinical traits to predict BAG.

We thank the reviewer for precisely summarizing our manuscript.

First, we summarized the major changes in the revised manuscript and addressed the comments the three reviewers consistently raised. Many of these comments among reviewers converge to similar constructive suggestions.

- The term "heterogeneity" was found to be potentially misleading, as reviewers #1 and #2 commented. Consequently, we have adjusted the emphasis on the term "heterogeneity" throughout the manuscript. We also reorganized the paper and rewrote many places, guided by all three reviewers.
- To justify the brain age prediction performance as presented in our manuscript compared to those reported in prior studies (commented by all three reviewers), we have integrated a new paragraph to discuss these findings. We also conducted an additional experiment involving GM-BAG. Specifically, we employed and compared the performance (brain age prediction and also GWASs) with Support Vector Regression (SVR) on voxel-wise RAVENS maps (GM-BAG-voxel) and MUSE-ROI features (GM-BAG-ROI).
- We performed three additional sensitivity checks (on top of split-sample, sex-stratified, and non-European GWASs) to test the robustness of our main GWASs using European ancestry (suggested by all reviewers #1 and #2).
 - We used SVR to perform feature type-specific GWASs, focusing on different feature types: GM-BAG-ROI versus GM-BAG-voxel.
 - We used MUSE ROIs (GM-BAG) to perform machine learning-specific GWASs, comparing GM-BAG derived from Lasso regression vs. SVR.
 - We reran the three main GWASs using fastGWA (i.e., mixed linear models to account for cryptic population stratification) to justify that there is no substantial genomic inflation in our main GWAS using PLINK linear regression models.
- Furthermore, we calculated the genomic inflation factor (λ) and LDSC intercept (b) in the main GWASs to support further that there is no potential genomic inflation in our main GWASs using European ancestry.
- We derived the polygenic risk scores (PRS) for the GM, WM, and FC-BAG using both Plink (Clumping + Threshold approach) and a more advanced Bayesian approach (PRS-CS) (suggested by all reviewers #1, #2, and #3). Overall, PRS-CS outperforms PLINK in predicting the three BAG-PRSs in the test dataset (split2 GWAS).
- We calculated the phenotypic and genetic correlations for each pair of BAGs in Fig. 1E and Fig. 2F (suggested by reviewer #2).
- We developed a multi-organ web portal (MEDICINE: <http://labs.loni.usc.edu/medicine/>) to disseminate our findings, including showing the Manhattan & QQ plots and allowing the community to download our GWAS summary statistics.

- The keyword "Caucasian" implies racism. We changed the word "Caucasian" to "European" throughout the paper guided by this Nature article (<https://www.nature.com/articles/d41586-021-02288-x>), as the word Caucasian roots in racist taxonomies used to justify slavery.

Secondly, considering that all three reviewers have highlighted our study's significance to imaging genetics, we would like to provide a brief overview (non-exhaustive) of previous GWAS papers conducted on BAG and outline the scientific progress that our study contributes.

It's important to acknowledge that brain age is not a newly emerging biomarker; it has been extensively studied in the literature, as evidenced by pertinent literature cited by reviewer #3. What sets our study apart regarding significance and innovation lies in two aspects. Firstly, prior GWASs of brain age have often been confined to specific aspects, such as GWASs focusing solely on GM-BAG. Secondly, some studies have restricted themselves to the analyses of genome-wide associations without proceeding to undertake comprehensive post-GWAS analyses (e.g., partitioned heritability, gene-drug-disease network, Mendelian randomization, etc.) aimed at partially validating the genetic signals identified. Our study advances the field by adopting a holistic approach that addresses both these limitations, thus contributing to a more comprehensive understanding of the genetic underpinnings of brain age. We have integrated this into the Main section in the revised manuscript.

We want first to acknowledge several representative previous GWAS works on this topic – science is an incremental process built upon the foundation of prior research:

- Kaufmann et al. Nat Neurosci, 2019 performed GWAS on **GM features**, and they then estimated the SNP-based heritability of the brain age using LDSC (<https://www.nature.com/articles/s41593-019-0471-7>).
- Jonsson et al., 2019, Nature Communications: (<https://www.nature.com/articles/s41467-019-13163-9>) also performed the GWAS on **GM-BAG** derived from a deep learning model.
- Smith, S. M. et al. 2020, eLfi (<https://elifesciences.org/articles/52677>) performed GWAS on BAGs derived from multimodal brain IDPs; they primarily only focused on GWAS but did not perform comprehensive post-GWAS analyses.
- Leonardsen et al., 2023, Molecular Psy (<https://www.nature.com/articles/s41380-023-02087-y>) performed GWAS on **GM-BAG** derived from a deep learning model and then investigated the causal relationship with other traits.

This summary is by no means exhaustive.

Many aspects of this study provide an important contribution to the field of neuroimaging genetics, including the systematic use of multiple imaging modalities, state-of-the-art statistical approaches, and the use of the largest available database that would permit investigating this question. The subject of multimodal brain age and its genetic underpinnings is worthwhile and timely, and it represents an impactful finding to show reliably that different imaging modalities yield distinct brain age phenotypes with distinct biological underpinnings, that may represent druggable targets.

We appreciate the recognition of the significance of our work from the reviewer.

I think, however, that this study requires some important additions to match the quality of research that tends to get published in Nature Communications. Those additions include, for example, a more nuanced discussion of the quality of their BAG phenotypes and GWAS which is fundamentally required to judge the quality and reliability of post-GWAS investigations.

First, we appreciate that the reviewers precisely capture the three main topics covered in this study: *i*) brain age prediction using AI and machine learning, *ii*) GWAS, and *iii*) post-GWAS analyses for potential biological validation of the genetics signals. We addressed the two questions raised by the reviewer as below.

The quality of the BAG derived from ML: As also pointed out by reviewer #3, previous studies have achieved a lower MAE in their brain age predictions, particularly those employing deep learning techniques (e.g., CNN) on voxel-wise imaging scans. The rationale behind our study's utilization of low-dimensional brain IDPs rather than voxel-based imaging scans stems from several considerations:

- In comparison to prior research on brain age that utilized Region of Interest (ROI)-based or parcel-based features, such as the systematic work conducted by More et al., 2023, Neuroimage (<https://www.sciencedirect.com/science/article/pii/S1053811923000940?via%3Dihub>), our MAE is in line the existing literature or even better than their parcel-based approaches (Lines 127-138).
- Deep learning models and voxel-wise images indeed can achieve a lower MAE, such as the one from Peng et al. 2021, Medical Image Analysis (<https://www.sciencedirect.com/science/article/pii/S1361841520302358>). However, in our previous work led by Bashyam et al., 2020, Brain. (<https://academic.oup.com/brain/article/143/7/2312/5863667>), we have shown that the GM-BAG derived from a tightly fitting CNN (a lower MAE) is less discriminative (a smaller effect size) between the disease and healthy control groups compared to a moderately fitting CNN model (a higher MAE).
- Considering that Dr. Davatzikos lab primarily focuses on T1-weighted MRI data, the application of CNN to voxel-wise imaging from T1, diffusion, and resting-state functional MRI (rsfMRI) data posed a significant challenge in this short-time revision. However, to partially address this limitation, we conducted supplementary analyses by applying Support Vector Regression (SVR) to voxel-wise RAVENS maps, denoted as GM-BAG-voxel, derived from the T1-weighted MRI scans at UPENN. We then compared the MAE of this approach against the SVR applied to MUSE ROIs, denoted as GM-BAG-ROI (the features used in the primary manuscript for GM-BAG). Furthermore, we performed GWASs for GM-BAG-ROI and GM-BAG-voxel and found a notably high concordance rate of 92.43% between them (as detailed in Lines 229-235):
 - We finally found a 92.43% concordance rate of the SNPs identified in the GM-BAG GWAS using the 119 MUSE ROIs⁴⁰ (as discovery, BAG MAE=4.39 years) and voxel-wide RAVENS⁴¹ maps (as replication, P-value < 0.05/3382, BAG MAE=5.12 years) (**Supplementary eFigure 8** and **Supplementary eFile 7**). The BAGs derived from the two types of features were significantly correlated ($r=0.74$; P-value< 1×10^{-10}). The brain age prediction performance using RAVENS showed marginal overfitting, with an MAE of 4.31 years in the training/validation/test dataset and an MAE of 5.12 years in the independent test dataset.

In the revised manuscript, we have now added one paragraph regarding our brain age prediction performance vs. these previous papers in the literature (Lines 127-138):

"In the literature, other studies³⁰⁻³³ have thoroughly evaluated age prediction performance using different machine learning models and input features. More et al.³⁴ systematically compared the performance of age prediction of 128 workflows (MAE between 5.23–8.98 years) and showed that voxel-wise feature representation (MAE approximates 5-6 years) outperformed parcel-based features

(MAE approximates 6-9 years) using conventional machine learning algorithms (e.g., Lasso regression). Using deep neural networks, Peng et al.³⁰ and Leonardsen et al.³¹ reported a lower MAE (nearly 2.5 years) with voxel-wise imaging scans. However, we previously showed that a moderately fitting convolutional neural network (CNN) obtained significantly higher differentiation (a larger effect size) than a tightly fitting CNN (a lower MAE) between the disease and health groups³⁵. To summarize, our study's brain age prediction performance aligns with those reported in the existing literature, considering the utilization of low-dimensional hand-crafted IDPs and conventional machine learning algorithms³⁴."

For the quality of the GWAS:

- We performed in total **six** sensitivity checks to check the robustness of our main GWASs: *i*) split-sample GWASs, *ii*) sex-stratified GWASs, *iii*) non-European GWAS, *iv*) fastGWA for mixed linear regression models, *v*) machine learning-specific GWASs (Lasso regression GM-BAG vs. SVR GM-BAG), and *vi*) feature type-specific GWASs (MUSE ROI SVR vs. voxel RAVENS SVR GM-BAG). The main GWAS findings are consistent in several analyses, such as split-sample, sex-stratified, fastGWA, machine learning-specific, and feature type-specific GWASs, but generalizability is limited to non-European ancestries. The updated results are presented in the revised manuscript in lines **196-239**.
- We also reported/added the genomic inflation factor (λ) and LDSC intercept (*b*) (Lines **166-171**), individual Manhattan & QQ plots in the **Supplementary eFigure2-7**.
- All these figures and GWAS summary statistics are publicly available at our MEDICINE multi-organ web portal: <http://labs.loni.usc.edu/medicine/>.

Furthermore, I suggest that the methods section requires more detail to make results reproducible, and that the manuscript requires a thorough discussion of methods limitations. On a personal note, I (JW) am sincerely grateful for the reviewer's insights concerning the crucial aspect of reproducibility in the intersection of AI and medicine. Throughout my career, I have adhered to the principles of reproducible machine learning. Notably, one of the accomplishments in my career is closely intertwined with this very theme, and I regard it as one of the most impactful works in my career:

<https://www.sciencedirect.com/science/article/pii/S1361841520300591>

To address the reproducibility in the revised manuscript, we have made substantial efforts to tackle this issue:

- We have made our code openly accessible. In the "Code Availability" section, we have furnished links to the software tools we developed for brain age prediction within this study (<https://anbai106.github.io/mlni/>) and provided references to all the open-source packages developed by other teams. This ensures readers can replicate our findings using our manuscript and these software resources.
- We have furnished comprehensive methodological descriptions, particularly on computational genomics techniques, including using 2SampleMR for Mendelian randomization. We have provided a thorough account of our exposure variable selection process, elucidated how the five distinct Mendelian randomization (MR) methods handle the underlying assumptions in MR, and expounded upon our rigorous sensitivity checks designed to scrutinize the robustness of our findings (**Method 4G**, Lines **731-768**). Overall, we made substantial efforts to provide details in the Method sections (All changes are in **yellow-colored** text in the revised manuscript).

- Transparent in the selection of GWAS summary statistics for LDSC and MR analyses, and report the sample sizes, P-value, and effect sizes throughout the paper and supplementary files (**Method 4D** and **4G**).
- We have publicly made all our GWAS summaries accessible through the MEDICINE multi-organ web portal (<http://labs.loni.usc.edu/medicine/>). Researchers are welcome to use this data to verify our GWAS results and conduct independent analyses.
- Finally, in the revised manuscript, we discussed the limitations regarding methodologies used in the current study (Lines **540-554**).

I have outlined more specific suggestions below.

Analyses supporting the quality of the BAG phenotypes and GWAS

- Phenotypic correlations: Considering it seems that the heterogeneity of different BAGs is a major focus of this study, the authors do not seem to present empirical evidence that there is indeed heterogeneity between the BAG phenotypes.

We appreciate the reviewer's input on this matter. As pointed out by reviewer #1, the term "heterogeneity" can be misleading. We have taken steps to de-emphasize it in the revised manuscript. Instead, we have emphasized that BAG derived from various imaging modalities might capture distinct and/or common neurobiological processes associated with human brain aging. Consequently, exploring the genetic foundations of these different facets may yield novel insights (Lines **70-81**):

“It is crucial to holistically identify the genetic factors associated with multimodal BAGs (GM, WM, and FC-BAG), where each BAG reflects distinct and/or similar neurobiological facets of human brain aging. Furthermore, dissecting the genetic architecture of human brain aging may determine their causal implications, which is essential for developing gene-inspired therapeutic interventions. Finally, numerous risk or protective lifestyle factors and neurobiological processes may also exert independent, synergistic, antagonistic, sequential, or differential influences on human brain health. Therefore, a holistic investigation of multimodal BAGs is urgent to fully capture the genetics of human brain aging, including the genetic correlation, gene-drug disease network, and potential causality. In this study, we postulate that AI-derived GM, WM, and FC-BAG can serve as robust, complementary endophenotypes²³ – close to the underlying etiology – for precise quantification of human brain health.”

In one sentence ("Fifth, we found considerable heterogeneity in the expression of GM, WM, and FC-BAG in this independent dataset"), they point the reader to Fig.1E which is challenging to interpret – in a 3-dimensional plot on a 2-dimensional papers it is near impossible to identify the true location of a dot – and they do not quantify the extent to which BAG phenotypes correlate.

We concur with this feedback and acknowledge that **Figure 1E** does not effectively convey crucial information. Originally, we intended to use this figure to represent human brain aging across diverse imaging modalities digitally. However, in the revised manuscript, we have replaced this figure with a new one, which illustrates the pairwise phenotypic correlations between BAGs (**Fig. 1E**).

Please consider reporting Pearson's correlations among the three BAG phenotypes used for subsequent GWAS investigations. If there truly is heterogeneity between BAGs derived from different imaging modalities, the correlations among BAGs should be moderate at best. This would represent an important part of the study to show genetic analyses are indeed investigating distinct phenotypes, which is especially important because BAGs are residual scores that capture measurement error in addition to true individual differences. Would the

authors also be able to more clearly visualise the heterogeneity (Fig.1E) and avoid a three-dimensional plot?

In the revised manuscript, we have substituted this with a correlation plot illustrating the pairwise Pearson's r coefficients between GM, WM, and FC-BAG (as shown in Fig. 1E). Furthermore, we have shifted our primary focus when presenting the key findings of the paper, as emphasizing heterogeneity is not suitable. We also added the results in Lines 139-143 in the revised manuscript:

“Finally, we calculated the phenotypic correlation (p_c) between GM, WM, and FC-BAG using Pearson's correlation coefficient. GM-BAG and WM-BAG showed the highest positive correlation ($p_c=0.38$; P-value $<1 \times 10^{-10}$; $N=30,733$); GM-BAG ($p_c=0.09$; P-value $<1 \times 10^{-10}$; $N=30,660$) and WM-BAG ($p_c=0.10$; P-value $<1 \times 10^{-10}$; $N=31,574$) showed weak correlations with FC-BAG (Fig. 1E).”

- Genetic correlations: Similarly, the different BAGs should only be moderately genetically correlated if they are truly distinct phenotypes. I suggest that reporting genetic correlations among BAGs is crucial to be able to interpret individual GWAS hits (as done in "Genomic loci associated with multimodal BAG show different phenome-wide associations"). Only if different BAGs are imperfectly genetically correlated, we can be confident that genetic follow-up analyses are truly indicative of distinct biological pathways underlying the heterogeneity of different BAGs.

This is an excellent suggestion. In the revised manuscript, we removed Fig. 2F – the heritability estimates were shown in Lines 172-175:

“The three BAGs were significantly heritable (P-value $<1 \times 10^{-10}$) after adjusting for multiple comparisons using the Bonferroni method using the genome-wide complex trait analysis (GCTA) software³⁷. GM-BAG showed the highest SNP-based heritability ($h^2=0.47 \pm 0.02$), followed by WM-BAG ($h^2=0.46 \pm 0.02$) and FC-BAG ($h^2=0.11 \pm 0.02$).”

Instead, we showed a correlation plot illustrating the pairwise genetic correlations (g_c) between GM, WM, and FC-BAG (Fig. 2F). We added the results in Lines 187-194:

“Finally, we calculated the genetic correlation (g_c) between the GM, WM, and FC-BAG using the LDSC software. GM-BAG and WM-BAG showed the highest positive correlation ($g_c=0.49$; P-value $<1 \times 10^{-10}$); GM-BAG ($g_c=0.20$; P-value=0.025) and WM-BAG ($g_c=0.29$; P-value=0.005) showed weak correlations with FC-BAG (Fig. 2F). The genetic correlations largely mirror the phenotypic correlations, supporting the long-standing Cheverud's Conjecture³⁹. We also verified that these genetic correlations exhibited consistency between the two random splits (split1 and split2: $15,778 < N < 16,008$), sharing a similar age and sex distribution (Supplementary eFigure 2).”

We also have an interesting observation. The results of genetic correlations correspond to the phenotypic correlations, supporting the long-standing Cheverud's Conjecture.

- PRS prediction: It is unclear to me why the authors chose to model genetic propensity (i.e., PRS) towards 16 other traits, instead of modelling genetic propensity towards BAGs from the GWAS they created as part of this manuscript. In my experience, studies that calculate GWAS summary statistics use PRS predictions to showcase the predictive ability (and thereby quality and validity) of their GWAS summary statistics. If a PRS derived from GWAS predicts its own base phenotype well (usually indicated by R^2), we take it as evidence that the GWAS indeed captures genetic propensity towards this trait. Would the authors consider reporting R^2 estimates of BAG PRSs predicting their base BAG as outcome? Could they also please explain why they chose to predict BAG phenotypes using PRSs of other traits, considering this study seem to focus on the genetic architecture of BAGs?

We really appreciate these constructive comments. All three reviewers raised similar questions/comments on the PRS calculation.

We removed these old results from the revised manuscript and performed a new PRS analysis. Specifically, we derived the PRSs using both Plink and PRC-CS (a Bayesian method) for GM, WM, and FC-BAG using the two-split GWASs. That is, we used split1 GWAS as training/base data for the weights and split2 GWAS as testing/target data to calculate the incremental R^2 to predict the phenotype of themselves (GM, WM, and FC-BAG). We compared the predictive power of the three BAG-PSCs and found that PRS-CS-derived PRSs are more predictive than those from the PLINK C+T approach (**Fig. 4D** and lines **340-350**):

"We derived the PRS for GM, WM, and FC-BAG using the conventional C+T (clumping plus P-value threshold) approach⁵⁵ via PLINK and a Bayesian method via PRS-CS⁵⁶ (**Method 4H**).

We found that the GM, WM, and FC-BAG-PRS derived from PRS-CS significantly predicted the phenotypic BAGs in the test data (split2 GWAS, $15,697 < N < 15,940$), with an incremental R^2 of 2.17%, 1.85%, and 0.19%, respectively (**Fig. 4D**). Compared to the PRS derived from PRS-CS, the PLINK approach achieved a lower incremental R^2 of 0.81%, 0.45%, and 0.14% for GM, WM, and FC-BAG, respectively (**Supplementary eFigure 9**). Overall, the predictive capacity of PRS is moderate, in line with our earlier discoveries involving raw imaging-derived phenotypes, as demonstrated in Zhao et al.¹³, where PRSs developed for seven selective brain regions were able to explain roughly 1.18% to 3.93% of the phenotypic variance associated with these traits."

- Heritability: Would the authors consider stating heritability estimates in the main manuscript (page 6)? Heritability estimates are important indicators of signal-to-noise captured by the GWAS (and thereby another indication of quality), and the reader should know that they are substantial here (Fig.2F).

This is an excellent suggestion. We have reported the SNP-based h^2 estimates using the GCTA software in the main text (Lines **172-175**).

"The three BAGs were significantly heritable ($P\text{-value} < 1 \times 10^{-10}$) after adjusting for multiple comparisons using the Bonferroni method using the genome-wide complex trait analysis (GCTA) software³⁷. GM-BAG showed the highest SNP-based heritability ($h^2 = 0.47 \pm 0.02$), followed by WM-BAG ($h^2 = 0.46 \pm 0.02$) and FC-BAG ($h^2 = 0.11 \pm 0.02$)."

Suggestion to include more general genome-wide trends in the discussion of the manuscript (rather than mostly focusing individual GWAS hits only)

- The manuscripts invests a lot of space into discussing the specific SNP associations yielded by different BAGs, and the authors end the section ("Genomic loci associated with multimodal BAG show different phenome-wide associations") by stating those associations are indicative of genetic overlap with other traits.

We agree with the reviewer on this comment. We initially intended to include summary paragraphs at the end of each result section to facilitate the transition to the subsequent section. We have opted to remove this paragraph in the revised manuscript.

Whether there is genetic overlap can indeed be directly tested using genetic correlations, and I would argue that the highly polygenic nature of BAGs does not permit interpreting distinct individual GWAS hits indicative of genetic overlap.

We agree with this comment – Indeed, in the next section, we explicitly performed genetic correlation via the LDSC software.

I would like to suggest the authors focus their discussion less on individual GWAS hits because significant hits are directly dependent upon GWAS sample size, and their significance is sensitive towards sampling variance. In the grand scheme of GWAS, the UKB

sample size used here is relatively small and it is not reasonable to expect that this analysis would pick up on all relevant associations (and the chances of spurious associations are high – winners curse).

We fully concur with this observation. Independent replication of the genetic signals identified in the discovery phase, conducted in the UK Biobank (UKBB), is crucial. However, to the best of our knowledge, we must acknowledge that no other extensive imaging genetics consortium encompasses multimodal imaging data along with comparable genetic sequences. Instead, we conducted six sensitivity checks within the UKBB dataset to bolster the reliability of our findings (Lines 196-239).

To respond to this feedback, we have relocated the discussion concerning the exemplary genomic locus associated with GM, WM, and FC-BAG (**Fig. 2D-E**) to **Supplementary eText 1** and removed the discussion of individual SNPs in the main GWAS section. Within this supplementary section, we have delved into the implications of each individual SNP/gene, referencing relevant literature.

Instead, I think it would greatly improve the manuscript to give more space to the discussion of genome-wide trends and BAGs genetic overlap among one another and with other traits. Such genome-wide trends tend to be more reliable than individual SNP associations. Could the authors please consider a more measured discussion of genome-wide trends alongside individual GWAS hits?

We appreciate the reviewer for this constructive suggestion. In the Discussion section, we have provided a comprehensive overview of the genetic signal trends for each BAG, their genetic correlations with other diseases, their causal relationship with other traits, and the overall genetic signals across the three BAGs.

- Genetic architecture of GM-BAG (line 449)
- Genetic architecture of WM-BAG (line 481)
- Genetic architecture of FC-BAG (line 501)

- In a related fashion, the Sensitivity analysis in page 10 (Section Sensitivity analyses for the genome-wide associations in split-sample experiments, sex-stratified experiments, and non-Caucasian populations) must acknowledge that sample size is much smaller than in the original GWAS which dictates significance findings.

In the revised manuscript, we indicated the sample sizes (N) for all six sensitivity checks.

Instead, genetic correlations among original, sex-specific, and split-sample GWAS would paint a broader picture on whether primary GWAS results generally converge and are robust. Also please indicate the sample sizes and effect sizes in this results section for better readability and interpretability.

These are really helpful comments to improve the reliability of our analyses. We have carefully added the sample sizes (N) for all analyses in the Results section.

To demonstrate the robustness of our genetic correlation results, we performed one additional analysis by computing the genetic correlation using the split1 and split2 GWAS vs. using the full sample-sized GWAS (**Fig. 2F**) between each pair of BAGs.

The results are shown in the **Supplementary eFigure 2:**

eFigure 2: Genetic correlation (g_c) between the GM, WM, and FC-BAG using the LDSC software in the split-sample analyses

A) Genetic correlation using the full samples. **B)** Genetic correlation using the split1 sample. **C)** Genetic correlation using the split1 sample.

As we can see here, the genetic correlations have a similar trend between the two splits (B and C) compared to the full sample size (A) results.

In the revised manuscript, we added on sentence in Line 191-194:

“We also verified that these genetic correlations exhibited consistency between the two random splits (split1 and spit2: $15,778 < N < 16,008$), sharing a similar age and sex distribution (Supplementary eFigure 2).”

Discussion is lacking a summary and limitation section

- The discussion section is lacking a strong introductory paragraph summarising the key trends found in this study. This may be a good opportunity to summarise evidence the authors may have for the fact that multimodal BAGs truly yield distinct phenotypes (i.e., phenotypic correlations between BAGs), and are associated with distinct genetic architecture (e.g., genetic correlations among BAGs, distinct GWAS hits & distinct druggable targets). Such a summary paragraph could then lead into a more nuanced discussion of the specific biological pathways.

We thank the reviewer for this helpful suggestion. We have now rewritten the first paragraph of the Discussion section guided by this comment. As mentioned in prior comments, we have de-emphasized the notion of heterogeneity in this study and emphasized that multimodal BAG analysis may reveal shared genetic elements and distinctive characteristics.

- In my opinion it is imperative to include a limitation section in the discussion, especially because the methods used in this manuscript are not trivial and their results can only be interpreted with certain caveats. For example, LDSC and MR come with many assumptions, brain age is a residual score that captures measurement error, cross-sectional data is limited by cohort effects and we don't know how accurately brain age captures real longitudinal changes, and many more.

We have now added a Limitation section (Lines 540-556), as suggested by all three reviewers. We have discussed all the limitations the three reviewers pointed out and provided perspectives for future research avenues that can potentially address these limitations:

“This study has several limitations. We can employ deep learning on voxel-wise imaging scans to enhance brain age prediction performance. Nevertheless, it warrants additional exploration to determine whether the resulting reduction in MAE translates into more robust genome-wide associations, as our previous work has demonstrated that BAGs derived from a CNN with a lower MAE did not exhibit heightened sensitivity to disease effects such as AD³⁵. Second, the generalization ability of the GWAS findings to non-European ancestry is limited, potentially due to small sample sizes and cryptic population stratification. Future investigations can be expanded to encompass a broader spectrum of underrepresented ethnic groups, diverse disease populations, and various age ranges spanning the entire lifespan. This expansion can be facilitated by leveraging the resources of large-scale brain imaging genetic consortia like ADNI⁸⁰, focused on Alzheimer's disease, and ABCD⁸¹, which centers on brain development during adolescence. Third, it's important to exercise caution when interpreting the results of this study due to the various assumptions associated with the statistical methods employed, including LDSC and MR. Lastly, it's worth noting that brain age represents a residual score encompassing measurement error. A recent study⁸² has underscored the significance of incorporating longitudinal data when calculating brain age. Future research should be conducted once the longitudinal scans from the UK Biobank become accessible to explore this impact on GWASs.”

- Would the authors consider adding a broad conclusion section?

We have added a new section (Outlook) (Lines: 559-564):

“In summary, our multimodal BAG GWASs provide evidence that the aging process of the human brain is a complex biological phenomenon intertwined with several organ systems and chronic diseases. We digitized the human brain from multimodal imaging and captured a complete genetic landscape of human brain aging. This opens new avenues for drug repurposing/repositioning and aids in identifying modifiable protective and risk factors that can ameliorate human brain health.”

Methods are lacking details to allow future studies to replicate results

- Page 29, Method 1: It is not transparent how UKB participants were included in the study (e.g., availability of MRI data?).

We have now added one sentence to state this (Line: 569-571) explicitly:

“The current study focused on participants from the imaging-genomics population who underwent both an MRI scan and genome sequencing (genotype array data and the imputed genotype data) under application number 35148.”

The authors state that participants with certain clinical diagnoses were excluded, but do not list the criteria (i.e., the use of "etc." prohibits replication). UKB field IDs and the sample size of exclusions and inclusions are required to allow future studies to replicate findings.

We have now added the UKBB field ID and sample sizes for all these disease diagnoses (Line: 583-586):

“As UKBB is a general population, we explicitly excluded participants with common brain diseases, including mental and behavioral disorders (ICD-10 code: F; $N=2678$) and diseases linked to the central nervous system (ICD-10 code: G group; $N=3336$).”

Please search the manuscript for more instances of etc. and specify explicitly what was done (e.g., page 35).

We have now added detailed information throughout the manuscript to address this comment. For example, we referred to the original paper for the 53 general functional categories of genetic variants if the readers want to read it in detail. In addition, the exact 53 categories are also presented in **Supplementary eTable 5** (lines: 716-718):

“The 53 functional categories are not specific to any cell type, including coding, UTR, promoter, and intronic regions. Details of the 53 categories are described elsewhere⁵⁰ and are also presented in **Supplementary eTable 5A**.”

To make the Methods more accessible, please indicate the sample size of the independent test dataset, as well as Ns available for imaging phenotypes.

In the revised manuscript, we now explicitly show the descriptive statistics (e.g., sample size, effect size, and p-values) to maximize the transparency of the analyses. For example, at the beginning of the Result section (Lines 97-100):

“To this end, we employed a nested cross-validation (CV) procedure in the training/validation/test dataset ($N=4000$); an independent test dataset ($N=38,089$)^{26,27} was held out – unseen until we finalized the models using only the training/validation/test dataset (**Method 1**).”

- Page 30, Method 2: I assume that the outcome variable here was age, and the model was trained using multiple imaging variables as predictors. This may be evident to a brain age researcher but needs to be made explicit. This section also requires more explicit details on how the models were trained: model assumptions and functionality, software packages, input parameters. Please explain what a ReLu layer is.

As the reviewer suggested, we moved the brain age prediction method to **Method 3** after the image preprocessing method (**Method 2**) in the revised manuscript. We added all the detailed information in **Method 3** on how we trained the brain age prediction model, such as the cross-validation procedure, hyperparameter tuning, etc.

All software is open-sourced packages. Specifically, for the software package to derive the three BAGs, we have included this in the Code availability section (Lines 789-799):

The software and resources used in this study are all publicly available:

- MLNI: <https://anbai106.github.io/mlni/>, brain age prediction
- MEDICINE: <https://labs.loni.usc.edu/medicine>, knowledge portal for dissemination
- MUSE: <https://www.med.upenn.edu/sbia/muse.html>, image preprocessing for GM-IDP
- PLINK: <https://www.cog-genomics.org/plink/>, GWAS and PRS
- FUMA: <https://fuma.ctglab.nl/>, gene mapping, genomic locus annotation
- GCTA: <https://yanglab.westlake.edu.cn/software/gcta/#Overview>, heritability estimates, and fastGWA
- LDSC: <https://github.com/bulik/ldsc>, genetic correlation, partitioned heritability
- TwoSampleMR: <https://mrcieu.github.io/TwoSampleMR/index.html>, MR
- PRS-CS: <https://github.com/getian107/PRSs>, PRS

- Page 31, Method 3: Could the authors please explain why Image processing is positioned in the manuscript after the machine learning models. I assume the imaging variables were processed prior to brain age predictions serving as predictor variables. Please consider swapping the sections.

This is a very good suggestion. We have moved the imaging preprocessing pipelines (**Method 3**) before the Machine learning section (**Method 2**).

Could the authors also please explain why they processed the variables themselves rather than using IDPs as provided by UKB? One may argue it would benefit comparability with other brain age studies to use preprocessed UKB IDPs.

To be clear at this point, at the University of Pennsylvania, Dr. Davatzikos team has locally processed the T1-weighted MRI to obtain the IDP (119 MUSE ROIs) to derive the GM-BAG. For IDPs derived from diffusion MRI and functional MRI, we directly downloaded the data from the UKBB website, as these MRI modalities are not the lab's main focus.

We acknowledge that IDP derived from different software, brain atlas, may contribute to the variance in brain age prediction. A recent study systematically investigated this. More et al. (<https://www.sciencedirect.com/science/article/pii/S1053811923000940>, Neuroimage) systematically compared the performance of age prediction of 128 workflows (MAE between 5.23–8.98 years). Previously, we have also reproducibly compared the AD classification accuracy from different imaging pipelines and brain atlases (<https://www.sciencedirect.com/science/article/pii/S1053811918307407>, Neuroimage). Lastly, we also compared the MUSE pipeline with Freesurfer in segmentation tasks. We found that MUSE obtains more consistent segmentations across scanners compared to Freesurfer, particularly in the deep structures (<https://pubmed.ncbi.nlm.nih.gov/32860881/>, Neuroimage).

To directly assess this impact on brain age prediction performance, we did one additional experiment to compare the brain age prediction performance in GM-BAG between the 119 MUSE ROIs vs. voxel-wise RAVENS maps (<https://www.sciencedirect.com/science/article/pii/S1053811901909371>) using SVR. The results are updated in the revised manuscript both in brain age prediction performance and GWAS (feature type-specific GWAS sensitivity checks) (Lines 229-235):

“We finally found a 92.43% concordance rate of the SNPs identified in the GM-BAG GWAS using the 119 MUSE ROIs⁴⁰ (as discovery, BAG MAE=4.39 years) and voxel-wise RAVENS⁴¹ maps (as replication, P-value < 0.05/3382, BAG MAE=5.12 years) (**Supplementary eFigure 8 and Supplementary eFile 7**). The BAGs derived from the two types of features were significantly correlated ($r=0.74$; P-value< 1×10^{-10}). The brain age prediction performance using RAVENS showed marginal overfitting, with an MAE of 4.31 years in the training/validation/test dataset and an MAE of 5.12 years in the independent test dataset.”

- Page 32, Diffusion MRI processing, "Finally, to overcome the overfitting problem due to the high correlations between FA, MD, ODI, and NDI within the same WM tract, we used only the 48 FA WM-IDP to fit the models for brain age prediction": It is unclear which 48 variables were selected here, and why.

We apologize for the confusion.

The 48 FA values included: UKBB has provided the FA, MD, ODI, and NDI metrics for the 48 white matter tracts from the commonly used WM brain atlas ("ICBM-DTI-81 white-matter labels" atlas). This atlas is also included in the FSL software:

https://web.mit.edu/fsl_v5.0.10/fsl/doc/wiki/Atlases.html.

[redacted]

References:

- Mori et al., MRI Atlas of Human White Matter. Elsevier, Amsterdam, The Netherlands (2005)
- Wakana et al., Reproducibility of quantitative tractography methods applied to cerebral white matter. NeuroImage 36:630-644 (2007)
- Hua et al., Tract probability maps in stereotaxic spaces: analysis of white matter anatomy and tract-specific quantification. NeuroImage, 39(1):336-347 (2008)

This is also explained on the UKBB website (Category code: 134 & 135)
<https://biobank.ctsu.ox.ac.uk/showcase/label.cgi?id=107>

In the revised manuscript, we made this clear (Lines 614-625):

“(B): Diffusion MRI processing: UKBB has processed diffusion MRI (dMRI) data and released several WM tract-based metrics for the Diffusion Tensor Imaging (DTI) model (single-shell dMRI) and Neurite Orientation Dispersion and Density Imaging (NODDI⁸⁶) model (multi-shell dMRI). The Eddy⁸⁷ tool corrected raw images for eddy currents, head motion, and outlier slices. The mean values of FA, MD, ODI, and NDI were extracted from the 48 WM tracts of the "ICBM-DTI-81 white-matter labels" atlas⁸⁸, resulting in 192 WM-IDP (category code:134). In addition, a tract-skeleton (TBSS)⁸⁴ and probabilistic tractography analysis⁸⁹ were employed to derive weighted-mean measures within the 27 major WM tracts, referred to as the 108 TBSS WM-IDP (category code: 135). Finally, since we observed overfitting – an increase of MAEs from the cross-validated test results to the independent test results – when incorporating features from FA, MD, ODI, and NDI (as detailed in **Supplementary eTable 1A**), we chose to use only the 48 FA WM-IDPs to train the models for generating GM-BAG.”

Why the 48 variables are selected: We observed clear overfitting when combining the 48 white matter features from FA, MD, ODI, and NDI, as evidenced by the results in **Supplementary eTable 1A**. For example, with Lasso regression, the cross-validated test result (CV test) obtained an MAE of 4.94 for all 192 FA/MD/ODI/NDI metrics, but the independent test result (Ind. test) obtained an MAE of 1.66. This overfitting did not happen while using only the 48 FA features. This is the reason why we only used the 48 FA metrics to derive the WM-BAG in the main manuscripts.

To clarify this, we have now revised these sentences:

“Finally, since we observed overfitting – an increase of MAEs from the cross-validated test results to the independent test results – when incorporating features from FA, MD, ODI, and NDI (as detailed in **Supplementary eTable 1A**), we chose to use only the 48 FA WM-IDPs to train the models for generating GM-BAG.”

- Page 32, Resting-state FC: "four components were removed due to artifactual components": It is unclear how an artifactual component was identified.

We apologize for the confusion.

As stated above, we did not process the rsfMRI locally. Instead, we directly download the rsfMRI metrics from the UKBB website. The features that we finally used are the functional connectivity derived from spatial-ICA by the UKBB imaging core team (led by Dr. Stephen Smith, Dr. Karla L Miller, and many others). The detailed image preprocessing is stated in the official document of UKBB:

https://biobank.ctsu.ox.ac.uk/crystal/crystal/docs/brain_mri.pdf, as well as on the showcase website: <https://biobank.ctsu.ox.ac.uk/showcase/field.cgi?id=25750>.

The four components that were regarded as artifactual components are decided here: "Anygroup-ICA components that are clearly identifiable as artefactual (i.e., not neuronally driven) are discarded during the network modelling described below; a text file is supplied with the group-ICA maps, listing the group-ICA components kept in the final network modelling." Here is an example of the final 210 FC-IDP provided by the UKBB Showcase web page: <https://biobank.ctsu.ox.ac.uk/showcase/refer.cgi?id=523>.

In the revised manuscripts, we provided detailed information and cited reference resources/papers to make this clear (Lines 627-633):

“For FC-IDP, we used the 21×21 resting-state functional connectivity (full correlation) matrices (data-field code: 25750) from UKBB^{90,91}. UKBB processed rsfMRI data and released 25 whole-brain spatial independent component analysis (ICA)-derived components⁹²; four components

were removed due to artifactual components. This resulted in 210 FC-IDP quantifying pairwise correlations of the ICA-derived components. Details of dMRI and rsfMRI processing are documented here: https://biobank.ctsuo.ox.ac.uk/crystal/crystal/docs/brain_mri.pdf.”

- Page 32, genetic analyses: The authors do not state specifically which quality steps were performed to clean the genetic data. For example, did they exclude/ include genetic variants based on minor allele frequency, missingness, imputation quality? Were participants excluded due to missingness, or sex checks?

We sincerely apologize for this. In the original version, we cited our first imaging genetics paper for the genetic quality check pipeline (Preliminarily accepted at another journal: <https://www.medrxiv.org/content/10.1101/2022.07.20.22277727v2>). We admit that it is our responsibility to explicitly state these details in any new publications, which is critical for reviewers to scrutinize the rigors of our GWAS. In the revised manuscript, we cited the paper and also summarized the QC pipeline (Lines 656-665):

“We summarize our genetic QC pipeline as below. First, we excluded related individuals (up to 2nd-degree) from the complete UKBB sample using the KING software for family relationship inference⁹⁵. We then removed duplicated variants from all 22 autosomal chromosomes. Individuals whose genetically identified sex did not match their self-acknowledged sex were removed. Other excluding criteria were: i) individuals with more than 3% of missing genotypes; ii) variants with minor allele frequency (MAF) of less than 1%; iii) variants with larger than 3% missing genotyping rate; iv) variants that failed the Hardy-Weinberg test at 1×10^{-10} . To adjust for population stratification⁹⁶, we derived the first 40 genetic principle components (PC) using the FlashPCA software⁹⁷. Details of the genetic quality check protocol are described elsewhere^{94,98}.”

In this regard, we sincerely appreciate the reviewer and the editor allowing us to clarify this matter in this revision.

What was the final amount of participants and SNPs?

We have clarified this in the revised manuscript (Lines 652-656):

“Imputed genotype data were quality-checked for downstream analyses. Our quality check pipeline (see below) resulted in 33,541 European ancestry participants and 8,469,833 SNPs. After merging with the multimodal MRI populations, we included 31,557 European participants for GM-BAG, 31,749 participants for WM-BAG, and 32,017 participants for FC-BAG GWAS. Details of the protocol are described elsewhere^{15,94}.”

Please justify the choice of the PLINK software, instead of another more sophisticated software such as REGENIE (<https://rgcgithub.github.io/regenie/>).

Thank the reviewer for this comment. We agree that more sophisticated software (such as REGENIE) may provide additional power for identifying more genetic signals and battling the cryptic population stratification that cannot be addressed by simply including genetic PCs and removing related individuals.

In the revised manuscript, we performed an additional sensitivity check analysis (Plink vs. fastFGWA; linear regression vs. linear mixed model). fastGWA (<https://www.nature.com/articles/s41588-019-0530-8>) was developed in GCTA and had a similar approach as the REGENIE linear mixed model (via a genetic-relatedness matrix). In this analysis (main GWAS for European ancestry), we found that (Lines 216-223):

“A mixed linear model employed via fastGWA (as replication, $31,557 < N < 32,017$) obtained 100% concordance rates for GM, WM, and FC-BAG compared to GWAS using PLINK linear regression (Supplementary eFile 5). The genetic loci, genomic inflation factor (λ), and the LDSC intercepts for GM, WM, and FC-BAG were similar between the PLINK and fastGWA analyses (Supplementary eFigure 6). For future GWASs, there is potential to enhance statistical power by

employing mixed linear models that do not necessitate the exclusion of related individuals, along with additional strategies to account for potential cryptic population stratification.”

We agree that the linear mixed model should be used because of the increased statistical power and further correction for population stratification; we will apply the mixed models in our future analyses using REGENIE and other sophisticated software in the field.

It is also unclear whether the GWAS were conducted in the whole sample, or whether it was performed in the training or validation subsets. Please include sample sizes throughout the manuscript, including the abstract.

We apologize for this confusion. The GWAS was done on all populations after we evaluated the brain age prediction task on the training (training/validation/test for nested CV; N=4000) and the independent test data). In Method 4A for GWAS, we stated for the covariates (i.e., training/validation/test or independent test as a covariate) included in the model (Lines 667-673):

“For GWAS, we ran a linear regression using Plink⁹⁹ for GM, WM, and FC-BAG, controlling for confounders of age, dataset status (training/validation/test or independent test dataset), age x squared, sex, age x sex interaction, age-squared x sex interaction, total intracranial volume, the brain position in the scanner (lateral, transverse, and longitudinal), and the first 40 genetic principal components. The inclusion of these covariates is guided by pioneer neuroimaging GWAS conducted by Zhao et al¹³. and Elliot et al.¹¹

In the revised manuscript (especially the Result section), we provided the sample size information whenever applicable.

- Page 35, two-sample MR: The authors do not name the five different MR methods they claim to have used, and they also do not name the 7 exposure variables included in the analysis.

We have added the names of the five MR methods and their references (Lines 734-738), if available, and the seven exposure variables (the procedure to unbiasedly choose them; see below). (Lines 762-763).

These details were also available in **Supplementary eFile 9** for the five MR methods' results and **Supplementary eTable 6** for the seven exposure variables used in our study.

MR assumptions are not explicitly stated, and it is not stated how the different methods address MR assumptions.

In the revised manuscript, we have now explicitly stated the main assumptions of Mendelian randomization and how the five different methods used in our study overcome these limitations (Lines 738-749):

“MR relies on a set of crucial assumptions to ensure the validity of its results. These assumptions include the requirement that the chosen genetic instrument exhibits a strong association with the exposure of interest while remaining free from direct associations with confounding factors that could influence the outcome. Additionally, the genetic variant used in MR should be independently allocated during conception and inheritance, guaranteeing its autonomy from potential confounders. Furthermore, this genetic instrument must affect the outcome solely through the exposure of interest without directly impacting alternative pathways that could influence the outcome (no horizontal pleiotropy). The five MR methods handle pleiotropy and instrument validity assumptions differently, offering various degrees of robustness to violations. For example, MR Egger provides a method to estimate and correct for pleiotropy, making it robust in the presence of horizontal pleiotropy. However, it assumes that directional pleiotropy is the only form of pleiotropy present.”

Please consider reporting MR findings according to standard guidelines as outlined here: Strengthening the reporting of observational studies in epidemiology using mendelian randomisation (STROBE-MR): explanation and elaboration | The BMJ

We appreciate the reviewer for this reference to improve the transparency and robustness of the interpretation of our MR results. Besides the abovementioned details for the methodologies, we also provided additional necessary information of MR:

- **Unbiased exposure variable selection procedure (Lines: 750-768):**

"To ensure an unbiased selection of exposure variables, we followed a systematic procedure guided by the STROBE-MR Statement¹⁰⁸. We pre-selected exposure variables across various categories based on our phenome-wide association query. These variables encompassed neurodegenerative diseases (e.g., AD), liver biomarkers (e.g., AST), cardiovascular diseases (e.g., the triglyceride-to-lipid ratio in VLDL), and lifestyle-related risk factors (e.g., BMI). Subsequently, we conducted an automated query for these traits in the IEU GWAS database¹⁰⁹, which provides curated GWAS summary statistics suitable for MR, using the *available_outcomes()* function. We ensured the selected studies used European ancestry populations and shared the same genome build as our GWAS (HG19/GRCh37). Additionally, we manually examined the selected studies to exclude any GWAS summary statistics overlapping with UK Biobank populations to prevent bias stemming from sample overlap¹¹⁰. This process yielded a set of seven exposure variables, comprising AD, breast cancer, type 2 diabetes, renin level, triglyceride-to-lipid ratio, aspartate aminotransferase (AST), and BMI. The details of the selected studies for the instrumental variables (IVs) are provided in **Supplementary eTable 6**.

We performed several sensitivity analyses. First, a heterogeneity test was performed to check for violating the IV assumptions. Horizontal pleiotropy was estimated to navigate the violation of the IV's exclusivity assumption⁶³ using a funnel plot, single-SNP MR approaches, and MR Egger estimator¹⁰⁶. Moreover, the leave-one-out analysis excluded one instrument (SNP) at a time and assessed the sensitivity of the results to individual SNP."

- **Statistical presentation:** Furthermore, in our presentation of MR statistics, we included the odds ratio (OR), P-values, and the count of instrumental variables (IVs). Comprehensive results are available in **Supplementary eFile 9**.
- **Sensitivity checks:** We created a new section (Line 390) for sensitivity assessments for one of the causal relationships uncovered in the MR analyses. These paragraphs exemplify our careful approach to interpreting these findings cautiously.

We have appropriately referenced the specified paper and adhered to the guidelines for reporting our Mendelian randomization findings. Additionally, we have discussed the methodological limitations of the MR in the Limitation section of the revised manuscript (Lines 551-552):

"Third, it's important to exercise caution when interpreting the results of this study due to the various assumptions associated with the statistical methods employed, including LDSC and MR"

- Page 36, PRS prediction: It is unclear from reading this section what the outcome and predictor variables were. I gathered from elsewhere in the manuscript that the outcome variables are BAG phenotypes and the predictor variables are PRSs capturing genetic propensity towards 36 other traits, diseases and risk factors. Please be more specific in this methods section.

We removed this sentence.

As reviewers #1 and #3 also gave very constructive comments on the PRS results, we removed these old results and performed a new PRS analysis. Specifically, we derived the PRSs using both Plink and PRC-CS (a Bayesian method) for GM, WM, and FC-BAG using the two-split GWASs. That is, we used split1 GWAS as training/base data for the weights and split2 GWAS as testing/target data to calculate the incremental R² to predict the

phenotype of themselves (GM, WM, and FC-BAG). We compared the predictive power of the three BAG-PSCs and found that PRS-CS-derived PRSs are more predictive than those from the PLINK C+T approach (Fig. 4D and lines 340-350):

"We derived the PRS for GM, WM, and FC-BAG using the conventional C+T (clumping plus P-value threshold) approach⁵⁵ via PLINK and a Bayesian method via PRS-CS⁵⁶ (Method 4H).

We found that the GM, WM, and FC-BAG-PRS derived from PRS-CS significantly predicted the phenotypic BAGs in the test data (split2 GWAS, $15,697 < N < 15,940$), with an incremental R^2 of 2.17%, 1.85%, and 0.19%, respectively (Fig. 4D). Compared to the PRS derived from PRS-CS, the PLINK approach achieved a lower incremental R^2 of 0.81%, 0.45%, and 0.14% for GM, WM, and FC-BAG, respectively (Supplementary eFigure 9). Overall, the predictive capacity of PRS is moderate, in line with our earlier discoveries involving raw imaging-derived phenotypes, as demonstrated in Zhao et al.¹³, where PRSs developed for seven selective brain regions were able to explain roughly 1.18% to 3.93% of the phenotypic variance associated with these traits."

Minor points that would improve the readability/accessibility of the manuscript

- Page 5: "The phenotypic heterogeneity of multimodal human brain age derived from three MRI modalities and four machine learning models". This statement seems misleading – the authors did not derive the heterogeneity of multimodal human brain age, but they derived brain age based on different imaging modalities and discovered they were heterogeneous in the genetic correlates they yielded. Would the authors be able to be more specific here?

Thank you again for this suggestion; Reviewer #1 also raised questions. In the revised manuscript, we changed the subtitle to: GM, WM, and FC-BAG derived from three MRI modalities.

- Page 3: Please consider re-wording the following sentence which is challenging to comprehend: "This deviation can be caused by adverse factors, such as AD9, leading to an accelerated brain age (i.e., a positive BAG) or protective factors, such as physical activity or education, resulting in a decelerated brain age (i.e., a negative BAG)."

Thank you for this suggestion. In the revised manuscript, we have rewritten this part (Lines 54-59):

"More precisely, the difference between an individual's AI-predicted brain age and chronological age – brain age gap (BAG) – provides a means of quantifying an individual's brain health by measuring deviation from the normative aging trajectory. BAG has demonstrated sensitivity to several common brain diseases, clinical variables, and cognitive functions⁹, presenting the promising potential for its use in the general population to capture relevant pathological processes."

- Page 4: "In this context, we postulate that AI-derived GM, WM, and FC-BAG can serve as robust, complementary endophenotypes²¹ – close to the underlying etiology – for precision medicine": Did the authors mean that endophenotypes are closer to their genetic underpinnings as compared with other phenotypes such as clinical diagnoses, for example?

Thank the reviewer for this comment.

Like other computational genomics statistical methods, such as Mendelian randomization, endophenotype (EP) has certain assumptions to satisfy. Here is one of the three popular definitions of EP:

Gottesman and Gould:

1. The endophenotype is associated with illness in the population.
2. The endophenotype is heritable.
3. The endophenotype is primarily state-independent (manifests in an individual whether or not illness is active).

4. Within families, endophenotype and illness co-segregate...
5. The endophenotype found in affected family members is found in nonaffected family members at a higher rate than in the general population.

This concept has been initially brought into psychiatric genetics (ref:<https://www.nature.com/articles/mp20108>; <https://ajp.psychiatryonline.org/doi/10.1176/appi.ajp.160.4.636>). The authors state, for example:

"An inherent appeal of EP concept is that a 'good' EP should be closer to the 'level of gene action' than the relevant PD. This concept should translate into the empirical observation that the genetic effects on EP should be stronger than on PD. In their recent review, Flint and Munafò state this explicitly: Much effort has been devoted to finding such endophenotypes, partly because it is believed that the genetic basis of endophenotypes will be easier to analyze than that of psychiatric disease. This belief depends in part on the assumption that the effect sizes of genetic loci contributing to endophenotypes are larger than those contributing to disease susceptibility, hence increasing the chance that genetic linkage and association tests will detect them."

In our case, as we focused on specifically AI-derived endophenotypes, which are inherently data-driven by underlying pathological related factors, we believe these AI-EPs could serve as more robust phenotypes than cognition or disease diagnosis themselves - which are often more heterogeneous and noisy.

For example, in one of our studies, we derived nine such AI-EPs (<https://www.medrxiv.org/content/10.1101/2023.08.16.23294179v1>; currently under review in another journal) from modeling disease heterogeneity in Alzheimer's diseases, autism, late-life depression, and schizophrenia. We found that GWASs on the original case-control diagnosis (e.g., AD vs. healthy control, using GWAS summary statistics from the Psychiatric Genetic Consortium) missed many genetic hits that our AI-EP GWASs captured.

- Page 5: Please provide references for the statement: "Other studies have thoroughly evaluated machine learning models for predicting brain age²⁸, but we selected these as they represent methods currently used in the field" [REF].

Thank you for this suggestion. In the revised manuscript, we removed this sentence and have added a new paragraph to thoroughly discuss our brain age prediction performance with those reported in the literature (Lines 127-138):

" In the literature, other studies³⁰⁻³³ have thoroughly evaluated age prediction performance using different machine learning models and input features. More et al.³⁴ systematically compared the performance of age prediction of 128 workflows (MAE between 5.23–8.98 years) and showed that voxel-wise feature representation (MAE approximates 5–6 years) outperformed parcel-based features (MAE approximates 6–9 years) using conventional machine learning algorithms (e.g., Lasso regression). Using deep neural networks, Peng et al.³⁰ and Leonardsen et al.³¹ reported a lower MAE (nearly 2.5 years) with voxel-wise imaging scans. However, we previously showed that a moderately fitting convolutional neural network (CNN) obtained significantly higher differentiation (a larger effect size) than a tightly fitting CNN (a lower MAE) between the disease and health groups³⁵. To summarize, our study's brain age prediction performance aligns with those reported in the existing literature, considering the utilization of low-dimensional hand-crafted IDPs and conventional machine learning algorithms³⁴."

- Page 5: Considering the focus of the manuscript is the heterogeneity between BAGs and their distinct genetic underpinnings, I suggest it may improve the readability of the manuscript to shift the discussion of how different machine learning algorithms performed to the supplementary. Instead, I suggest it would be better suited to showcase evidence that there indeed is heterogeneity between BAGs.

We appreciate this suggestion. As suggested by Reviewer #1 for down-toning the keyword "heterogeneity", and Reviewer #3 for discussing the age prediction performance in the literature, we reorganized the Results section based on these comments from all three reviewers.

In the revised manuscript, we have added the results for the phenotypic correlation between BAGs (**Fig. 1E**) and genetic correlations (**Fig. 2F**) between each pair of BAGs.

Regarding the ML part for brain age prediction, we tend to keep this in the main manuscript because brain age prediction is the foundation of the subsequent GWAS and post-GWAS analyses - we want to make sure the readers clearly understand how we trained the models and why we chose not to achieve a "better" (a lower MAE) prediction model.

So, in the revised manuscript, we rewrote the first paragraph of the Result section (**Lines 96-107**):

“In the first section, we objectively compared the age prediction performance of four machine learning methods using these GM, WM, and FC-IDPs (**Fig. 1A**). To this end, we employed a nested cross-validation (CV) procedure in the training/validation/test dataset ($N=4000$); an independent test dataset ($N=38,089$)^{26,27} was held out – unseen until we finalized the models using only the training/validation/test dataset (**Method 1**). The four machine learning models included support vector regression (SVR), LASSO regression, multilayer perceptron (MLP), and a five-layer neural network (i.e., three linear layers and one rectified linear unit layer; hereafter, NN)²⁸ (**Method 3**). The second section focused on the main GWASs using the European ancestry population ($31,557 < N < 32,017$) and their sensitivity checks in six scenarios (**Method 4A**). In the last section, we validated the GWAS findings in several post-GWAS analyses, including genetic correlation, gene-drug-disease network, partitioned heritability, PRS calculation, and Mendelian randomization (**Method 4**).”

- Page 5: Could the authors please explain which indicators they used to find that some machine learning models were over-fitted?

We apologize for this confusion and unclearness.

First, we used a nested cross-validation procedure in the training/validation/test dataset ($N=4000$) to train the brain age prediction model for GM, WM, FC-BAG. This resulted in a CV-test MAE (presented in **Supplementary eTable 1**). Besides this training/validation/test dataset, we also have an independent test dataset ($N=38,089$), resulting in an Ind-Test MAE (**Supplementary eTable 1**).

We evaluated four ML models using three different sets of WM IDP features:

- 48 FA: the mean value of the 48 white matter tracts derived from the ICBM-DTI-81 white-matter labels" atlas;
- 192 FA/MD/ODI/NDI: the mean value of the 48 WM tracts in the four diffusion metrics together; FA/MD is derived from the DTI model, and ODI/DNI are derived from the multi-shell NODDI model;
- 108 TBSS: Similar to above, but a tract-skeleton (TBSS) and probabilistic tractography analysis were employed to derive weighted-mean measures within the 27 major WM tracts, resulting in only 108 IDPs

So, we observed clear overfitting phenomena using IDPs from the 192 FA/MD/ODI/NDI and 108 TBSS, but not with the 48 FA IDPs (**Supplementary eTable 1**). For example, the CV-Train for 192 FA/MD/ODI/NDI using Lasso regression is MAE=4.14, which increased to 21.66 in the independent test dataset (Ind-Test).

In the revised manuscript, we clarified this point (**Lines 622-625**):

“Finally, since we observed overfitting – an increase of MAEs from the cross-validated test results to the independent test results – when incorporating features from FA, MD, ODI, and NDI (as detailed in **Supplementary eTable 1A**), we chose to use only the 48 FA WM-IDPs to train the models for generating GM-BAG.”

- Page 8, "The genomic loci linked to GM, WM, and FC-BAG were distributed throughout the human genome, with many locations being distinct": It is unclear what the authors mean by many locations being distinct. Is it that individual BAG phenotypes yielded associations with SNPs that were not found for other BAG phenotypes? Would the authors be able to make this statement clearer?

Thank you for your suggestion and correction. All three reviewers raised questions about this sentence. In the revised manuscripts, we removed this sentence. Initially, we wanted to show that some genomic loci are consistently (by physical position) found between different BAGs. We could directly test this by Bayesian colocalization analysis, but we removed this sentence due to the current results' density.

- Page 10, "While the other four genomic loci did not surpass the genome-wide P-value threshold, they all exhibited local minima and featured the same top lead SNPs or existed in a state of high linkage disequilibrium.": If this is what the authors mean by "local minima", would you not expect the most significant SNPs to have the largest rather than the smallest effect sizes?

We apologize for the misunderstanding and the incorrect wording. Since we are comparing the split-sample or sex-stratified GWAS ($N \sim 15K$) genomic loci vs. the full sample European population (i.e., the main GWAS, $N > 30k$), we expect to obtain less significant results in split-sample or sex-stratified GWASs due to lower sample sizes (P-value depends on sample sizes, but not effect sizes). The term "local minima" may not be precise; what we meant is the "peak signal", where the top lead SNP has the most significant P-value (highest $-\log_{10}(P\text{-value})$) and normally the largest effect size compared to nearby SNPs (they have the same sample sizes).

In the revised manuscript, we rewrote the GWAS sensitivity check sections. Instead of visually checking the concordance of these results, we now reported the concordance rate of the "replication" compared to the genome-wide significant SNPs from the "discovery" data. For example, for the split-sample sensitivity check, we calculated the concordance rate as follows:

"Applying the Bonferroni method to correct for multiple comparisons, we noted high concordance rates between the split1 (as discovery, $15,778 < N < 16,008$) and split2 (as replication, $15,778 < N < 16,008$) GWASs. Specifically, for GM-BAG, we observed a concordance rate of 99% [P-value $< 0.05/3092$; 3092 significant SNPs passing the genome-wide P-value threshold ($< 5 \times 10^{-8}$) in the discovery data], and for WM-BAG, the concordance rate reached 100% (P-value $< 0.05/116$). FC-BAG did not achieve significant genome-wide results in the split-sample GWASs (Supplementary eFigure 3 and Supplementary eFile 2)."

- Figure 4: I suggest this Figure would be more accessible to the reader if there was one location in the x-axis for each trait, whereby estimates for GM, WM and FC would be displayed alongside each other. It took me a while to understand the x-axis repeated the same traits multiple times

We thank the reviewer for this comment. In the revised manuscript, we have changed a new figure followed by this suggestion.

Reviewer #3 (Remarks to the Author):

Overall

This is a very interesting article exploring the relation of Brain Age Gap (BAG) with genetic information. The authors consider X key questions:

- what is the genetic heterogeneity of multimodal BAG
- what are the phenotypes associated with BAG related loci and what are their functional / disease related characteristics
- can we establish the causal relationships between protective/risk factors and decelerated/accelerated brain age

We thank the reviewer for this summary.

First, we summarized the major changes in the revised manuscript and addressed the comments the three reviewers consistently raised. Many of these comments among reviewers converge to similar constructive suggestions.

- The term "heterogeneity" was found to be potentially misleading, as reviewers #1 and #2 commented. Consequently, we have adjusted the emphasis on the term "heterogeneity" throughout the manuscript. We also reorganized the paper and rewrote many places, guided by all three reviewers.
- To justify the brain age prediction performance as presented in our manuscript compared to those reported in prior studies (commented by all three reviewers), we have integrated a new paragraph to discuss these findings. We also conducted an additional experiment involving GM-BAG. Specifically, we employed and compared the performance (brain age prediction and also GWASs) with Support Vector Regression (SVR) on voxel-wise RAVENS maps (GM-BAG-voxel) and MUSE-ROI features (GM-BAG-ROI).
- We performed three additional sensitivity checks (on top of split-sample, sex-stratified, and non-European GWASs) to test the robustness of our main GWASs using European ancestry (suggested by all reviewers #1 and #2).
 - We used SVR to perform feature type-specific GWASs, focusing on different feature types: GM-BAG-ROI versus GM-BAG-voxel.
 - We used MUSE ROIs (GM-BAG) to perform machine learning-specific GWASs, comparing GM-BAG derived from Lasso regression vs. SVR.
 - We reran the three main GWASs using fastGWA (i.e., mixed linear models to account for cryptic population stratification) to justify that there is no substantial genomic inflation in our main GWAS using PLINK linear regression models.
- Furthermore, we calculated the genomic inflation factor (λ) and LDSC intercept (b) in the main GWASs to support further that there is no potential genomic inflation in our main GWASs using European ancestry.
- We derived the polygenic risk scores (PRS) for the GM, WM, and FC-BAG using both Plink (Clumping + Threshold approach) and a more advanced Bayesian approach (PRS-CS) (suggested by all reviewers #1, #2, and #3). Overall, PRS-CS outperforms PLINK in predicting the three BAG-PRSs in the test dataset (split2 GWAS).
- We calculated the phenotypic and genetic correlations for each pair of BAGs in Fig. 1E and Fig. 2F (suggested by reviewer #2).
- We developed a multi-organ web portal (MEDICINE: <http://labs.loni.usc.edu/medicine/>) to disseminate our findings, including showing the Manhattan & QQ plots and allowing the community to download our GWAS summary statistics.

- The keyword "Caucasian" implies racism. We changed the word "Caucasian" to "European" throughout the paper guided by this Nature article (<https://www.nature.com/articles/d41586-021-02288-x>), as the word Caucasian roots in racist taxonomies used to justify slavery.

Secondly, considering that all three reviewers have highlighted our study's significance to imaging genetics, we would like to provide a brief overview (non-exhaustive) of previous GWAS papers conducted on BAG and outline the scientific progress that our study contributes.

It's important to acknowledge that brain age is not a newly emerging biomarker; it has been extensively studied in the literature, as evidenced by pertinent literature cited by reviewer #3. What sets our study apart regarding significance and innovation lies in two aspects. Firstly, prior GWASs of brain age have often been confined to specific aspects, such as GWASs focusing solely on GM-BAG. Secondly, some studies have restricted themselves to the analyses of genome-wide associations without proceeding to undertake comprehensive post-GWAS analyses (e.g., partitioned heritability, gene-drug-disease network, Mendelian randomization, etc.) aimed at partially validating the genetic signals identified. Our study advances the field by adopting a holistic approach that addresses both these limitations, thus contributing to a more comprehensive understanding of the genetic underpinnings of brain age. We have integrated this into the Main section in the revised manuscript.

We want first to acknowledge several representative previous GWAS works on this topic – science is an incremental process built upon the foundation of prior research:

- Kaufmann et al. Nat Neurosci, 2019 performed GWAS on **GM features**, and they then estimated the SNP-based heritability of the brain age using LDSC (<https://www.nature.com/articles/s41593-019-0471-7>).
- Jonsson et al., 2019, Nature Communications: (<https://www.nature.com/articles/s41467-019-13163-9>) also performed the GWAS on **GM-BAG** derived from a deep learning model.
- Smith, S. M. et al. 2020, eLife (<https://elifesciences.org/articles/52677>) performed GWAS on BAGs derived from multimodal brain IDPs; they primarily only focused on GWAS but did not perform comprehensive post-GWAS analyses.
- Leonardsen et al., 2023, Molecular Psy (<https://www.nature.com/articles/s41380-023-02087-y>) performed GWAS on **GM-BAG** derived from a deep learning model and then investigated the causal relationship with other traits.

This summary is by no means exhaustive.

I found the article methods well described and questions of interest. The findings are worth communicating but the general impression is that results are not very biologically compelling and significance maybe mostly due to the very large sample available in the UK Biobank. Indeed, the analyses conducted in this study were constrained by the available imaging-genetics population within UKBB, which encompassed individuals who had undergone both multimodal imaging scans and genetic sequencing. Consequently, the sample size for our study was restricted to approximately 40,000 participants. Nevertheless, it's important to note that to the best of our knowledge, this represents one of the most extensive datasets with such comprehensive data. As open neuroscience advances, studies like the Adolescent Brain Cognitive Development (ABCD) and Alzheimer's Disease Neuroimaging Initiative (ADNI) expand their sample sizes. This expansion holds promise for the future, enabling us to work with larger sample sizes essential for GWAS that can yield more biologically compelling results.

That being stated, we believe that our post-GWAS analyses, while not offering direct biological validation (as mentioned by reviewer #2), do contribute in some measure to substantiating the GWAS signals discovered in our study.

Compared to previous GWASs on brain age gaps, our study partially validates these GWAS signals by conducting several post-GWAS analyses using state-of-the-art computational genomics statistical methods. These include:

- Genetics correlation: we expected the three BAGs to be genetically correlated with other brain diseases (**Fig. 4A**). For example, we found a significant positive genetic correlation between the GM- and WM-BAG and one subtype of AD, defined by a semi-supervised AI method (<https://www.biorxiv.org/content/10.1101/2022.09.16.508329v2>, under review) and characterized by global brain atrophy. Of note, AD1 was originally defined in ADNI, and the GM-, WM-BAG in this paper was defined in part of the UKBB data.
- Gene-set enrichment analysis (GSEA) (additional experiment suggested by the reviewer in the subsequent comment): In **Fig. 2B**, we found that GM-, and WM-BAG-associated loci were previously linked to AD in the GWAS Catalog. We performed one additional GSEA analysis using GENE2FUNC on FUMA to test (i.e., hypergeometric test) if the genes associated with the GM and WM-BAG were enriched in a pre-defined gene set defined by "Alzheimer's disease in APOE e4-carriers" by GWAS Catalog. We found significant enrichment for GM-BAG genes: P-value = 2.84460968895911e-07.
- Cell type-specific partitioned heritability estimates (**Fig. 4C**): WM-BAG exhibited significant heritability enrichment in oligodendrocytes – one type of neuroglial cells. FC-BAG showed such enrichment in astrocytes, the most prevalent glial cells in the brain. This, especially with the WM-BAG, should be biologically expected, as oligodendrocytes are primarily responsible for forming the lipid-rich myelin structure, whereas astrocytes play a crucial role in various cerebral functions, such as brain development and homeostasis. Convincingly, our previous GWAS on WM-IDP (not BAG) (Zhao et al. Science, PMID: 34140357) also independently identified considerable heritability enrichment in glial cells, especially oligodendrocytes. This consistency largely validated our current GWAS findings.
- Gene-drug-disease network (**Fig. 3**): Our bioinformatic analyses performed gene mapping (by physical position, chromatin interaction, and/or eQTL mapping), searched these mapped genes as targets in the Drug Bank database and the Therapeutic Target Database, and provided putative evidence that these BAG-related genes were used as target genes for treating brain-related diseases, such as Semorinemab (RG6100), an anti-tau IgG4 antibody, being investigated in a phase-2 clinical trial (trial number: NCT03828747), which targets extracellular tau in AD – the most common form of dementia – to reduce microglial activation and inflammatory responses.
- Similarly, Mendelian randomization results also provided confirmation that other brain diseases (e.g., AD) or systemic diseases (e.g., type 2 diabetes) could causally influence human brain age.

The authors present a large set of methods and results, each could deserve more in depth validation or analyses.

Thank you for pointing this out regarding the methodological consideration in our study. Overall, we admit that the methods used in the current study, including the genomics statistical methods, have strong assumptions. For example, Mendelian randomization has several assumptions, including the Instrumental Variable (IV) Assumption, Independence

Assumption, and No Pleiotropy Assumption. Hence, it is imperative to exercise caution when interpreting these findings. For example, we did sensitivity checks in the result section to check whether these assumptions were violated in MR (Line 390). We also did six sensitivity checks for the main GWAS using the European ancestry (Line 196). At the end of the discussion, we added one Limitation section to discuss these limitations (Line 551-552).

For instance the image derived phenotypes used to compute brain age aren't the most sensitive ones reported (More et al, 2023).

We specifically addressed this comment below.

In terms of the general approach, the authors seem to first look at genomic loci linked to BAG - then look if these loci are related to disease with a genome wide association - but a stronger analysis would be to compare directly the loci found with GAP to the loci found in association to disease, and assess the likelihood of the proximity of these loci using non parametric techniques.

We strongly agree with the reviewer regarding this comment. To be precise and clear, in **Fig. 2B**, we performed a bioinformatic query on these genomic loci linked to BAG in the previous literature in the GWAS Catalog to understand what clinical traits were previously associated with the genetic variants within the BAG-associated loci.

We agree that we can use some nonparametric statistical method to directly test the enrichment of certain genes or genetic variants (e.g., genes associated with WM-BAG) that are overrepresented in any of the pre-defined gene sets linked to a certain disease (e.g., AD).

In addition, one can also test the genetic colocalization signal between the BAG loci vs. the disease-specific loci (e.g., *APOE* gene loci in AD).

As we have used alternative approaches, such as genetic correlations and MR analyses, we will test this type of analysis in future studies.

Specific

* BAG : could the authors explain why is the MAE greater than the one found in many brain age studies ? (e.g. Leonardsen et al. 2022, He et al. 2021, Hahn et al. 2022, Wood et al. 2022, Peng et al. 2021)

We appreciate the reviewer for these comments and for allowing us to explain this from our perspective and experience. First, we admit that brain age prediction has been extensively studied, and the reported MAE varies across studies, similar to other machine learning tasks, such as AD classification

(<https://www.sciencedirect.com/science/article/pii/S1361841520300591>).

In our opinion, several observations can be drawn from these studies and our previous works: *i*) deep learning models (such as CNN) with voxel-wise MRI scans normally obtained a lower MAE compared to ROI-based features (as evidenced in these papers: Leonardsen et al. 2022, He et al. 2021, Hahn et al. 2022, Wood et al. 2022, Peng et al. 2021, and another one from Couvy-Duchesne et al., 2020.

<https://www.ncbi.nlm.nih.gov/pmc/articles/PMC7770104/> from the PAC2019 competition); *ii*) Voxel-wise features performed better than parcel-wise features (especially when deep learning models were employed, and *iii*) we showed that moderately-fitting brain age models (a lower MAE) obtain significantly higher differentiation (a larger effect size) than tightly-fitting models (a lower MAE) between the disease and health groups (e.g., AD vs. CN) (Bashyam et al., 2020, Brain. <https://academic.oup.com/brain/article/143/7/2312/5863667>).

In general, the accuracy of our brain age prediction aligns with the findings of two recent studies (Couvy-Duchesne et al., 2020, Front Psychiatry; More et al., 2023, Neuroimage) that systematically and consistently evaluated the MAE across various sets of

machine learning models and imaging features, particularly those based on regions of interest (ROI). Especially for the latter, More et al. systematically evaluate the age prediction MAE in different feature types and machine learning algorithms (but not deep learning). They observed that voxel features (5-6 features, but some of the experiments used PCA feature reduction) outperform parcel-based features (MAE=6-9 years). We obtained an MAE of ~5 years using GM IDPs (119 MUSE GM ROI).

In the revised manuscript, we have now added a new paragraph to discuss this (Lines 127-138):

"In the literature, other studies³⁰⁻³³ have thoroughly evaluated age prediction performance using different machine learning models and input features. More et al.³⁴ systematically compared the performance of age prediction of 128 workflows (MAE between 5.23–8.98 years) and showed that voxel-wise feature representation (MAE approximates 5-6 years) outperformed parcel-based features (MAE approximates 6-9 years) using conventional machine learning algorithms (e.g., Lasso regression). Using deep neural networks, Peng et al.³⁰ and Leonardsen et al.³¹ reported a lower MAE (nearly 2.5 years) with voxel-wise imaging scans. However, we previously showed that a moderately fitting convolutional neural network (CNN) obtained significantly higher differentiation (a larger effect size) than a tightly fitting CNN (a lower MAE) between the disease and health groups³⁵. To summarize, our study's brain age prediction performance aligns with those reported in the existing literature, considering the utilization of low-dimensional hand-crafted IDPs and conventional machine learning algorithms³⁴."

We also added this as a limitation in the Limitation section (Lines 540-556):

"This study has several limitations. We can employ deep learning on voxel-wise imaging scans to enhance brain age prediction performance. Nevertheless, it warrants additional exploration to determine whether the resulting reduction in MAE translates into more robust genome-wide associations, as our previous work has demonstrated that BAGs derived from a CNN with a lower MAE did not exhibit heightened sensitivity to disease effects such as AD³⁵. Second, the generalization ability of the GWAS findings to non-European ancestry is limited, potentially due to small sample sizes and cryptic population stratification. Future investigations can be expanded to encompass a broader spectrum of underrepresented ethnic groups, diverse disease populations, and various age ranges spanning the entire lifespan. This expansion can be facilitated by leveraging the resources of large-scale brain imaging genetic consortia like ADNI⁸⁰, focused on Alzheimer's disease, and ABCD⁸¹, which centers on brain development during adolescence. Third, it's important to exercise caution when interpreting the results of this study due to the various assumptions associated with the statistical methods employed, including LDSC and MR. Lastly, it's worth noting that brain age represents a residual score encompassing measurement error. A recent study⁸² has underscored the significance of incorporating longitudinal data when calculating brain age. Future research should be conducted once the longitudinal scans from the UK Biobank become accessible to explore this impact on GWASs."

* More et al 2023 suggest that voxel / vertex wise methods are to be preferred for brain age computation, do the author think the results would have been different with the associated BAG ?

Thanks for this comment. To be precise, More et al. systematically evaluate the age prediction MAE in different feature types and machine learning algorithms (but not deep learning). They observed that voxel features (5-6 features, but some experiments used PCA feature reduction) outperform parcel-based features (MAE=6-9 years).

We did an additional experiment to explore this impact on brain age prediction performance. As the lab (Dr. Davatzikos) at UPENN only downloaded and processed the T1-weighted MRI (the diffusion MRI and rsfMRI's features were directly requested and downloaded from the UKBB showcase website), we re-performed brain age prediction using linear SVR + voxel-wise RAVENS maps (GM-BAG-voxel) using the same cross-validation procedure as we derived the GM-BAG (GM-BAG-ROI). We found that in the

training/validation/test dataset, the CV-test indeed obtained a lower MAE (4.31 years), but this MAE increased to 5.12 years in the independent dataset – marginal overfitting potential due to the high dimensions of the voxel features (similar to the ones obtained in More et al.) (Lines 232-235):

“The BAGs derived from the two types of features were significantly correlated ($r=0.74$; $P\text{-value}<1\times 10^{-10}$). The brain age prediction performance using RAVENS showed marginal overfitting, with an MAE of 4.31 years in the training/validation/test dataset and an MAE of 5.12 years in the independent test dataset.”

Furthermore, we performed another sensitivity GWAS analysis (feature type-specific GWASs, lines: 229-235):

“We finally found a 92.43% concordance rate of the SNPs identified in the GM-BAG GWAS using the 119 MUSE ROIs⁴⁰ (as discovery, BAG MAE=4.39 years) and voxel-wide RAVENS⁴¹ maps (as replication, $P\text{-value} < 0.05/3382$, BAG MAE=5.12 years) (Supplementary eFigure 8 and Supplementary eFile 7).”

In the Limitation section, we mentioned that future studies using CNN on voxel-wise features need to be done to assess whether these lower MAEs will provide more robust GWAS signals (line: 229-235):

“This study has several limitations. We can employ deep learning on voxel-wise imaging scans to enhance brain age prediction performance. Nevertheless, it warrants additional exploration to determine whether the resulting reduction in MAE translates into more robust genome-wide associations, as our previous work has demonstrated that BAGs derived from a CNN with a lower MAE did not exhibit heightened sensitivity to disease effects such as AD³⁵.”

* How does the genetic loci of BAG related to loci of AD / aging ? can you think of an analysis that would demonstrate if the relation isn't found by chance ?

We appreciate the reviewer for this suggestion. To address this suggestion:

Relation to aging: we compared our GWAS to these genetic loci identified in the previous GWASs on brain age gap (mainly on GM-BAG) in the previous GM-BAG GWASs (WM-BAG and FC-BAG were barely investigated in GWASs). The genomic loci found in our study with GM-BAG were consistently identified and independently shown (by different teams on the same dataset) in the previous literature.

For example, the strongest genetic signal was found on the top lead SNP rs534115641 at 17q21.31 (Chromosome 17, although we know that this region is usually very pleiotropy).

- The Manhattan plot of our GM-BAG for this locus:

- The Manhattan plot from Ning et al., Neurobiology of aging; $N=16,998$ European ancestry individuals, <https://pubmed.ncbi.nlm.nih.gov/34098431/>

[redacted]

- The Manhattan plot from Jonsson et al., 2019, Nature Communications (<https://www.nature.com/articles/s41467-019-13163-9>):

[redacted]

- The Manhattan plot from Leonardsen et al. 2023, Mol Psy (<https://www.nature.com/articles/s41380-023-02087-y>):

[redacted]

- The Manhattan plot from Smith et al. 2019, eLife (<https://elifesciences.org/articles/52677>) - the orange dots on that genomic region where the brain age was trained by using all 3913 UKBB IDPs (including T1 & T2 ROIs, DTI & NODDI metrics, and fMRI metics, etc.):

[redacted]

Unfortunately, as far as we know, all these previous GM-BAG GWASs did not share their GWAS summary statistics, so we could not do any further analyses to support further the consistency of the genetic signals found across studies. If these were publicly available, we could have run:

1. Genetic correlation (expected to have a high r_g)
2. Genetic colocalization (expected to obtain a high PP.H4.ABF)

To facilitate future BAG GWASs, we made our GWAS summary statistics in the MEDICINE web portal: <http://labs.loni.usc.edu/medicine/>.

Relate to AD: our current genetic correlation and Mendelian randomization have already shown the relation of the multimodal BAG with AD.

- GM-BAG and WM-BAG are genetically correlated with one subtype of AD (AD1, Fig. 4A), which was defined using AI (semi-supervised clustering using GAN) in another study (<https://www.biorxiv.org/content/10.1101/2022.09.16.508329v2>; under review at another journal).
- AD shows a potential causal relationship to WM-BAG (Fig. 5A)
- To further support this relationship, we also performed a gene-set enrichment analysis (using GENE2FUNC on FUMA) to test (i.e., hypergeometric test) if the genes associated with the three BAGs (especially GM and WM-BAG have shown SNP-BAG signals in AD in GWAS Catalog literature) were enriched in a pre-defined gene set defined by Alzheimer's disease in APOE e4- carriers by GWAS Catalog. We found a significant enrichment of GM-BAG in this gene set: P-value = 2.84460968895911e-07.

* BAG computed cross sectionally may be questionable in the view of the brain charting results (Di Biase, PNAS 2023)

Thank you for this reference. We knew this work after our submission to Nature Communications. This is an important work to show that longitudinal data is critical while calculating brain age using ML.

In the revised manuscript, we added this as a limitation (Lines 553-556). As UKBB plans to release the second point of imaging scans (currently only a very small proportion of individuals who have longitudinal scans), future works can explicitly address and validate this from genetic perspectives - as GWAS always needs a large sample size:

“Lastly, it's worth noting that brain age represents a residual score encompassing measurement error. A recent study⁸² has underscored the significance of incorporating longitudinal data when calculating brain age. Future research should be conducted once the longitudinal scans from the UK Biobank become accessible to explore this impact on GWASs.”

* There has been considerable analytical variability reported in brain imaging - are results robust with choice of pipelines to derived IDP ?

This is a very good suggestion. The imaging pipeline is a critical part of reproducible machine learning. In our previous studies (the main PhD work of Dr. JW), we have

reproducibly evaluated the impact of imaging pipeline on brain age prediction (Couvy-Duchesne et al., 2020, <https://www.ncbi.nlm.nih.gov/pmc/articles/PMC7770104/>), as well as AD classification (Wen et al., 2020, <https://www.sciencedirect.com/science/article/pii/S1361841520300591>; Samper-González et al., 2018, <https://www.sciencedirect.com/science/article/pii/S1053811918307407>). We agree that this aspect is worth further investigation in a future study. Partially, we have seen the convergence of genetic signals in our feature type-specific GWAS sensitivity checks (lines 229-232):

“We finally found a 92.43% concordance rate of the SNPs identified in the GM-BAG GWAS using the 119 MUSE ROIs⁴⁰ (as discovery, BAG MAE=4.39 years) and voxel-wide RAVENS⁴¹ maps (as replication, P-value < 0.05/3382, BAG MAE=5.12 years) (**Supplementary eFigure 8 and Supplementary eFile 7**).”

It's worth noting that MUSE ROI and RAVENS are not entirely distinct imaging pipelines. In the future, a study could be conducted that compares the use of MUSE ROIs versus Freesurfer ROIs in the brain age prediction task and subsequent GWAS analyses.

* "the locus associated with GM-BAG (top lead SNP: rs61732315, 1q32.1) and 164th locus related to WM-BAG (top lead SNP: rs11118475, 1q32.2) were in proximity" : it is unclear if it can be established that this proximity is indicative of a specific result: is this testable (can we test the hypothesis that loci are related) ?

Thank you for your suggestion and correction. Initially, we wanted to show that some genomic loci are consistently (by physical position) found between different BAGs. We could directly test this by Bayesian colocalization analysis, but we removed this sentence in the revised manuscript due to the density of the current results.

* Was the discovery of the 6 loci done on the full sample before doing the split-sample analysis ? what is the number of times these loci are found again across many splits ? We thank the reviewer for this comment (also commented by the other two reviewers regarding sample sizes). We apologize for this confusion.

In the revised manuscript, we explicitly added the sample sizes for all analyses whenever applicable. To answer the reviewer's specific questions: Yes, the six loci associated with GM-BAG used the full sample (N=31557 European ancestry), and the split-sample analyses (split1 and split2) used only half of the entire European population (ensuring age and sex did not differ between the two splits). Here, we want to show the concordance rate of the genetic signals between the two splits. Compared to the GWAS using full sample sizes, of course, we expected to discover fewer loci in both of the two splits' GWASs.

We updated the number of sample sizes in the updated sensitivity check results (lines 196).

* Genetic covariance: for interpretation, it would be important to report the heritability of the traits on which the correlation is computed

We agree on this. In the revised manuscript, we added this information in **Supplementary eTable 4**.

Of note, the SNP-based heritability reported here was obtained using the LDSC software and the GWAS summary statistics. This differs from the h^2 estimates using GCTA, which has slightly different hypotheses, allele frequency, etc. In addition, the latter uses the raw imputed genetic data (individual-level) but not the summary statistics to compute the SNP-based h^2 . We also briefly discussed this in **Method 4D**.

Here are some works that systematically compare the h2 estimates across different software:

- <https://www.nature.com/articles/s41588-018-0108-x>
- <https://www.ncbi.nlm.nih.gov/pmc/articles/PMC8425307/>

* Polygenic risk scores of other diseases weakly predict multimodal BAG: while the p-values are very small, this seems to be a side effect of the number of sample. The additional variance explained of less than 0.3% does not seem to be biologically relevant. One possible analysis would be to see which loci are shared between PRS predicting loci and those associated with BAGs.

We agree with the reviewer regarding the rationale of this analysis, as also commented by reviewers #2 and #1.

In the revised manuscript, we removed this section and replaced it with a new analysis. Specifically, we derived the PRSs using both Plink and PRC-CS (a Bayesian method) for GM, WM, and FC-BAG using the two-split GWASs. That is, we used split1 GWAS as training/base data for the weights and split2 GWAS as testing/target data to calculate the incremental R² to predict the phenotype of themselves (GM, WM, and FC-BAG). We compared the predictive power of the three BAG-PSCs and found that PRS-CS-derived PRSs are more predictive than those from the PLINK C+T approach (**Fig. 4D** and lines **340-350**):

"We derived the PRS for GM, WM, and FC-BAG using the conventional C+T (clumping plus P-value threshold) approach⁵⁵ via PLINK and a Bayesian method via PRS-CS⁵⁶ (**Method 4H**).

We found that the GM, WM, and FC-BAG-PRS derived from PRS-CS significantly predicted the phenotypic BAGs in the test data (split2 GWAS, 15,697<N<15,940), with an incremental R² of 2.17%, 1.85%, and 0.19%, respectively (**Fig. 4D**). Compared to the PRS derived from PRS-CS, the PLINK approach achieved a lower incremental R² of 0.81%, 0.45%, and 0.14% for GM, WM, and FC-BAG, respectively (**Supplementary eFigure 9**). Overall, the predictive capacity of PRS is moderate, in line with our earlier discoveries involving raw imaging-derived phenotypes, as demonstrated in Zhao et al.¹³, where PRSs developed for seven selective brain regions were able to explain roughly 1.18% to 3.93% of the phenotypic variance associated with these traits."

* Causality analyses: The authors performed MR analyses to establish whether the clinical traits previously associated with the genomic loci associated with BAG were a cause or a consequence of GM, WM, and FC-BAG. These are interesting analyses but there are strong assumptions for MR analyses and it is unclear if these are met here (e.g., the "no horizontal pleiotropy" assumption).

We strongly agree with the reviewer for this comment. In the revised manuscript, we have now explicitly stated several assumptions of Mendelian randomization and how the five different methods used in our study overcome these limitations (Lines **737-749**):

"We reported the results of IVW in the main text and the four others in the **Supplementary eFile 9**. MR relies on a set of crucial assumptions to ensure the validity of its results. These assumptions include the requirement that the chosen genetic instrument exhibits a strong association with the exposure of interest while remaining free from direct associations with confounding factors that could influence the outcome. Additionally, the genetic variant used in MR should be independently allocated during conception and inheritance, guaranteeing its autonomy from potential confounders. Furthermore, this genetic instrument must affect the outcome solely through the exposure of interest without directly impacting alternative pathways that could influence the outcome (no horizontal pleiotropy). The five MR methods handle pleiotropy and instrument validity

assumptions differently, offering various degrees of robustness to violations. For example, MR Egger provides a method to estimate and correct for pleiotropy, making it robust in the presence of horizontal pleiotropy. However, it assumes that directional pleiotropy is the only form of pleiotropy present.”

We have performed extensive sensitivity checks for our MR results in the last two paragraphs of the MR result section (a new section starting from line **390-414, Fig. 5B-E**) by showcasing the potential causal relationship from the triglyceride-to-lipid ratio to GM-BAG.

In addition, we also added this point in the Limitation section for interpreting our MR results:

“Third, it's important to exercise caution when interpreting the results of this study due to the various assumptions associated with the statistical methods employed, including LDSC and MR.”

REVIEWER COMMENTS

Reviewer #1 (Remarks to the Author):

The authors have done a good job replying to reviewer concerns.

The writing, however, needs to be further polished. Several connective sentences and explanatory words/phrases have been added into the manuscript, all facilitates an easy reading (great and thank you!).

As a reader and reviewer, however, I still found the manuscript a bit difficult to read - I encounter, from time to time, broken logics. For example, the first paragraph of results provides an overview of what's been done in the next a few paragraphs, which results correspond to which methods. Yet, please note, method 2 has never been cited. As emphasized in my round-1 review comments, Nature journal has its own structure, results come right after introduction. I strongly urgent the authors polish this manuscript in a careful way, both structurally (more connective sentences and phrases are good!), grammatically and in formats (math symbols for example).

I have no further comments on the scientific values of the manuscript.

Reviewer #2 (Remarks to the Author):

I would like to congratulate the authors on this interesting article, and the important findings they present. I am certain that my future work will draw on the provided GWAS summary statistics and reference the findings. I would also like to thank the authors for so thoroughly addressing all the comments I have raised. It is evident that a lot of hard work has led to the production of this manuscript, and it is much appreciated that the authors were willing to tailor their work according to my suggestions. Thank you for making the code publicly available, which makes this an even more valuable contribution to advancing neuroimaging genetics.

I have a last few very minor comments that may have gotten lost between the more major changes.

- To improve the readability of the following sentence in the abstract, I suggest the authors could split the sentence in two: "GM-BAG showed the highest heritability enrichment for genetic variants in 40 conserved regions, whereas WM-BAG exhibited the highest heritability enrichment in the 5' 41 untranslated regions; oligodendrocytes and astrocytes, but not neurons, showed significant 42 heritability enrichment in WM and FC-BAG, respectively". Consider using a full stop [.] rather than a semicolon [;].

- Typo in line 205: "spit-sample"

- Line: 583-586: In addition to indicating the ICD codes, could the authors please indicate the UKB field ID/ data field code used to screen their population? Was it, for example, field code 41270?

- Typo in eFigure 2 title: "eFigure 2: Genetic correlation (gc) between the GM, WM, and FC-BAG using the LDSC software in the split-sample analyese"

- Line 702: Finally, would the authors consider adding one sentence to explain what AI-derived subtypes of AD/ASD/SCZ are, and how they have been derived? I am not familiar with the cited studies and cannot derive from the manuscript how the different traits differ (i.e., AD vs. AD1 vs. AD2) and what they represent

Reviewer #3 (Remarks to the Author):

The authors have improved the manuscript and have clarified aspects raised by the comments made during the first round of reviews. Rereading of the manuscript there are still a few points that seem

important to investigate.

* As the BAG computation is not in par with the current literature it would be a good confirmation if a more precise estimate could be used and this would go a long way assessing the robustness of the results. The fact that "BAGs derived from a CNN with a lower MAE did not exhibit heightened sensitivity to disease effects such as AD (ref 35)" but this could still impact the genetic analyses that are at the core of the article results. Confirming (or not) results with a more accurate BAG seems important. If the results are only found for this specific BAG estimation, the interpretation will be profoundly altered.

* An independent validation is probably needed to assert the solidity of the results. This could be done with an number of datasets including ADNI, PREVENT-AD, and others in Europe.

* It is hard to understand why MR identifies a potential risk of AD on WM-BAG but not on GM-BAG given the well known mechanisms for GM reduction with AD progression.

* While significant, PRS-CS R2 increase of 2.17%, 1.85%, and 0.19% for BAGs phenotypes are very small values, the authors might want to update the description of the results to emphasize the small effect sizes

* The effort in reproducibility of the work are important and excellent, and this reviewer wholeheartly commend the authors for releasing part of their code on GitHub.

Other :

* The new manuscript <https://www.biorxiv.org/content/10.1101/2022.09.16.508329v2> ... ?

* Cheverud conjecture doesn't seem to be realized here with much larger genetic correlations than phenotypic correlations

* The fact that the genetic correlations is found between GM-BAG and disease sub-types rather than with disease diagnoses might simply reflect that by clustering the population in groups for which several phenotypic dimensions are capture, there is simply more chance to have significant correlations. The authors could assess the correction needed on "null" data (e.g. other populations for which correlation is not expected).

* The heritability numbers are rather impressive : is really 47% of the variance of GM-BAG explained by SNPs ? Could the authors explain why this is plausible (or is the number due to a methodological issue?)

* It is hard to understand the overfitting issue "..., with Lasso regression, the cross-validated test result (CV test) obtained an MAE of 4.94 for all 192 FA/MD/ODI/NDI metrics, but the independent test result (Ind. test) obtained an MAE of 1.66." Can the author provide insight on how overfitting can occur in completely independent data ? what were the cross validation results (i.e. results on validation data in the CV loop) ?

Author response letter, ID: NCOMMS-23-20963A

We extend our sincere gratitude to the three reviewers for their insightful comments, which have played a pivotal role in enhancing our manuscript. In this response letter, the comments from each reviewer are in black font in this response letter; our responses are in blue font. In the revised manuscript, we tracked the changes in the yellow-colored text.

REVIEWER COMMENTS

Reviewer #1 (Remarks to the Author):

The authors have done a good job replying to reviewer concerns.

We thank the reviewer for this encouragement.

The writing, however, needs to be further polished. Several connective sentences and explanatory words/phrases have been added into the manuscript, all facilitates an easy reading (great and thank you!).

As a reader and reviewer, however, I still found the manuscript a bit difficult to read - I encounter, from time to time, broken logics. For example, the first paragraph of results provides an overview of what's been done in the next a few paragraphs, which results correspond to which methods. Yet, please note, method 2 has never been cited.

We sincerely apologize for this. **Method 2** was not cited in the Results section but was cited in **Method 3** (removed in the revised manuscript now). As the reviewer pointed out below, Nature journals put the Method section behind the Results section. To address this, we have now added one paragraph in the first paragraph of the Results section (**Line: 101**):

"The GM, WM, and FC-IDPs were derived from three MRI modalities (**Method 2**)."

As emphasized in my round-1 review comments, Nature journal has its own structure, results come right after introduction. I strongly urgent the authors polish this manuscript in a careful way, both structurally (more connective sentences and phrases are good!), grammatically and in formats (math symbols for example).

We appreciate the reviewer for this comment and her/his prior feedback regarding the writing and organization of our manuscript. For the revised manuscript, W.J. and I.S. have independently double-checked the manuscript to correct typos, grammar mistakes, and figure/table numbers.

In the revised manuscript, we tried to add more connective sentences between these Results sections to guide the readers to understand the main messages better. In the revised manuscript, at the beginning of each subsection, we added one sentence to link the motivation of the analysis from the previous results. For example, for partitioned heritability analysis (**Line: 288-290**), we added these sentence at the beginning of the subsection:

"As the three BAGs showed significant SNP-based heritability estimates, we conducted a partitioned heritability analysis⁵² to investigate further the heritability enrichment of these genetic variants in the 53 functional categories and specific cell types (**Method 4E**)."

In addition, we moved the sensitivity check analyses for our three primary GWAS into **Supplementary eText1**, as the content is quite dense. Also, we moved the demographic table (original **Table 1**) into **Supplementary eTable 7**.

We sincerely appreciate the reviewer's attention and efforts to guide us in improving the manuscript! We believe that we have dedicated our utmost care and attention to enhancing these

facets, improving writing, grammar, and overall structure.

I have no further comments on the scientific values of the manuscript.
Thank you again for helping improve our work!

Reviewer #2 (Remarks to the Author):

I would like to congratulate the authors on this interesting article, and the important findings they present. I am certain that my future work will draw on the provided GWAS summary statistics and reference the findings. I would also like to thank the authors for so thoroughly addressing all the comments I have raised. It is evident that a lot of hard work has led to the production of this manuscript, and it is much appreciated that the authors were willing to tailor their work according to my suggestions. Thank you for making the code publicly available, which makes this an even more valuable contribution to advancing neuroimaging genetics.

We appreciate the reviewer's words of encouragement. We firmly believe in open science within AI, neuroimaging, and genetics as a fundamental catalyst for advancing scientific knowledge in our community. We are enthusiastic about sharing our source code and GWAS summary statistics, allowing the wider community to scrutinize our work and utilize our data for future analyses.

I have a last few very minor comments that may have gotten lost between the more major changes.

- To improve the readability of the following sentence in the abstract, I suggest the authors could split the sentence in two: "GM-BAG showed the highest heritability enrichment for genetic variants in 40 conserved regions, whereas WM-BAG exhibited the highest heritability enrichment in the 5' 41 untranslated regions; oligodendrocytes and astrocytes, but not neurons, showed significant 42 heritability enrichment in WM and FC-BAG, respectively".

Consider using a full stop [.] rather than a semicolon [;].

We have revised these sentences in the abstract. We have changed the abovementioned sentences to:

GM-BAG displayed the most pronounced heritability enrichment in genetic variants within conserved regions. WM-BAG showcased the highest heritability enrichment in the 5' untranslated regions. Among the three cell types considered, oligodendrocytes and astrocytes, but not neurons, exhibited notable heritability enrichment in WM and FC-BAG, respectively.

- Typo in line 205: "spit-sample"

Thanks for this correction!

- Line: 583-586: In addition to indicating the ICD codes, could the authors please indicate the UKB field ID/ data field code used to screen their population? Was it, for example, field code 41270?

This is a great suggestion to enable the reproducibility of our analyses. In the revised manuscript, we added this information (Line: 561).

Yes, the Filed ID that we used to define the training sample is from 41270:

```
print('Generate data for the following variable: %s' % var_list[i])
if var_list[i] == 'diagnoses_icd10_f41270':
    df_vari = df_ukbb_data.filter(like=var_list[i])
```

- Typo in eFigure 2 title: "eFigure 2: Genetic correlation (gc) between the GM, WM, and FC-BAG using the LDSC software in the split-sample analyse"

Thanks for this correction!

- Line 702: Finally, would the authors consider adding one sentence to explain what AI-derived subtypes of AD/ASD/SCZ are, and how they have been derived? I am not familiar with the cited studies and cannot derive from the manuscript how the different traits differ (i.e., AD vs. AD1 vs. AD2) and what they represent

This is a great suggestion. In the revised manuscript, we added more information regarding this aspect (Line: 273-277):

" To illustrate this, AD1 and AD2 distill the neuroanatomical heterogeneity of Alzheimer's disease into two distinct imaging patterns: AD1 represents a widespread brain atrophy pattern, while AD2 exhibits a focal atrophy pattern in the medial temporal lobe⁴. These subtypes, in essence, capture more homogeneous disease effects than the conventional "unitary" disease diagnosis, hence serving as robust endophenotypes²³."

Reviewer #3 (Remarks to the Author):

This is a comment from the first round from the reviewer, and we think we did not fully address this question. Here, we provided additional experiments to address this comment.

* How does the genetic loci of BAG related to loci of AD / aging ? can you think of an analysis that would demonstrate if the relation isn't found by chance ?

Relate to AD: this has been fully addressed from our perspectives.

Relate to aging: This is an excellent suggestion. It would be interesting to test genetic relationships with aging traits less closely tied to predictors (these IDPs) used to compute the BAG, such as **longevity** or **telomere length**. To perform this additional analysis, we searched the GWAS Catalog to download the corresponding GWAS summary statistics and performed the same quality checks for these two phenotypes.

- **Longevity:** For longevity phenotype, we finally included the GWAS summary statistics from this study (PMID: 31413261; <https://www.ebi.ac.uk/gwas/publications/31413261>) for genetic correlation analysis. However, we cannot perform Mendelian randomization analysis due to the limited power (only one genomic locus was identified in the GWAS figure in that paper). 2SampleMR packages only used these genome-wide significant SNPs (considering LD) as instrumental variables for the association between the exposure and outcome variables.
- **Telomere length:** we download the GWAS summary statistics from this study (PMID: 35681050). This study is one of the GWASs whose summary statistics (common variants, but not rare variants) are publicly available on the GWAS Catalog (https://www.ebi.ac.uk/gwas/efotraits/EFO_0004505). However, the sample size here is low (N=902 European). This leads to 0 genomic locus that passed the genome-wide significance threshold (5×10^{-8}), and LDSC cannot converge (function: "munge_sumstats.py") this with other clinical traits: "Warning: 0 genome-wide significant SNPs (some may have been removed by filtering)". This led to the following error when running the main function (ldsc.py):

```

*****
* LD Score Regression (LDSC)
* Version 1.0.1
* (C) 2014-2019 Brendan Bulik-Sullivan and Hilary Finucane
* Broad Institute of MIT and Harvard / MIT Department of Mathematics
* GNU General Public License v3
*****
Call:
./ldsc.py \
--ref-ld-chr /cbica/home/wenju/Project/ldsc/pre-computed_Eu_GWAS/eur_w_ld_chr/ \
--out /cbica/home/wenju/Reproducible_paper/BiologicalAge/output/GWAS/Brain_age_gap/gc/Brain_age_gap_vs_telomere_length \
--rg /cbica/home/wenju/Reproducible_paper/BiologicalAge/output/GWAS/Brain_age_gap/gc/Brain_age_gap.sunstats.gz,telomere_length.sunstats.gz \
--w-ld-chr /cbica/home/wenju/Project/ldsc/pre-computed_Eu_GWAS/eur_w_ld_chr/

Beginning analysis at Fri Oct 20 19:39:17 2023
Reading summary statistics from /cbica/home/wenju/Reproducible_paper/BiologicalAge/output/GWAS/Brain_age_gap/gc/Brain_age_gap.sunstats.gz ...
Read summary statistics for 768875 SNPs.
Reading reference panel LD Score from /cbica/home/wenju/Project/ldsc/pre-computed_Eu_GWAS/eur_w_ld_chr/[1-22] ... (ldscore_fromlist)
Read reference panel LD Scores for 1293150 SNPs.
Removing partitioned LD Scores with zero variance.
Reading regression weight LD Score from /cbica/home/wenju/Project/ldsc/pre-computed_Eu_GWAS/eur_w_ld_chr/[1-22] ... (ldscore_fromlist)
Read regression weight LD Scores for 1293150 SNPs.
After merging with reference panel LD, 759220 SNPs remain.
After merging with regression SNP LD, 759220 SNPs remain.
Computing rg for phenotype 2/2.
Reading summary statistics from telomere_length.sunstats.gz ...
Read summary statistics for 1217311 SNPs.
After merging with summary statistics, 759220 SNPs remain.
699925 SNPs with valid alleles.
/cbica/home/wenju/Project/ldsc/ldscore/irwls.py:161: FutureWarning: 'rcond' parameter will change to the default of machine precision times ``max(M, N)`` where M and N are the input matrix dimensions.
To use the future default and silence this warning we advise to pass 'rcond=None', to keep using the old, explicitly pass 'rcond=1'.
  coef = np.linalg.lstsq(x, y)
ERROR computing rg for phenotype 2/2, from file telomere_length.sunstats.gz.
Traceback (most recent call last):
  File "/cbica/home/wenju/Project/ldsc/ldscore/sumstats.py", line 410, in estimate_rg
    rghat = rg(looper, args, log, M annot, ref_id_cnames, w_ld_cname, i)
  File "/cbica/home/wenju/Project/ldsc/ldscore/sumstats.py", line 539, in _rg
    intercept_gencov=intercepts[2], n_blocks=n_blocks, twostep=args.two_step)
  File "/cbica/home/wenju/Project/ldsc/ldscore/regressions.py", line 705, in __init__
    np.multiply(hsq2_tot_delete_values, hsq2_tot_delete_values))
FloatingPointError: invalid value encountered in sqrt

```

We finally only included the results for longevity. The updated results are presented in the revised manuscript (**Line: 282**):

"Furthermore, we found that the WM BAG ($g_c = -0.23 \pm 0.10$; P-value=0.02; $N=28,967$ European ancestry) was negatively associated with longevity, defined as cases surviving at or beyond the age corresponding to the 99th survival percentile⁵¹."

Below are the new comments from this reviewer in this round:

The authors have improved the manuscript and have clarified aspects raised by the comments made during the first round of reviews. Rereading of the manuscript there are still a few points that seem important to investigate.

We are grateful for the reviewer's comprehensive comments in both the previous and current rounds, which we consider crucial for enhancing the scientific rigor of our study.

In this response letter and the latest revised manuscript, we have responded to these new comments by conducting additional experiments or offering clarification to address any misunderstandings arising from insufficient information in our initial versions.

* As the BAG computation is not the in par with the current literature it would be a good confirmation if a more precise estimate could be used and this would go a long way assessing the robustness of the results.

We concur with the reviewer's observation. As we demonstrated in the previous round of the review (several additional experiments in evaluating the impact of different elements on the GWAS signals), we would like to reaffirm that our MAE performance for ROI features in brain age prediction aligns with the results found in prior literature using the same feature sets (i.e., **low-dimensional ROIs features, not the CNN with voxel-wise imaging data**).

That being said, we fully addressed the reviewer's comments. We tested this by requesting a previous GM-BAG GWAS (<https://pubmed.ncbi.nlm.nih.gov/37165155/>; PMID: 37165155), which used CNN on voxel-wise images to achieve a lower MAE (~2.5 years) than our GM-BAG (4.39 years) using MUSE ROIs and Lasso regression (**details are presented below**).

With the reviewer's suggestion of the generalizability to an external dataset, we have performed **seven** sensitivity check analyses to scrutinize the robustness of our primary GWAS results (using both **P-value** and the **beta values** of the linear regression). All results are detailed in **Supplementary eText 1**.

There are several thoughts and considerations on why we did not employ CNN on voxel-wise imaging data (by ourselves) to derive the BAGs:

1. Employing a CNN on the voxel-wise images may yield a lower MAE, and this avenue warrants a more comprehensive exploration. However, we believe that delving into this aspect might extend beyond the scope of the current study, especially considering requesting & processing raw MRI scans for all three modalities and constructing a CNN with the same cross-validation procedure could significantly delay the progress of the present work. In our current analyses, we directly requested the processed diffusion (category code:134) and fMRI (data-field code: 25750) low-dimensional features from the UK Biobank.
2. We have employed four different machine learning models (both linear and non-linear) on these low-dimensional features to derive the BAGs and evaluated the impact of different elements (e.g., feature types, machine learning models, etc.) on the GWAS signals. With the two additional quality checks raised by the reviewer (CNN GWAS and Independent ADNI GWAS) presented in **Supplementary eText1**, we believe that we have done our best to address this comment.
3. We have shown **below** that, using the requested GWAS summary statistics, a lower MAE does not necessarily lead to stronger genetic signals. However, this needs to be more thoroughly investigated in future studies (using the same population, genetic data, etc).

The fact that "BAGs derived from a CNN with a lower MAE did not exhibit heightened sensitivity to disease effects such as AD (ref 35)"

We want to guide the reviewer to this reference paper

(<https://academic.oup.com/brain/article/143/7/2312/5863667>). In Fig. 3, the authors showed that a lower MAE may lead to smaller effect sizes, differentiating the disease group (i.e., AD, MCI, schizophrenia, and depression) vs. the healthy control group.

but this could still impact the genetic analyses that are at the core of the article results. Confirming (or not) results with a more accurate BAG seems important. If the results are only found for this specific BAG estimation, the interpretation will be profoundly altered.

To address this, we have contacted the authors (Dr. Yunpeng Wang) of this study (<https://pubmed.ncbi.nlm.nih.gov/37165155/>; PMID: 37165155) to request the GWAS summary statistics. Their study used neural networks and voxel-wise imaging to achieve an MAE below 2.5 years for their brain age prediction tasks, much lower than our MAE for GM-BAG (~4 years).

Using GM-BAG GWAS, we have performed an additional quality check analysis compared the GWAS results obtained from our Lasso MUSE ROIs (i.e., **GM-BAG-Lasso**) vs. the results from the CNN voxel-wise image-derived GM-BAG (i.e., **GM-BAG-CNN**). We quantitatively assess the **concordance rate** using both the P-value and beta values of the regression between the two GWASs. Our results showed high concordance rates between our GM-BAG-Lasso vs. the GM-BAG-CNN (detailed number are below).

The results are updated in the revised manuscript (**Supplementary eText 1**):

"**Machine learning model-specific GWAS**

We used GM-BAG to demonstrate this sensitivity check by comparing *i*) SVR using MUSE ROIs and *ii*) CNN using voxel images² (GWAS summary statistics shared by the authors) to our main results obtained from Lasso using MUSE ROIs.

P-value:

When comparing the SVR using MUSE ROIs (as replication, MAE=4.43 years) to Lasso using MUSE ROIs (as discovery, MAE=4.39 years), we found a 100% concordance rate of the SNPs identified for the GM-BAG GWAS. The BAGs derived from the two machine learning models were highly correlated ($r=0.99$; $P\text{-value}<1\times 10^{-10}$).

When comparing the CNN using voxel-wise images (MAE~2.5 years²) to Lasso using MUSE ROIs (as discovery), we found an 82.70% concordance rate (2533; 319 missing SNPs) after Bonferroni correction ($P\text{-value}<0.05/3063$).

β value:

When comparing the SVR using MUSE ROIs to Lasso using MUSE ROIs (as discovery), we found that the 3382 significantly replicated SNP ($P\text{-value}<0.05$) showed the same sign of β values from the linear regression models (Pearson's $r=1$; $P\text{-value}<1\times 10^{-10}$).

When comparing the CNN using voxel-wise images (MAE~2.5 years²) to Lasso using MUSE ROIs (as discovery), we found that all 2762 significantly replicated SNP ($P\text{-value}<0.05$) showed the same sign of β values from the linear regression models (Pearson's $r=1$; $P\text{-value}<1\times 10^{-10}$). (**Supplementary eFigure 5 and eFile 5**)."

To appreciate the help from Dr. Yunpeng Wang, we have added one sentence in the Acknowledge section (**Line: 1043**): "We are grateful to Dr. Yunpeng Wang for generously providing us with their GWAS summary statistics²¹ during the revision."

This additional quality check is compelling evidence supporting the conclusion that the GWAS signals exhibit high concordance despite the variance in MAE obtained from these machine-learning models.

We also did three additional analyses were conducted. We prefer to present these findings in the response letter rather than including them in the revised manuscript since the shared GWAS summary statistics (from Dr. Yunpeng Wang) are not publicly accessible and may entail additional considerations, such as authorship.

1) Genetic correlation between GM-BAG-Lasso and GM-BAG-CNN:

We obtained a very high genetic correlation ($g_c=0.5369$; $P\text{-value}=1.55\times E^{-22}$) between the two GM-BAGs. Detailed results are shown in the screenshot below:

```
Genetic Correlation
-----
Genetic Correlation: 0.5369 (0.855)
Z-score: 9.7676
P: 1.5505e-22

Summary of Genetic Correlation Results
p1      p2      rg      se      z      p      h2_obs  h2_obs_se  h2_int  h2_int_se  gcov_int  gcov_int_se
/cbica/home/wenju/Reproducible_paper/BrainAge/output/GM45/muse/output/gc/muse.sunstats.gz  cnn_bag.sunstats.gz  0.5369  0.055  9.7676  1.5505e-22  0.1932  0.0237  1.0087  0.0071  0.2569  0.0062

Analysis Finished at Wed Nov  8 11:47:17 2023
Total time elapsed: 17.66s
```

2) Using LDSC to derive SNP-based heritability (h^2) estimate for a fair comparison:

Dr. Yunpeng Wang's GM-BAG-CNN paper reported a heritability of 0.27 ± 0.036 using LDSC. To fairly compare our GM-BAG-Lasso, we derived the LDSC-based estimate: $h^2=0.30\pm 0.03$ (Supplementary eTable 4). Our result is slightly higher than the GM-BAG-CNN. Regarding the reviewer's comment on the SNP-based heritability estimate, we will provide more details below in that specific comment.

3) Genetic correlation for GM-BAG-Lasso and GM-BAG-CNN vs. other traits in Fig. 4a:

For this analysis, we found two observations:

GM-BAG-Lasso obtained higher genetic correlations with several traits than GM-BAG-CNN (see screenshot below):

	↕ group	↕ trait	↕ gc_mean	↕ gc_std	↕ Z	▲ P
4	GM-BAG-Lasso	AD1	0.40010	0.03920	10.19760	0.00000
24	GM-BAG-Lasso	SCZ1	0.49080	0.07670	6.40170	0.00000
10	GM-BAG-Lasso	ASD1	0.31240	0.06710	4.65770	0.00000
5	GM-BAG-CNN	AD1	0.22310	0.05740	3.88480	0.00010
14	GM-BAG-Lasso	ASD3	-0.20780	0.05920	-3.50960	0.00040
GM-BAG-CNN	SCZ1	0.33270	0.11730	2.83700	0.00460
31	GM-BAG-CNN	education	-0.15930	0.05730	-2.78140	0.00540
15	GM-BAG-CNN	ASD3	-0.21330	0.07990	-2.66940	0.00760
20	GM-BAG-Lasso	OCD	-0.17620	0.06650	-2.65020	0.00800
13	GM-BAG-CNN	ASD2	-0.14140	0.06720	-2.10280	0.03550
1	GM-BAG-CNN	ADHD	0.11110	0.06260	1.77330	0.07620
11	GM-BAG-CNN	ASD1	0.14910	0.08490	1.75650	0.07900
GM-BAG-CNN	intellige...	-0.08310	0.04880	-1.70250	0.08870
GM-BAG-Lasso	education	-0.07640	0.04820	-1.58530	0.11290
28	GM-BAG-Lasso	intellige...	-0.06800	0.04420	-1.53790	0.12410
2	GM-BAG-Lasso	AD	0.15950	0.10430	1.52930	0.12620
6	GM-BAG-Lasso	AD2	-0.07800	0.05170	-1.50830	0.13150
17	GM-BAG-CNN	BIP	0.06620	0.04540	1.45870	0.14460
26	GM-BAG-Lasso	SCZ2	0.08790	0.06160	1.42710	0.15350
8	GM-BAG-Lasso	ASD	0.07180	0.05240	1.36850	0.17120
12	GM-BAG-Lasso	ASD2	-0.07130	0.05700	-1.25040	0.21110
27	GM-BAG-CNN	SCZ2	-0.09320	0.07510	-1.24130	0.21450
22	GM-BAG-Lasso	SCZ	0.05430	0.04740	1.14580	0.25190
0	GM-BAG-Lasso	ADHD	0.04780	0.04900	0.97670	0.32870
16	GM-BAG-Lasso	BIP	0.03140	0.03890	0.80730	0.41950
18	GM-BAG-Lasso	MDD	0.06370	0.08030	0.79320	0.42760
19	GM-BAG-CNN	MDD	0.06370	0.08030	0.79320	0.42760
3	GM-BAG-CNN	AD	0.09980	0.14740	0.67690	0.49840
7	GM-BAG-CNN	AD2	-0.02470	0.06110	-0.40370	0.68640
21	GM-BAG-CNN	OCD	-0.03190	0.09140	-0.34920	0.72690
23	GM-BAG-CNN	SCZ	0.01060	0.05550	0.19080	0.84870
9	GM-BAG-CNN	ASD	0.00340	0.06860	0.05010	0.96000

We found that GM-BAG-Lasso showed the four highest g_c estimates, especially for these significant results after Bonferroni correction ($P\text{-value} < 0.05/16 = 0.003$). We hypothesize that this phenomenon arises because a neural network with a reduced MAE emphasizes optimizing data fitting, potentially at the expense of predictive power across other domains (e.g., disease effects or cognitive scores). We have observed a similar pattern in another study that compares the predictive capability of Polygenic Risk Scores (PRS), which is currently under review. A higher incremental R^2 does not lead to higher prediction power for classifying chronic diseases vs. controls.

Comparison within the same traits (see screenshot below):

↕ group	▲ trait	↕ gc_mean	↕ gc_std	↕ Z	↕ P
GM-BAG-Lasso	AD	0.15950	0.10430	1.52930	0.12620
GM-BAG-CNN	AD	0.09980	0.14740	0.67690	0.49840
GM-BAG-Lasso	AD1	0.40010	0.03920	10.19760	0.00000
GM-BAG-CNN	AD1	0.22310	0.05740	3.88480	0.00010
GM-BAG-Lasso	AD2	-0.07800	0.05170	-1.50830	0.13150
GM-BAG-CNN	AD2	-0.02470	0.06110	-0.40370	0.68640
GM-BAG-Lasso	ADHD	0.04780	0.04900	0.97670	0.32870
GM-BAG-CNN	ADHD	0.11110	0.06260	1.77330	0.07620
GM-BAG-Lasso	ASD	0.07180	0.05240	1.36850	0.17120
GM-BAG-CNN	ASD	0.00340	0.06860	0.05010	0.96000
GM-BAG-Lasso	ASD1	0.31240	0.06710	4.65770	0.00000
GM-BAG-CNN	ASD1	0.14910	0.08490	1.75650	0.07900
GM-BAG-Lasso	ASD2	-0.07130	0.05700	-1.25040	0.21110
GM-BAG-CNN	ASD2	-0.14140	0.06720	-2.10280	0.03550
GM-BAG-CNN	ASD3	-0.21330	0.07990	-2.66940	0.00760
GM-BAG-Lasso	ASD3	-0.20780	0.05920	-3.50960	0.00040
GM-BAG-Lasso	BIP	0.03140	0.03890	0.80730	0.41950
GM-BAG-CNN	BIP	0.06620	0.04540	1.45870	0.14460
GM-BAG-Lasso	MDD	0.06370	0.08030	0.79320	0.42760
GM-BAG-CNN	MDD	0.06370	0.08030	0.79320	0.42760
GM-BAG-Lasso	OCD	-0.17620	0.06650	-2.65020	0.00800
GM-BAG-CNN	OCD	-0.03190	0.09140	-0.34920	0.72690
GM-BAG-CNN	SCZ	0.01060	0.05550	0.19080	0.84870
GM-BAG-Lasso	SCZ	0.05430	0.04740	1.14580	0.25190
GM-BAG-Lasso	SCZ1	0.49080	0.07670	6.40170	0.00000
GM-BAG-CNN	SCZ1	0.33270	0.11730	2.83700	0.00460
GM-BAG-Lasso	SCZ2	0.08790	0.06160	1.42710	0.15350
GM-BAG-CNN	SCZ2	-0.09320	0.07510	-1.24130	0.21450
GM-BAG-Lasso	education	-0.07640	0.04820	-1.58530	0.11290
GM-BAG-CNN	education	-0.15930	0.05730	-2.78140	0.00540
GM-BAG-CNN	intelligence	-0.08310	0.04880	-1.70250	0.08870
GM-BAG-Lasso	intelligence	-0.06800	0.04420	-1.53790	0.12410

Specifically, after correcting multiple comparisons ($P < 0.05/16 = 0.003$), there are several cases where

- both GM-BAGs achieved significant genetic correlations within the same traits (i.e., AD1)
- GM-BAG-Lasso is significant but not GM-BAG-CNN (i.e., ASD1, ASD3, SCZ1)

It's important to highlight that these comparisons were made with GM-BAG derived from different training datasets, all sourced from UK Biobank, which introduced variations in sample sizes (we had 31,557 UK individuals, while the CNN paper had 28,104 participants), as well as differences in cross-validation procedures and the covariates included in the linear regression. Consequently, conducting future investigations with the same study population, cross-validation methods, and consistent GWAS models will be necessary to provide a more accurate assessment of this aspect.

It's worth noting that we requested processed diffusion and fMRI features from UK Biobank but not the raw MRI scans. This is why we consider applying, processing, and constructing the CNN for all three modalities to be beyond the scope of the present study. We hope our comprehensive explanation and experiments will address the reviewer's concerns.

An independent validation is probably needed to assert the solidity of the results. This could be done with an number of datasets including ADNI, PREVENT-AD, and others in Europe. This is a valuable suggestion. In our initial analysis, we did incorporate additional independent datasets, such as the ADNI study, into our manuscript. Indeed, we have access to whole-genome sequencing (WGS) data collated by the ADSP consortium (<https://www.nia.nih.gov/research/dn/alzheimers-disease-sequencing-project-consortia>) and AI4AD consortium (led by Paul Thompson: <http://ai4ad.org/>) for ADNI subjects. It's worth noting that the publicly available ADNI data from the LONI server only encompasses genotype data.

We didn't include this in our initial submission for several reasons. In the latest revised manuscript, we included these results based on the reviewer's comments:

- We only have MUSE ROIs processed for ADNI (not for diffusion and fMRI features with the same pipeline as UKBB). Therefore, we only use GM-BAG to demonstrate this generalizability ability.
- The age distribution significantly differs between the UKBB and the ADNI study – ADNI people are significantly older than UKBB people. The trained brain age prediction model on UKBB (MAE=4.39) cannot be well generalized to ADNI – resulting in a much larger MAE in the ADNI study (MAE=9.16). Therefore, we trained the GM-BAG model from scratch using ADNI healthy control subjects and applied the model to MCI and AD patients. This resulted in MAE=4.24 years in the nested cross-validation procedure using Lasso regression and MUSE ROIs (**GM-BAG-ADNI**).
- We then performed the GM-BAG-ADNI GWAS using ADNI WGS data. The Manhattan and QQ plots are shown in **Supplementary eFigure 7**, along with the plots from the UKBB imputed genotyping data (**GM-BAG-UKBB**).

Visually, the genetic signals identified in UKBB can not be generalized to the ADNI data, potentially due to the much smaller sample size ($N=1104$). We then quantified the **concordance rate** on both **P-value** and the **Beta values** as an additional sensitivity check analysis (Independent dataset for GM-BAG GWAS using ADNI WGS data). We found that despite a low concordance rate based on P-values, the beta values showed a high concordance between the UKBB and ADNI GWASs for significant SNPs (P-value < 0.05). Detailed numbers are presented in **Supplementary eText 1** and below:

"ADNI WGS GWAS

P-value:

We evaluated the generalizability of the GM-BAG GWAS findings from the UKBB dataset to the ADNI whole-genome sequencing (WGS) data. When considering the concordance rate based on P-values, we observed a high concordance rate (83.57 %) for the GWASs performed using the ADNI WGS data ($N=1104$) as a replication dataset ($N=2583$ out of 3091; 291 SNPs missing from the ADNI data) using a nominal P-value threshold. No SNPs survived the Bonferroni correction.

β value:

However, it's noteworthy that the β values of these significant SNPs exhibited a significant correlation ($r=0.83$; $P\text{-value}<1\times 10^{-10}$) between the two datasets. This observation underscores the importance of collecting genetic data within specific disease populations and throughout the entire lifespan (**Supplementary eFigure 7 and eFile 7**)."

To conclude, we have performed seven quality check analyses to scrutinize the robustness of our three primary GWASs. We believe that these are sufficient to convey the main message of our GWAS. In addition, we decided to move the sensitivity check analysis of the GWAS to **Supplementary eText 1** due to the length of the current paper. In the main manuscript, we summarized the main messages of the sensitivity checks (**Line: 178-194**):

" We showed the robustness of our GWAS findings with several different approaches. We first calculated the genomic inflation factor (λ) and the LDSC intercept (b) for the GWAS of GM-BAG ($\lambda=1.118$; $b=1.0016\pm 0.0078$), WM-BAG ($\lambda=1.124$; $b=1.0187\pm 0.0073$), and FC-BAG ($\lambda=1.046$; $b=1.0039\pm 0.006$). All LDSC intercepts were close to 1, indicating no substantial genomic inflation. The individual Manhattan and QQ plots of the three GWASs are presented in **Supplementary eFigure 1** and are publicly available at the MEDICINE knowledge portal: <https://labs.loni.usc.edu/medicine>. We also checked the robustness of the main GWASs using the European populations (**Fig. 2A**) via seven sensitivity analyses (**Method 4A**). Overall, the primary GWASs were robust across sexes (female vs. male), random splits, imaging features (ROI vs. voxel-wise images), GWAS methods (linear vs. mixed linear model⁴⁰), and machine learning methods (Lasso regression vs. SVR. vs. CNN²¹); however, their generalizability to non-European populations ($4646 < N < 5091$) and independent disease-specific populations (i.e., ADNI⁴¹, $N=1104$) is limited potentially due to the small sample sizes. It's worth noting that their β values compared to the primary GWASs were significantly correlated: $r=0.83$ for ADNI and $r=0.97-0.99$ for the non-European populations. (**Supplementary eText 1, Supplementary eFile 1-7, and Supplementary eFigure 1-7**). All subsequent post-GWAS analyses were conducted using the main GWAS results of European ancestry."

* It is hard to understand why MR identifies a potential risk of AD on WM-BAG but not on GM-BAG given the well known mechanisms for GM reduction with AD progression.

We thank the reviewer for this remark. There are multiple potential explanations from our perspectives:

- AD is not merely a gray matter disease, although brain atrophy in the medial temporal lobe may be the most prominent biomarker in the AD field. White matter integrity and

functional network disruption have also proven to be early biomarkers in AD. Conversely, brain aging is a multifaceted process that involves multiple organ systems (PMID: 37024597) and many chronic diseases – not only AD contributes to brain atrophy. One possibility is that other chronic diseases "horizontally" affect brain aging-induced atrophy.

- Mendelian randomization relies on several underlying IV assumptions dependent on the chosen instrumental variables. In our MR analyses, we employed 2SampleMR, which specifically considers the significant SNPs ($P < 5 \times 10^{-8}$) after LD clumping. It's important to note that the AD exposure variables used in our analysis are from an earlier GWAS (PMID: 24162737, 2013, Nature Genetics), as the MR methods we utilized require no overlapping samples between the exposure and outcome variables. Most recent and larger-scale AD GWASs predominantly utilize UK Biobank populations, which can pose challenges when applying MR techniques. We chose not to do an additional analysis using more recent AD GWAS (potentially larger samples lead to more valid IVs) in order not to "cherry-pick" the significant signal for AD and GM-BAG causality.
- We present the results based on the data, allowing the data to guide our interpretation. As we have emphasized in the previous revision, it is essential to exercise caution when interpreting these causal results, as noted in the Limitation section. In the newly revised manuscript, we have included an additional sentence in the caption of **Fig. 5**:
 - "Interpreting these potential causal relationships should be cautiously undertaken despite our efforts to perform multiple sensitivity checks to evaluate the possible violations of underlying assumptions."

* While significant, PRS-CS R2 increase of 2.17%, 1.85%, and 0.19% for BAGs phenotypes are very small values, the authors might want to update the description of the results to emphasize the small effect sizes

We agree with the reviewer on this observation. Our results are consistent with the literature for the PRS derived from brain IDPs, evidenced in several previous papers from the co-author (Dr. Bingxin Zhao). In the revised manuscript, we had one sentence to state the limited prediction power of these PRSs (**Line: 324-328**):

"Overall, the predictive power of PRS is not high, in line with earlier discoveries involving raw imaging-derived phenotypes, as demonstrated in Zhao et al.¹³. The authors developed PRSs for seven selective brain regions, which explained roughly 1.18% to 3.93% of the phenotypic variance associated with these traits."

* The effort in reproducibility of the work are important and excellent, and this reviewer wholeheartedly commend the authors for releasing part of their code on GitHub.

We appreciate the reviewer for bringing up this important point. As a dedicated proponent of open science in neuroimaging, machine learning, and genetics/genomics, W.J. is proud to ensure that our research is publicly accessible to the community. This includes sharing our source code on GitHub and disseminating our findings through the MEDICINE knowledge portal.

Other :

* The new manuscript [https://www.biorxiv.org/content/10.1101/2022.09.16.508329v2 ... ?](https://www.biorxiv.org/content/10.1101/2022.09.16.508329v2)

We are sorry for the confusion. We are not sure where we referred to this manuscript in our previous manuscripts. In the revised manuscript, we indeed cited this paper as this paper details the genetic quality checks of the WGS data from ADNI (**Supplementary eText 1**).

* Cheverud conjecture doesn't seem to be realized here with much larger genetic correlations than phenotypic correlations

We agree that our results do not show that genetic correlation is larger than phenotypic correlation.

Based on previous literature: "The observation that genetic correlations usually mirror phenotypic correlations is known as "Cheverud's Conjecture" (PMID 28581166)". Wikipedia link: https://en.wikipedia.org/wiki/Genetic_correlation. This does not imply that the genetic correlation's magnitude should be larger than the phenotypic correlation's.

In the revised manuscript, we deleted the specific sentence regarding this.

* The fact that the genetic correlations is found between GM-BAG and disease sub-types rather than with disease diagnoses might simply reflect that by clustering the population in groups for which several phenotypic dimensions are capture, there is simply more chance to have significant correlations. The authors could assess the correction needed on "null" data (e.g. other populations for which correlation is not expected).

Thank the reviewer for this comment. We want to clarify this question:

- The disease subtypes were not defined in UKBB populations, as we know that UKBB is a general population, not a disease-specific clinical population. For example, we trained the AI clustering model (Surreal-GAN: <https://openreview.net/forum?id=nf3A0WZsXS5>) in ADNI data for Alzheimer's, and we then applied the model to UKBB to derive these two dimensions (AD1 and AD2)
- LDSC corrects (or cancels out) population overlaps for "unbiased" genetic correlation estimates between the two traits (<https://pubmed.ncbi.nlm.nih.gov/25642630/>).
- Potential bias might emerge during the GWAS summary statistics harmonization process for both traits. Nevertheless, we obtained the GWAS summary statistics directly from previous literature or the GWAS platform (e.g., GWAS Catalog). Ensuring that all traits have the same set of SNPs after harmonization is challenging due to variations in genetic sequencing, allele frequencies, and other factors used across different studies. This is indeed an interesting scientific question to investigate, but we sense that this is out of the scope of the current study.

* The heritability numbers are rather impressive : is really 47% of the variance of GM-BAG explained by SNPs ? Could the authors explain why this is plausible (or is the number due to a methodological issue?)

We thank the reviewer for this remark! Our answer is: the data (individual genotype data vs. GWAS summary statistics), LD reference panel (external reference panel from 1000 Genome vs. local UKBB panel), allele frequency (common vs. rare variants), and method used (LDSC vs. GCTA) to compute the SNP-based heritability **all** play a role. See the detailed explanation below:

Indeed, we also observed the difference between our SNP-based heritability and previous GM-BAG GWAS. For example, in an early GM-BAG GWAS paper (PMID:31551603, Nature Neuroscience), Kaufmann et al. obtained an SNP-based heritability of 0.24 (0.47 in our GM-BAG results). Similarly, two other studies estimated the SNP-based heritability using LDSC and obtained similar results: Leonardsen et al. (PMID: 37165155, Molecular Psy, 2023) reported a 0.27 estimate, and Jonsson et al. (PMID: 31776335, Nature Communications, 2019) reported a 0.19 estimate.

All three studies used LDSC, which leverages GWAS summary statistics (not the raw genotype data) and probably an external LD reference panel (by default in LDSC from 1000 Genome EUR ancestry). However, we used GCTA, which utilizes individual genotype data to estimate the SNP-based heritability, and claims in the original paper (<https://www.ncbi.nlm.nih.gov/pmc/articles/PMC3014363/>) that it accounts for part of the "missing heritability".

Supplementary eTable 4 provided the SNP-based heritability using LDSC for GM-BAG (0.30±0.03) to support this further. This value is still slightly larger than Kaufmann's and Jonsson's and comparable with Leonardsen's estimate but lower than the one we obtained using GCTA for GM-BAG (0.47).

More convincingly, this observation also happens in several large-scale imaging-derived phenotype (IDP) GWAS. For example, Elliot et al. (Nature, 2018, PMID: 30305740, 2018) used LDSC to estimate the SNP-based heritability for multimodal IDPs (see figure below):

[redacted]

As we can see, the mean SNP-based heritability for T1 IDP is smaller than 0.4. In a follow-up study conducted by the co-author (Dr. Bingxin Zhao, Nature Genetics, PMID: 31676860, 2019), we used GCTA to estimate the T1 MRI-derived IDPs and obtained a higher mean SNP-based heritability. In the original paper, they wrote: "The h^2 estimates, standard errors, and raw and Bonferroni-corrected P values from the one-sided likelihood ratio tests are provided in Supplementary Table 1. In the combined data, h^2 of most ROIs was significant after Bonferroni correction for multiple testing (mean $h^2 = 0.40$, h^2 range = (0.12, 0.72), standard error = 0.15)."

Therefore, this discrepancy is mainly due to the software of choice (LDSC vs. GCTA). In the revised manuscript, we added two sentences to discuss this (**Line: 172-177**):

" Our GM-BAG showed a higher SNP-based heritability than several previous GM-BAG GWAS^{9,21,22} ($0.19 < h^2 < 0.27$), which used the linkage disequilibrium score regression (LDSC) software³⁷. LDSC uses GWAS summary statistics but not the individual genotype data as in GCTA. This discrepancy may depend on the choice of methods, genetic data employed, underlying statistical assumptions, and allele frequency^{38,39}. "

* It is hard to understand the overfitting issue "..., with Lasso regression, the cross-validated test result (CV test) obtained an MAE of 4.94 for all 192 FA/MD/ODI/NDI metrics, but the independent test result (Ind. test) obtained an MAE of 1.66."

We believe there is a critical typo here from the reviewer. Did the reviewer mean: "with Lasso regression, the cross-validated test result (CV test) obtained an MAE of 4.142 for all 192 FA/MD/ODI/NDI metrics, but the independent test result (Ind. test) obtained an MAE of 21.66." Otherwise, the comment does not make sense. We were referring to the overfitting that the *CV Test* MAE is 4.142, but it dropped to MAE=21.66 in the independent test data (*Ind. Test*). This big drop indicates the model's poor generalizability to unseen data. More details of the CV are explained below.

Can the author provide insight on how overfitting can occur in completely independent data ? what were the cross validation results (i.e. results on validation data in the CV loop) ?

As we have a large sample size in our brain age experiment, we chose to have a strict cross-validation procedure by holding out an independent test dataset – independent unseen data is always preferable when sample size allows testing the model's generalizability.

Specifically, as detailed in **Method 1**, we first randomly sub-sampled 500 (250 females) participants within each decade's range from 44 to 84 years old, resulting in the same 4000 participants for GM, WM, and FC-IDP. These 4000 participants were defined as "*Training/validation/test* dataset". This is where we performed the **nested cross-validation** (i.e., outer & inner loop) procedure:

- Outer loop: the outer loop CV was performed for 100 repeated random splits: 80% of the data were used for *training+validation* (0.8x4000), and the remaining 20% was used for *testing*, resulting in the *CV test* (the mean of the 100 repetitions is presented in **Supplementary eTable 1**). We have added the std information (mean±std) in **Supplementary eTable 1** for these *CV test* results in the newly revised manuscript.
- For the *training+validation* data (0.8x4000), we performed a 10-fold CV for grid-searching the hyperparameters of SVR, if applicable.

Thanks to the reviewer, we realized that the sentence regarding our nested cross-validation has some typos (misleading). In the revised manuscript, we corrected it (**Line: 610-613**):

"In detail, the outer loop CV was conducted with 100 repeated random splits: 80% of the data served for training and validation, while the remaining 20% was allocated for testing. In the inner loop, if applicable, a 10-fold CV was performed for a grid search for hyperparameter tuning of the machine learning models."

All the rest of the population in the UKBB that were not in the training+validation+test population was defined as the *independent test* dataset, resulting in one value for *Ind. Test* after applying the trained model to this dataset, as shown in **Supplementary eTable 1**.

We hope this explanation is clear now for the CV procedure used in our experiments. This nested CV is implemented in our open-source software on GitHub:

<https://anbai106.github.io/mlni/>.

Specifically, the outer loop of the nested CV is implemented in line 55 at

<https://github.com/anbai106/mlni/blob/master/mlni/regression.py>:

```

54
55     self._validation = RepeatedHoldOut(self._algorithm, n_iterations=self._n_iterations, test_size=self._test_size)

```

The inner loop of the nest CV is implemented in line 169 (function: *evaluate*) at

<https://github.com/anbai106/mlni/blob/master/mlni/regression.py>:

```

"""
"""
"""
"""
"""
"""
"""
"""
"""
"""
"""
"""
"""
"""
"""
"""
"""
"""
"""
"""
"""
"""
"""
"""
"""
"""
"""
"""
"""
"""
"""
"""
"""
"""
"""
"""
"""
"""
"""
"""
"""
"""
"""
"""
"""
"""
"""
"""
"""
"""
"""
"""
"""
"""
"""
"""
"""
"""
"""
"""
"""
"""
"""
"""
"""
"""
"""
"""
"""
"""
"""
"""
"""
"""
"""
"""
"""
"""
"""
"""
"""
"""
"""
"""
"""
"""
"""
"""
"""
"""
"""
"""
"""
"""
"""
"""
"""
"""
"""
"""
"""
"""
"""
"""
"""
"""
"""
"""
"""
"""
"""
"""
"""
"""
"""
"""
"""
"""
"""
"""
"""
"""
"""
"""
"""
"""
"""
"""
"""
"""
"""
"""
"""
"""
"""
"""
"""
"""
"""
"""
"""
"""
"""
"""
"""
"""
"""
"""
"""
"""
"""
"""
"""
"""
"""
"""
"""
"""
"""
"""
"""
"""
"""
"""
"""
"""
"""
"""
"""
"""
"""
"""
"""
"""
"""
"""
"""
"""
"""
"""
"""
"""
"""
"""
"""
"""
"""
"""
"""
"""
"""
"""
"""
"""
"""
"""
"""
"""
"""
"""
"""
"""
"""
"""
"""
"""
"""
"""
"""
"""
"""
"""
"""
"""
"""
"""
"""
"""
"""
"""
"""
"""
"""
"""
"""
"""
"""
"""
"""
"""
"""
"""
"""
"""
"""
"""
"""
"""
"""
"""
"""
"""
"""
"""
"""
"""
"""
"""
"""
"""
"""
"""
"""
"""
"""
"""
"""
"""
"""
"""
"""
"""
"""
"""
"""
"""
"""
"""
"""
"""
"""
"""
"""
"""
"""
"""
"""
"""
"""
"""
"""
"""
"""
"""
"""
"""
"""
"""
"""
"""
"""
"""
"""
"""
"""
"""
"""
"""
"""
"""
"""
"""
"""
"""
"""
"""
"""
"""
"""
"""
"""
"""
"""
"""
"""
"""
"""
"""
"""
"""
"""
"""
"""
"""
"""
"""
"""
"""
"""
"""
"""
"""
"""
"""
"""
"""
"""
"""
"""
"""
"""
"""
"""
"""
"""
"""
"""
"""
"""
"""
"""
"""
"""
"""
"""
"""
"""
"""
"""
"""
"""
"""
"""
"""
"""
"""
"""
"""
"""
"""
"""
"""
"""
"""
"""
"""
"""
"""
"""
"""
"""
"""
"""
"""
"""
"""
"""
"""
"""
"""
"""
"""
"""
"""
"""
"""
"""
"""
"""
"""
"""
"""
"""
"""
"""
"""
"""
"""
"""
"""
"""
"""
"""
"""
"""
"""
"""
"""
"""
"""
"""
"""
"""
"""
"""
"""
"""
"""
"""
"""
"""
"""
"""
"""
"""
"""
"""
"""
"""
"""
"""
"""
"""
"""
"""
"""
"""
"""
"""
"""
"""
"""
"""
"""
"""
"""
"""
"""
"""
"""
"""
"""
"""
"""
"""
"""
"""
"""
"""
"""
"""
"""
"""
"""
"""
"""
"""
"""
"""
"""
"""
"""
"""
"""
"""
"""
"""
"""
"""
"""
"""
"""
"""
"""
"""
"""
"""
"""
"""
"""
"""
"""
"""
"""
"""
"""
"""
"""
"""
"""
"""
"""
"""
"""
"""
"""
"""
"""
"""
"""
"""
"""
"""
"""
"""
"""
"""
"""
"""
"""
"""
"""
"""
"""
"""
"""
"""
"""
"""
"""
"""
"""
"""
"""
"""
"""
"""
"""
"""
"""
"""
"""
"""
"""
"""
"""
"""
"""
"""
"""
"""
"""
"""
"""
"""
"""
"""
"""
"""
"""
"""
"""
"""
"""
"""
"""
"""
"""
"""
"""
"""
"""
"""
"""
"""
"""
"""
"""
"""
"""
"""
"""
"""
"""
"""
"""
"""
"""
"""
"""
"""
"""
"""
"""
"""
"""
"""
"""
"""
"""
"""
"""
"""
"""
"""
"""
"""
"""
"""
"""
"""
"""
"""
"""
"""
"""
"""
"""
"""
"""
"""
"""
"""
"""
"""
"""
"""
"""
"""
"""
"""
"""
"""
"""
"""
"""
"""
"""
"""
"""
"""
"""
"""
"""
"""
"""
"""
"""
"""
"""
"""
"""
"""
"""
"""
"""
"""
"""
"""
"""
"""
"""
"""
"""
"""
"""
"""
"""
"""
"""
"""
"""
"""
"""
"""
"""
"""
"""
"""
"""
"""
"""
"""
"""
"""
"""
"""
"""
"""
"""
"""
"""
"""
"""
"""
"""
"""
"""
"""
"""
"""
"""
"""
"""
"""
"""
"""
"""
"""
"""
"""
"""
"""
"""
"""
"""
"""
"""
"""
"""
"""
"""
"""
"""
"""
"""
"""
"""
"""
"""
"""
"""
"""
"""
"""
"""
"""
"""
"""
"""
"""
"""
"""
"""
"""
"""
"""
"""
"""
"""
"""
"""
"""
"""
"""
"""
"""
"""
"""
"""
"""
"""
"""
"""
"""
"""
"""
"""
"""
"""
"""
"""
"""
"""
"""
"""
"""
"""
"""
"""
"""
"""
"""
"""
"""
"""
"""
"""
"""
"""
"""
"""
"""
"""
"""
"""
"""
"""
"""
"""
"""
"""
"""
"""
"""
"""
"""
"""
"""
"""
"""
"""
"""
"""
"""
"""
"""
"""
"""
"""
"""
"""

```